# Evaluating Efficient Performance Estimators of Neural Architectures

**Xuefei Ning**[1♯]    **Changcheng Tang**[2]    **Wenshuo Li**[1]    **Zixuan Zhou**[1]

**Shuang Liang**[2]    **Huazhong Yang**[1†]    **Yu Wang**[1‡]

Department of Electronic Engineering, Tsinghua University[1]
Novauto Technology Co. Ltd.[2]
♯foxdoraame@gmail.com, †yanghz@tsinghua.edu.cn, ‡yu-wang@tsinghua.edu.cn

## Abstract

Conducting efficient performance estimations of neural architectures is a major challenge in neural architecture search (NAS). To reduce the architecture training costs in NAS, **one-shot** estimators (OSEs) amortize the architecture training costs by sharing the parameters of one "supernet" between all architectures. Recently, **zero-shot** estimators (ZSEs) that involve no training are proposed to further reduce the architecture evaluation cost. Despite the high efficiency of these estimators, the quality of such estimations has not been thoroughly studied. In this paper, we conduct an extensive and organized assessment of OSEs and ZSEs on five NAS benchmarks: NAS-Bench-101/201/301, and NDS ResNet/ResNeXt-A. Specifically, we employ a set of NAS-oriented criteria to study the behavior of OSEs and ZSEs, and reveal their biases and variances. After analyzing *how and why* the OSE estimations are unsatisfying, we explore *how to mitigate* the correlation gap of OSEs from three perspectives. Through our analysis, we give out suggestions for future application and development of efficient architecture performance estimators. Furthermore, the analysis framework proposed in our work could be utilized in future research to give a more comprehensive understanding of newly designed architecture performance estimators. The code is available at `https://github.com/walkerning/aw_nas` [24].

## 1   Introduction

Neural architecture search (NAS) can automatically discover architectures that outperform the hand-crafted ones for various applications [48, 10, 11]. Early NAS methods [48, 30] suffer from an extremely heavy computational burden, and can take tens of thousands of GPU hours to run. One of the major reasons for the computational challenge of NAS is that evaluating each candidate architecture is slow, which includes a full training and testing process. In the past years, studies [2, 27, 3, 5, 8, 45, 20, 1] have been focusing on developing more efficient performance estimators of neural architectures.

**One-shot Estimator (OSE)** Traditional NAS methods [48, 30, 2] conduct a costly separate training process to acquire the suitable parameters to evaluate each candidate architecture. To make NAS computationally tractable, ENAS [27] proposes the parameter-sharing technique to accelerate the architecture evaluation. Following this work, the parameter sharing technique is widely used for architecture search in different search spaces [39, 16] or incorporated with different search strategies [19, 16, 23, 40]. We refer to the parameter-sharing estimations as the "one-shot" estimations since it requires the training cost of *one* supernet.

How well the one-shot estimations are correlated with the standalone architecture performances is essential for the efficacy of NAS methods. Despite the widespread use of OSEs, studies [43] have

revealed that the OSE estimations might fail to reflect the true ranking of architectures. However, their experiments are conducted in a toy search space with only 32 architectures. In this work, we conduct a more comprehensive study on OSEs in five search spaces with distinct properties, including three topological search spaces (NAS-Bench-101 [41], NAS-Bench-201 [9], and NAS-Bench-301 [32]), and two non-topological search spaces [29] (NDS ResNet, and NDS ResNeXt-A). We further analyze *how and why* OSE estimations have bias and variance, and explore *how to improve* OSEs.

**Zero-shot Estimator (ZSE)** More recently, in order to further reduce the architecture evaluation cost, several studies [20, 1, 15, 17, 26, 6] introduce "zero-shot" estimators that involve no training. In this work, we study various ZSEs on several benchmarks and reveal their properties and weakness.

**Knowledge** Our work reveals pieces of knowledge on OSEs and ZSEs. First of all, some behaviors of OSEs and ZSEs vary across search spaces (Appendix A.1.1, Sec. 4.2). Some of the common knowledge for OSEs revealed by our work include 1) OSEs bias towards architectures with lower complexity in the early training phase [18]. And this bias can be alleviated to various extents with sufficient training in different spaces (Sec. 5.1). 2) OSEs have variance and can be mitigated to some extent (Sec. 5.3, Sec. 6.1). 3) Reducing the sharing extent of OSEs can potentially improve their ranking quality [47] (Sec. 6.3).

As for ZSEs, we reveal that 1) Current ZSEs cannot benefit from one-shot training. The ranking qualities of ZSEs utilizing high-order information (i.e., gradients) even degrade a lot after one-shot training (Sec. 4.2). 2) Parameter-level ZSEs adapted from pruning literature are not suitable for ranking architectures, and their ranking qualities cannot surpass those of parameter size (#Param) or #FLOPs (Sec. 4.2). 3) Existing ZSEs have improper biases, some overestimate linear architectures without skip connections, and some overestimate architectures with smaller kernel sizes and receptive fields (Sec. 5.2). 4) The relative effectiveness of ZSEs varies between search spaces, *relu_logdet* [21] is the best on the three topological search spaces, and *synflow* [33] is better on the two non-topological search spaces (Sec. 4.2). 5) Most ZSEs are not sensitive to the input data distribution: They get similar architecture rankings when using random noises as the input (Appendix B.1).

**Suggestions** Based on our experiments and analyses, we give out suggestions for future OSE applications. For example: 1) Longer training makes one-shot estimations better (Sec. 4.1); 2) Using one-shot loss instead of accuracy significantly improves the ranking qualities in the DARTS space [16] (Sec. 4.1); 3) One should use enough validation data for OSEs, instead of merely several batches as in ZSEs (Sec. 4.1); 4) Using temporal ensemble helps reduce the ranking instability, and brings non-negative improvements on the ranking quality in different search spaces (Sec. 6.1); 5) In search space with isomorphic architectures, augmenting the sampling strategy to improve the sampling fairness is essential to avoid overestimating simple architectures (Sec. 6.2); 6) Affine operation should not be used in batch normalization (BN) during supernet training (Sec. 6.3).

As for ZSEs, we point out several open research problems: 1) Is there a general ZSE suitable for different types of search spaces? 2) Do we need to make ZSEs utilize the input data information better, and how can we do that? 3) Can we develop ZSEs that distinguish top architectures better? We also list out some technical suggestions for improving ZSEs: 1) Future ZSEs should conduct architecture-level analysis instead of using parameter-level analysis (Sec. 4.2). 2) According to some prominent bias of existing ZSEs, we can add some structural knowledge into ZSE voting ensembles, e.g., receptive field analysis seems promising for improving *jacob_cov* or *relu_logdet* (Sec. 5.2). 3) In future developments of ZSEs, researchers should add two simple comparison baselines, #Params, and #FLOPs, as they are actually very competitive baseline ZSEs (Sec. 4.2).

Our work provides strong baselines and diagnosis tools for future research of architecture performance estimators, and we suggest future research to utilize these baselines and tools for a more comprehensive understanding of newly designed performance estimators.

**Analysis Framework** Our analysis framework of efficient architecture performance estimators is organized as follows. We first introduce the evaluation criteria for estimator quality in Sec. 3. And Sec. 4 presents the quality evaluation of multiple OSEs and ZSEs. Then, we conduct an organized analysis on *how and why* the OSE and ZSE estimations have biases and variances in Sec. 5. Specifically, their complexity-level, operation-level, and architecture-level biases are demonstrated and analyzed. And the stability of OSE accuracy and ranking along the training process are analyzed. And in Sec. 6, based on our analysis framework, we present several case studies on improving OSEs from three perspectives: i.e. reducing the variance, bias, and parameter sharing extent.

## 2 Related Work

### 2.1 Efficient Performance Estimators of Neural Architectures

**One-shot Estimators** The vanilla NAS method [48] trains each architecture for 50 epochs to acquire its suitable parameters, which makes the NAS process prohibitively costly. As a remedy, ENAS [27] proposes to amortize the separate training costs by sharing parameters among architectures. Specifically, ENAS constructs an over-parametrized supernet such that all architectures can be evaluated using its parameter subsets. Throughout the search process, the shared supernet parameters are updated on the training set, and an RNN controller is updated alternatively on the validation set.

There are two types of parameter-sharing methods: 1) One-shot NAS methods [3, 13] that first train a supernet and then conduct architecture search without further supernet tuning. 2) Non-one-shot methods [27, 16, 40] that conduct supernet training and architecture search (i.e. controller update) jointly. And this work focuses on evaluating the estimations of the "one-shot" supernet, since it is the cleaner case without the complexity of varying controller settings and possible controller-supernet co-adaption. In each supernet training step, S architectures are randomly sampled to process a batch of training data. Here S denotes the number of Monte-Carlo architecture samples. Then, the gradients of these architectures are averaged to update the supernet.

**Correlation of One-shot Estimators** There exist some studies that carry out correlation evaluation for one-shot estimators. Zhang et al. [45] compare the correlation of OSEs and their proposed hyper-network-based estimator. However, their work is not aiming for a large-scale evaluation of OSEs and ZSEs, thus they only evaluate the OSE correlations on one search space, and do not conduct further analysis. Yu et al. [43] conduct parameter sharing NAS in a toy RNN search space with only 32 architectures in total, and discover that the parameter sharing rankings do not correlate with the true rankings of architectures. Zela et al. [44] also report that the correlation of parameter-sharing estimations is not satisfying with a Spearman correlation coefficient between -0.25 and 0.3 on a larger search space with around 15k architectures. Pourchot et al. [28] evaluate the Spearman's ranking correlation of OSEs on NAS-Bench-101. Yu et al. [42] provide an analysis on how the heuristics and hyperparameters influence the supernet training on three benchmarks (i.e. NAS-Bench-101, NAS-Bench-201, and DARTS-NDS). But they only use the variants of Kendall's Tau as the evaluation criteria, and do not further explore the biases and failing reasons of OSEs. Zhang et al. [47] point out the instability and poor ranking correlation of OSEs, and claim that the high extent of parameter sharing causes the unsatisfying performance. However, they only conduct experiments on a small search space with about 200 architectures.

In this paper, we conduct a more comprehensive study of OSE behaviors across five search spaces, and further investigate how and why the OSE estimations are not satisfying. We also propose and compare several techniques to mitigate the OSE correlation gap.

**Zero-shot Estimators** More recently, in order to further reduce the architecture evaluation cost, several researches [20, 1] propose "zero-shot" estimators that conduct no training and use random initialized models to estimate architecture performances. Based on the observation that good architectures have distinct local jacobian on different images, Mellor et al. [20] propose an indicator based on input jacobian correlation. Lopes et al. [17] improve the above indicator by calculating the jacobian correlation with respect to the class. Abdelfattah et al. [1] adapt several ZSEs from the pruning literature, and claim that these adapted ZSEs can perform well on NAS-Bench-201. Lin et al. [15] define the expected Gaussian complexity to measure the network expressivity, and efficiently discover architectures with state-of-the-art accuracy on ImagetNet. With a small training overhead, Ru et al. [31] propose to evaluate an architecture's performance by its training speed. Concurrent to our work, White et al. [38] also evaluate various performance estimators on multiple benchmarks.

### 2.2 NAS Benchmarks

NAS benchmarks are proposed to enable researchers to verify the effectiveness of NAS methods efficiently. NAS-Bench-101 (NB101) [41] provides the performances of the 423k valid architectures in a cell-based search space. OSE cannot be easily applied for the whole NB101 search space due to its specific channel number rule. To reuse NB101 for benchmarking OSE, NAS-Bench-1shot1 (NB1shot) [44] picks out three sub-spaces of NB101, and a supernet can be easily constructed for these sub-spaces. In this work, we use the largest sub-space in NB1shot: NB1shot-3, and use the name

"NB101" to refer to it. Another benchmark, NAS-Bench-201 (NB201) [9], provides the performances of all the 15625 architectures in a single-cell search space. Previous tabular benchmarks exhaustively train all architectures in a search space much smaller than commonly-used ones (e.g. DARTS [16] with size over $10^{18}$). Recently, NAS-Bench-301 (NB301) [32] is proposed as a benchmark in the DARTS space. It adopts a surrogate-based methodology that predicts architecture performances with the performances of about 60k anchor architectures.

Besides these benchmarks on cell-based topological search spaces, we also experiment with two non-topological benchmarking search spaces [29], NDS ResNet, and NDS ResNeXt-A. The architectural decisions in these search spaces are the non-topological hyper-parameters of pre-defined blocks, including kernel size, width, depth, convolution group number, and so on. The properties of these benchmarking search spaces are summarized in Appendix Tab. A1.

## 3 Evaluation Criteria

This section introduces the major evaluation criteria used in our analysis framework, while the analysis criteria and methods of ranking bias and variance will be introduced in Sec. 5. We denote the total number of architectures as $M$, the true (ground-truth, GT) performances and approximated estimated scores of architectures $\{a_i\}_{i=1,\cdots,M}$ as $\{y_i\}_{i=1,\cdots,M}$ and $\{s_i\}_{i=1,\cdots,M}$, respectively, and the ranking of the true and estimated score $y_i, s_i$ as $r_i, n_i \in \{1, \cdots, M\}$, respectively ($r_i = 1$ indicates that $a_i$ is the best architecture). The correlation criteria used in our framework are

- Pearson coefficient of linear correlation (LC): $\text{corr}(y, s)/\sqrt{\text{corr}(y, y)\text{corr}(s, s)}$.
- Kendall's Tau ranking correlation (KD $\tau$): The relative difference of concordant pairs and discordant pairs $\sum_{i<j} \text{sgn}(y_i - y_j)\text{sgn}(s_i - s_j)/\binom{M}{2}$.
- Spearman's ranking correlation (SpearmanR): The pearson correlation coefficient between the ranking variables $\text{corr}(r, n)/\sqrt{\text{corr}(r, r)\text{corr}(n, n)}$.

Since the ability of differentiating between good architectures matters more than differentiating between bad ones, criteria that emphasize more on the relative order of architectures with good performances are desired. Denoting $A_K = \{a_i | n_i < KM\}$ as the set of architectures whose estimated scores $s$ are among the top $K$ portion of the search space, we use two set of criteira [25]:

- Precision@K (P@topK) $\in (0, 1] = \frac{\#\{i | r_i < KM \wedge n_i < KM\}}{KM}$: The proportion of true top-K proportion architectures in the top-K architectures according to the scores.
- BestRanking@K (BR@K) $\in (0, 1] = \arg\min_{\alpha_i \in A_K} r_i/M$: The best normalized ranking among the top K proportion of architectures according to the scores (Lower is better).

Corresponding to P@topK, we also compare P@bottomK $= \frac{\#\{i | r_i > (1-K)M \wedge n_i > (1-K)M\}}{KM}$ to reveal how the worst architectures are distinguished. And corresponding to BR@K, we inspect WorstRanking@K (WR@K) $= \arg\max_{\alpha_i \in A_K} r_i/M$ to reveal how the supernet is likely to regard a bad architecture to be good (Lower is better). Note that the rankings and architecture numbers are all relative numbers normalized by the total architecture number $M$.

## 4 Evaluating Efficient Performance Estimators

### 4.1 Evaluation of One-shot Estimators

**Trend of Different Criteria** We inspect how these proposed criteria evolve during the training process. Unless otherwise noted, MC sample $S$=1 is used in the experiments. And all training and evaluation settings are summarized in Appendix D. Fig. 1 and Appendix Fig. A24 show the criteria trend on topological and non-topological search spaces, respectively. We can see that the convergence speeds of criteria are different, and on all search spaces except NB101, all criteria show a rising trend as the training goes on, indicating that **OSE gives better rankings with sufficient training.**

Another fact is that on all search spaces except NB101, **OSEs are better at distinguishing bad architectures (higher P@bottom5%) than distinguishing good ones (lower P@top5%).** This indicates that, although identifying the exactly optimal architecture might be difficult for OSEs, using

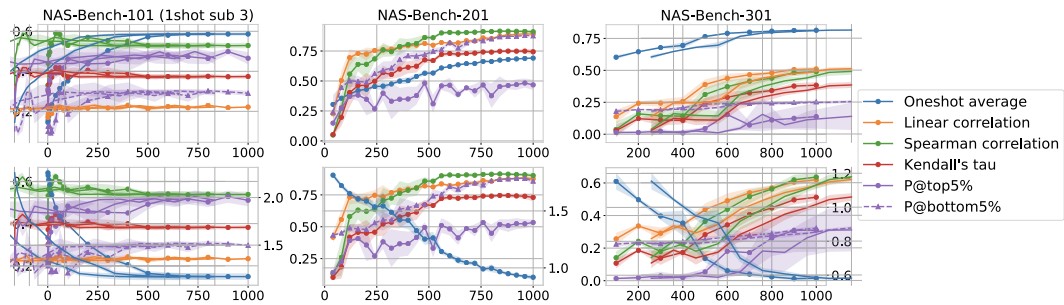

Figure 1: Top /Bottom: Criteria of using OS accuracy / OS loss as the estimations (right Y-axis: OS loss value). "Oneshot average" means the average oneshot score (accuracy or loss).

them to filter bad architectures or warm-up sample-based NAS can be very effective. The results of more criteria are shown in Appendix Fig. A1.

**In the NB301 (DARTS) space, OS loss gives significantly better estimations than OS accuracy**. For example, at epoch 1000, the KD $\tau$ of OS acc and loss are 0.381 and 0.512, respectively, while their P@top5% are 13.8% and 31.0%. This is because the loss value considers the network's output distribution rather than a single label prediction, it has a less concentrated distribution and is more informative in ranking architectures. Fig. 2 shows that the OS accuracy distribution is indeed more concentrated than the OS loss on NB301.

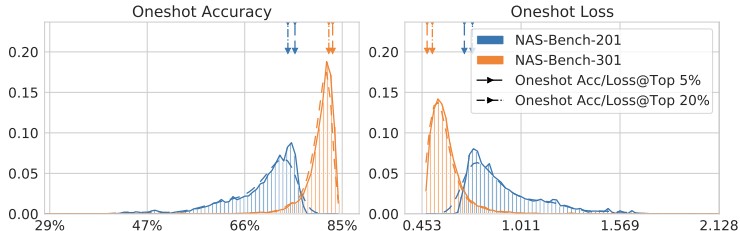

Figure 2: The distribution of OS accuracy and loss.

**Effect of the Validation Data Size** We inspect OSEs' ranking quality when using different numbers of validation data batches to evaluate the OS scores, and find that on both NB201/NB301, **using more data improves the estimation quality.** Specifically, we compute the average OS accuracies over N validation batches, where each batch contains 128 examples. And the effect of the batch number N on the ranking quality is shown in Fig. 3 and Appendix Fig. A3. Fig. 3 shows that on NB301, criteria get better when the batch number increases from 1 to 10 at epoch 1000. Interestingly, when the training is not sufficient (epoch 200), the criteria decrease with more data batches (especially those of the OS acc). To explain this, Fig. 3(upper right) shows the intra-"level" KD histogram, where architectures with the same OS accuracy using one validation batch are said to be in the same level. We can see that when the supernet is under-trained, it is not good at distinguishing between intra-level architectures (negative intra-level KDs). Therefore, using more validation data might bring negative impacts, while giving tie scores can avoid making wrong comparisons between similar architectures.

## 4.2 Evaluation of Zero-shot Estimators

Our work evaluates six parameter-level ZSEs and two architecture-level ZSEs. The six parameter-level ZSEs are *grad_norm*, *plain* [22], *snip* [14], *grasp* [36], *fisher* [34, 35], and *synflow* [33]. These ZSEs are named after sensitivity indicators initially designed for fine-grained network pruning that measure the approximate loss change when certain parameters or activations are pruned. A recent work [1] proposes to sum up parameter-wise sensitivities of all parameters to evaluate an architecture. And architecture-level ZSEs measure the architecture's discriminability by inference differences between different input images: *jacob_cov* [20] uses the input jacobian correlation, and *relu_logdet* [21] uses activation differences.

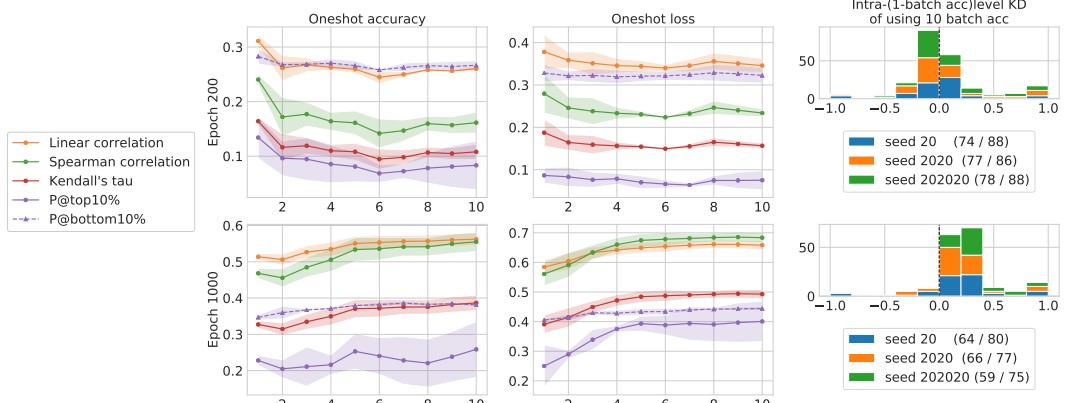

Figure 3: Criteria vary on NB301 as the batch number (X-axis) changes. Right: The histogram of intra-level KDs using 10-batch OS acc, the "levels" are partitioned according to 1-batch OS acc. Since batch_size=128, at most 128 levels can exist. The legend gives out the actual number of levels in 1-batch evaluation with format "#acc levels with #arch>1 / #total".

The full evaluation results of ZSEs are shown in Appendix B and C, and Fig. 4 shows some of the results on NB201 and NB301. We can see that **the ranking correlations of ZSEs except *relu_logdet* are even worse than the GT-Param correlation**. Also, **the relative effectiveness of ZSEs varies between search spaces**. For example, on NB301, *plain* performs better than other ZSEs except

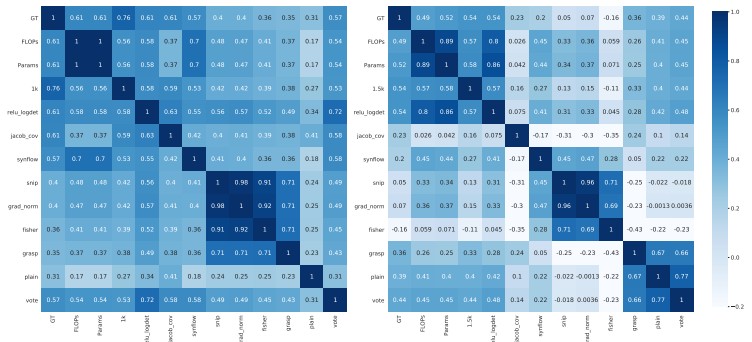

Figure 4: KD between GT, FLOPs/Params, OSEs (1k epoch) and ZSEs. Left: NB201; Right: NB301.

*relu_logdet*, while on NB201, *plain* performs worst among all ZSEs. And *jacob_cov* and *synflow* give relatively good estimations with KD of 0.61 / 0.57, but they do not perform well on NB301 (KDs are 0.23 / 0.2). Also, as shown in Appendix Tab. A12, the best-performing ZSE on topological search spaces, *relu_logdet*, does not perform well on non-topological NDS ResNet and ResNeXt-A.

The *vote* ZSE [1] conducts a majority vote between various metrics to compare each pair of architectures. We choose three best-performing ZSEs as the voting experts, and find that this simple form of voting does not bring improvements over the best constituent ZSE. Better ways of ensembling different ZSEs need to be developed.

It is a natural idea to apply ZSEs on trained networks. Thus we explore whether ZSEs can benefit from one-shot training. According to Appendix Tab. A11, current ZSEs cannot benefit from one-shot training. The ranking qualities of ZSEs except *relu_logdet* even degrade a lot after one-shot training. A possible explanation is that these ZSEs utilize the gradient information, and the gradient magnitudes in a trained supernet are too small and obscure for architecture ranking.

# 5  How & Why the Estimations Are Not Satisfying

## 5.1  Bias of One-shot Estimators

**Complexity-level Bias** To identify which architectures are under- or overestimated, we investigate the relationship of the true-estimated Ranking Difference (RD) $r_i - n_i; i = 1, \cdots, M$ and the architecture complexity (i.e. Params, FLOPs). RD serves as an indicator of overestimation for arch $i$: A positive RD indicates that this architecture is overestimated. Otherwise, it is underestimated.

Sub-architectures have different amounts of calculation and might converge with a different speed. Thus, we conduct the complexity-level bias analysis. In Fig. 5, we divide the architectures into five

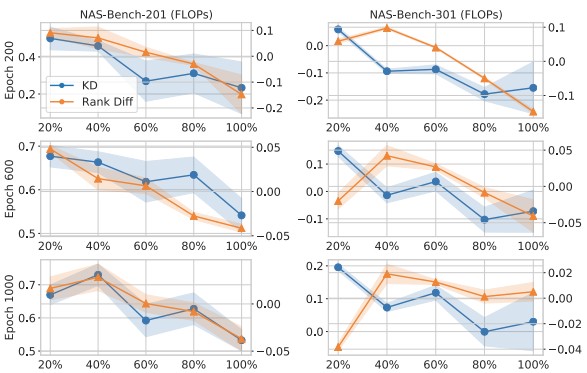

complexity groups according to the amount of calculation (FLOPs), and show the KD and average RD in each group. In the early training stages (the 1st row), the average RD shows a decreasing trend, which means that the larger the model, the easier it is to be underestimated. This is because larger models converge at a slower speed. As the training goes on (the 2nd and 3rd rows), the absolute average RD decreases, indicating that the issue of underestimating larger models gets alleviated. And on both spaces, the decreasing intra-group KD $\tau$ indicates that it is harder for OSEs to compare larger models than comparing smaller ones.

Figure 5: Complexity-level bias. Left/right Y-axis: KD $\tau$ / Average RD within the complexity group. X-axis: Complexity groups (the group with the smallest FLOPs is at the leftmost).

**Op-level Bias** We inspect the changes of GT and OS accuracy when one operation is mutated to another (edit distance=1). On NB301, we examine 23476 mutation pairs and find that the OSE estimations overes-

timate the effects brought by dilation (Dil) convolutions (Convs): All mutation types from other operations *to* DilConvs witness a higher OS increase ratio than the GT one. And the skip_connect operation is underestimated: All mutation pairs *from* skip_connect cause the OS increase ratio to be higher than the GT one. For example, when mutating one skip_connect operation to dil_conv_5x5, only 39.0% out of 2336 pairs get GT increases, while 94.9% get OS increases. This phenomenon is more remarkable when we only consider mutation pairs within the largest complexity group (grouped by Param): Only 15.3% of 569 pairs get GT increases, while 92.3% get OS increases. On NB201, based on a similar inspection of the mutation pairs, we find that OSE estimations slightly overestimate avgpool3x3 and underestimate conv3x3. Generally speaking, the op-level bias on NB201 is not as large as that on NB301. See Appendix A.2.2 for the figures and more results.

## 5.2 Bias of Zero-shot Estimators

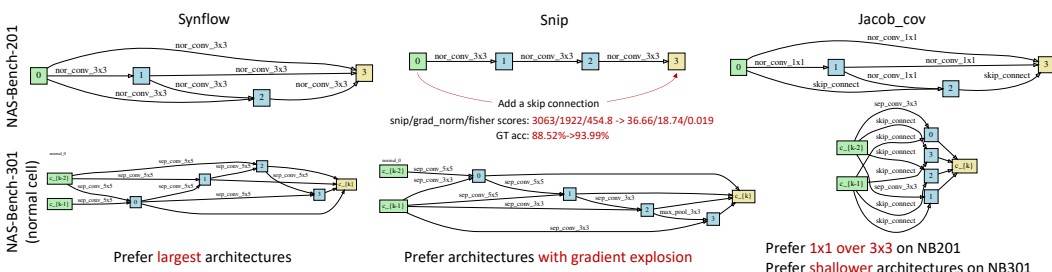

Figure 6: The best architectures ranked by several ZSEs on NB201 and NB301.

**Arch-level Bias** By inspecting the best and worst architectures indicated by ZSEs, we find that existing ZSEs have improper biases. Fig. 6 shows that *synflow* has an excessive preference for large architectures. *snip*, *grad_norm* and *fisher* give similar rankings of architectures (see Fig. 4), and show improper preferences for architectures with gradient explosion: On NB201, they show a clear preference for architectures without skip connections, which are far from optimal. This is because gradient magnitudes in these architectures get exploded, and the absolute parameter-wise sensitivity is high. In a word, the parameter-level ZSEs adapted from the fine-grained pruning literature are not very suitable for ranking architectures, since they are designed to reflect the relative *parameter-wise* sensitivity. And due to their sensitivity to scales and gradient explosion, a simple form of adding up the parameter-wise sensitivity provides improperly biased estimations for architecture performances.

In contrast, architecture-level ZSEs (*jacob_cov*, *relu_logdet*) are more reasonable attempts that measure the architectures' discriminability by inference differences between input images. Nevertheless, as shown in Fig. 6 and Appendix B.2, these two ZSEs prefer architectures with smaller receptive fields (prefer smaller kernel sizes or shallow architectures). Consequently, although these two ZSEs have relatively good ranking correlations on topological search spaces, they have difficulties in picking out top architectures (Poor P@topKs, see Appendix Tab. A9).

### 5.3 Variance of One-shot Estimators

**Accuracy Forgetting** Due to the parameter sharing and the random sample training scheme, the training of subsequent architectures overwrites the weights of previous ones, thus degrades their OS accuracy. This "multi-model forgetting" phenomenon [4, 46] accounts for the variance of OS accuracies. Appendix Fig. A14 verifies the existence of the forgetting phenomenon. For each architecture in one epoch, we define its forgetting value (FV) as $acc_2 - acc_1$, where $acc_1$ refers to its valid accuracy right after its training, and $acc_2$ refers to its accuracy after all the architectures in this epoch have been trained. Appendix Fig. A14 shows that the forgetting phenomenon exists in the early training stages, where the FVs are negative. As training progresses, the variance of the FVs decreases, which is natural due to the learning rate decay. Also, the mean FV becomes positive, indicating that training other architectures can have positive transferring effects on previous architectures instead of negative ones (i.e. forgetting). This observation can be explained by the increasing trend of inter-architecture gradient similarity in Appendix Fig. A15.

**Ranking Stability** We demonstrate the ranking stability in Fig. 7, since it plays an important role that influences the NAS process more directly than the accuracy stability. The criteria in this figure (i.e. relative KD, relative P@top/bottomK) are calculated with two sets of adjacent OS estimations, while the estimations of the latter checkpoint are taken as the GT one. We can see that the ranking stability increases with sufficient training and the OS rankings of bad architectures are relatively stable (relP@bottomK). On NB301, even with rather sufficient training

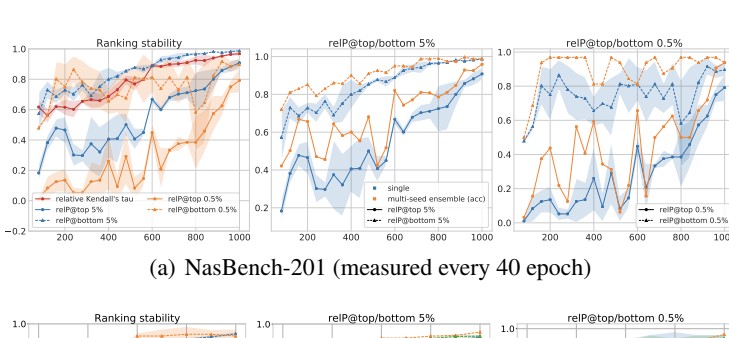

(a) NasBench-201 (measured every 40 epoch)

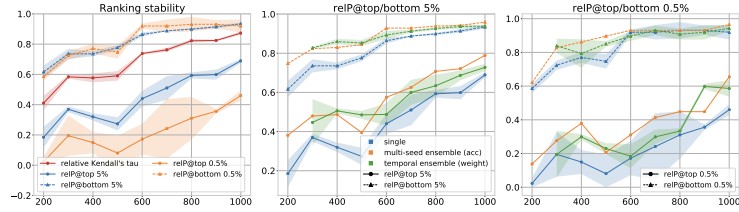

(b) NasBench-301 (measured every 100 epoch)

Figure 7: Ranking stability of OSEs.

(1k epoch) where the mean OS accuracy already saturates (Fig. 1), the ranking stability of top architectures is still not high (relP@top 0.5%∼0.46). This is reasonable since that the accuracy differences between architectures in the DARTS space are smaller. And as expected, averaging the OS accuracy of multiple supernets stabilizes OSE estimations. Also, the temporal weight ensemble of multiple checkpoints can stabilize the estimations (Sec. 6.1).

## 6 How to Improve One-shot Estimations

Since different architectures require different values for supernet parameters, as the side effect of acceleration, parameter sharing serves as the intrinsic reason for the OSE correlation gap. Appendix Fig. A15 shows the gradient similarity distribution between architecture pairs on NB201. We can see that the inter-architecture gradient similarities vary in a large range, and one common phenomenon on NB201 and NB301 is that the mean similarity between architecture pairs is lower in the middle-stage layers and the architectures' gradients in the very first and last layers are more similar. Another

slightly counterintuitive fact is that the gradient directions become more similar as the training goes on, especially on NB201. This can explain the positive transferring effect in the latter training stages.

Due to parameter sharing, the random sample training scheme of OSE causes estimation variances. On the other hand, improper sampling distribution causes estimation biases. There are two types of reasons for the bias: 1) Some architectures (e.g. with larger complexity) might need higher sampling probability to match their relative performance in standalone training. 2) Architectures are sampled from an unfair distribution, i.e., some architectures have undesirable higher equivalent probabilities.

Echoing the above analysis, this section conducts case studies to improve the OSE estimations from 3 perspectives, i.e. reducing the variance, bias, and parameter sharing extent. Sec. 6.1 experiments with 2 techniques that can reduce the OS estimation variance. And in Sec. 6.2, we demonstrate that using de-isomorphic sampling in space with isomorphic architectures (NB201) helps improve the sampling fairness, thus reduce the estimation bias.

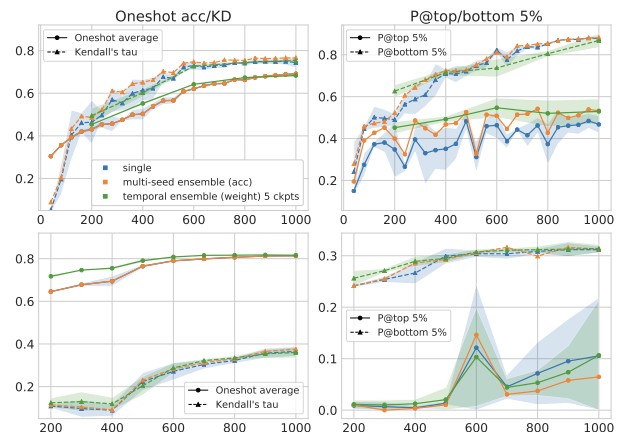

Figure 8: Effect of ensemble techniques on OSEs. Top: NB201; Bottom: NB301.

## 6.1 Variance Reduction

**Temporal Variance Reduction** Sec. 5.3 shows that averaging OS scores of several supernets stabilizes the estimations. However, this technique is not practical due to its linearly enlarged consumption, as training k supernets takes k-times more computation. As a remedy, Guo et al. [12] propose to only train one supernet, and stabilize OS estimations by temporally averaging weights of supernet checkpoints. Besides the variance reduction effect shown in Fig. 7, Fig. 8 shows whether ensembling techniques can bring other ranking quality improvements. We can see that temporally ensembling 3 or 5 checkpoints brings improvements on NB201 but brings no bias improvements on NB301.

**Sampling Variance Reduction** We compare the results of using different MC sample numbers $S$ in supernet training. We also adapt Fair-NAS [7] sampling strategy to NB201 and NB301. Using multiple MC architecture samples has different influences in different spaces: It is beneficial for the estimation quality on NB301, while the estimation quality on NB201 decreases slightly as the MC sample number increases. See Appendix A.3.2 for more detailed results.

## 6.2 Sampling Fairness Improvement

Besides the complexity-level and op-level biases shown in Sec. 5.1, OSEs also have some evident architecture-level biases. The NB201 search space contains many isomorphic architectures with different representations, and there are 6466 unique structures (out of 15625) after de-isomorphism. We find that even after sufficient training, the supernet still overestimates some simple architectures significantly. Fig. 9(left) shows the top-2 ranked architectures by the average of 3 supernet's OS scores at epoch 1000. With vanilla sampling (Iso), OS estimations bias towards simple architectures (a single Conv) with many isomorphic counterparts (Iso group size=31). We find that this is because isomorphic architectures have identical gradients w.r.t. shared parameters, so that the shared parameters tend to be optimized towards the gradient directions of architectures with many isomorphic counterparts.

We compare the results of sampling w. or w.o. isomorphic architectures in Fig. 9(right). We can see that **using the de-isomorphism (deiso) sampling strategy helps pick out top architectures and brings significant improvements on BR@0.5% and P@5%** (1.9% to 0.23%, 21.3% to 46.7%). We also experiment with a post-de-isomorphism (post-deiso) technique, in which the estimations of architectures in an isomorphic group are averaged during testing, while no changes are made during training. We can see that "post-deiso" brings slight improvements on BR@Ks and P@topKs

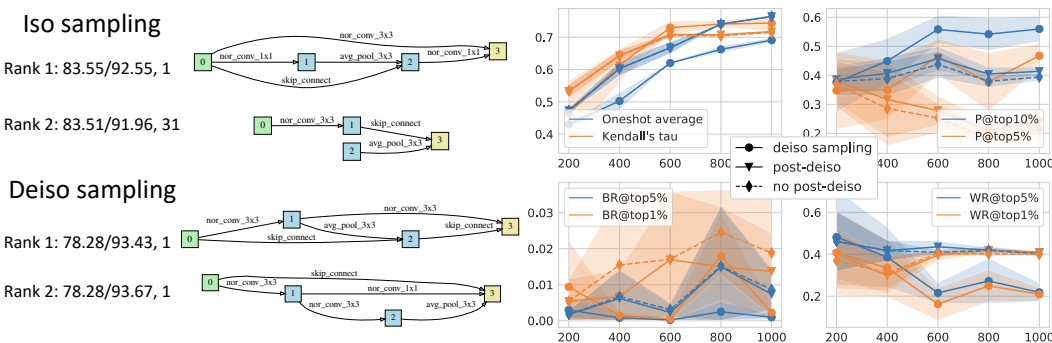

Figure 9: Comparison of Iso / Deiso sampling strategy. Left: Top-2 ranked architectures when the supernet is trained with Iso / Deiso sampling strategy, the legend's format is "OS acc (%)/GT acc (%), Iso group size". Right: Criteria comparison along the training process.

compared with "no post-deiso", which might owe to the decreased estimation variances. Actually, the deiso sampling strategy is to find a de-isomorphic representation space and conduct uniform sampling in it, and our study provides another evidence for the statement made by [37] that representations can be critical for NAS methods. More detailed results and discussions are in Appendix A.3.3.

### 6.3 Sharing Extent Reduction

**Operation Pruning** We remove one or two operations in the search space (SS) and conduct supernet training on the resulting sub-SS. After training the supernet, we compare the OS estimations on the sub-SS provided by the supernet trained on full SS and the sub-SS. The detailed results and analyses can be found in Appendix A.3.4. And the conclusion is: Sharing extent reduction by removing operations can bring improvements to the average OS scores of the remaining architectures in the sub-SS, especially in the early training stages. However, **whether the improved absolute OS scores can bring ranking quality improvements is questionable, and the results vary across SSes**.

**One-shot Pruning** We conduct SS pruning on NB201 by selecting the top 10%, 25%, 50% architectures ranked by the OS scores of supernet (epoch 600), and continue to finetune the supernet to 1000 epoch with these architectures. The good news is that on NB201, OS pruning brings improvements on both the average OS score and ranking quality in the sub-SS: 2.2%/1.3%/0.1% average OS score increases and 0.189/0.046/0.086 KD increases when the sub-SS contains 10%/25%/50% architectures, respectively. The results reveal **the potential of dynamic SS pruning for improving the OSE quality, especially for good architectures**. However, this per-architecture hard pruning scheme is not practical since it needs an exhaustive test of the full search space. To explore practical dynamic SS pruning methods, we conduct a case study on per-architecture soft pruning with a jointly-trained controller, where the controller gives higher sampling probability to the architectures with higher OS scores. The results and analyses are shown in Appendix A.3.4.

**Remove the Affine Operation in BN** We compare using or not using BN affine operations, and give out the comparison results in Appendix A.3.4. And the suggestion is that **one should not use BN affine operations in the search process**.

## 7 Conclusion

We present an analysis framework of efficient architecture performance estimators in NAS, containing carefully developed criteria and organized analyses. Within the framework, we conduct an in-depth analysis of OSEs and ZSEs on five benchmarking search spaces with distinct properties. Our work reveals the properties, weaknesses (variance and bias) of current architecture performance estimators. For OSEs, we further conclude three directions for their improvements and experiment with several mitigations accordingly. Our work gives out suggestions for future NAS applications and points out research directions to further improve current OSEs and ZSEs. Besides the take-away knowledge, our work also provides strong baselines for future research of efficient performance estimators, and the analysis framework could be utilized to diagnose new performance estimators.

## Acknowledgements

This work was supported by National Natural Science Foundation of China (No. U19B2019, 61832007), Tsinghua EE Xilinx AI Research Fund, Beijing National Research Center for Information Science and Technology (BNRist), and Beijing Innovation Center for Future Chips. We thank Zinan Lin, Tianchen Zhao, and Hanbo Sun for their valuable discussions. Finally, we thank all anonymous reviewers for their constructive suggestions.

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
