# Appendix for Evaluating Efficient Performance Estimators of Neural Architectures

**Xuefei Ning**[1♯]    **Changcheng Tang**[2]    **Wenshuo Li**[1]    **Zixuan Zhou**[1]

**Shuang Liang**[2]    **Huazhong Yang**[1†]    **Yu Wang**[1‡]

Department of Electronic Engineering, Tsinghua University[1]
Novauto Technology Co. Ltd.[2]
♯foxdoraame@gmail.com, †yanghz@tsinghua.edu.cn, ‡yu-wang@tsinghua.edu.cn

Tab. A1 gives out the basic information and main distinct characteristics of the search spaces.

Table A1: Properties of the five benchmarking search spaces.

| Search Space | Search Space Size | Op on Edge or Node | #Nodes, #Edges | #Shared Positions | #Operation Types | Main Characteristics |
|---|---|---|---|---|---|---|
| NB101 (NB1shot-3) | 14.6k (w.o. isomorphic ones) | Node | 7, 9 | 5 | 3 | Operation on node (larger sharing extent) |
| NB201 | 6.5k / 15.6k (w.o. / w. isomorphic ones) | Edge | 4, 6 | 6 | 5 | Large proportion of isomorphic architectures |
| NB301 | $10^{18}$ (w. isomorphic ones) | Edge | 6, 8 | 14 | 7 | 1. Large search space size 2. The GT accuracy difference between architectures is small |

| Search Space | Search Space Size | #Choices of Architectural Decisions | | | | Main Characteristics |
|---|---|---|---|---|---|---|
| | | Depth | Width | Ratio | Group Number | |
| NDS ResNet | 1260k | 9 | 12 | - | - | Non-topological |
| NDS ResNeXt | 11391k | 5 | 5 | 3 | 3 | 1. Non-topological 2. Contain convolution group number in the search space |

## A  Discussions and Results about One-shot Estimators

### A.1  Evaluation of One-shot Estimators

#### A.1.1  Trend of P@top/bottom K & BR/WR@K

Fig. A1 shows the P@top/bottom K, B/WR@K for multiple Ks on NB101-1shot, NB201, and NB301. On NB101-1shot, the performances of OSEs converge quickly, and OSEs are capable of distinguishing good architecture relatively well (P@top5% $\approx 0.49$). On NB201, OSEs can distinguish bad architectures very well (P@bottom 5% $\approx 0.8772$), while relatively speaking, their ability in distinguishing good architectures is weaker (P@top 5% $\approx 0.4675$). On the harder NB301 search space, the P@top Ks are not that high as those on NB201, and the P@bottomKs are still worse than P@topKs.

The second column in the figure shows the best GT ranking (Best/WorstR@Ks) & accuracy (Best/WorstAcc@Ks) in the top-K-proportion of OSE ranked architectures, and the third column in the figure shows the worst GT ranking (Best/WorstR@Ks) & accuracy (Best/WorstAcc@Ks). On NB201, BR@5% converges very fast to 0, which means only after tens of epochs of training, one can find the best architecture in the top 5% of the OSE-ranked architectures. BR/Acc@0.1%, the best GT ranking and accuracy of the top-6 ($6466 \times 0.1\%$) OSE ranked architectures, have a large variance across temporal epochs and multiple supernets. In a NAS flow where one takes out several top-ranked

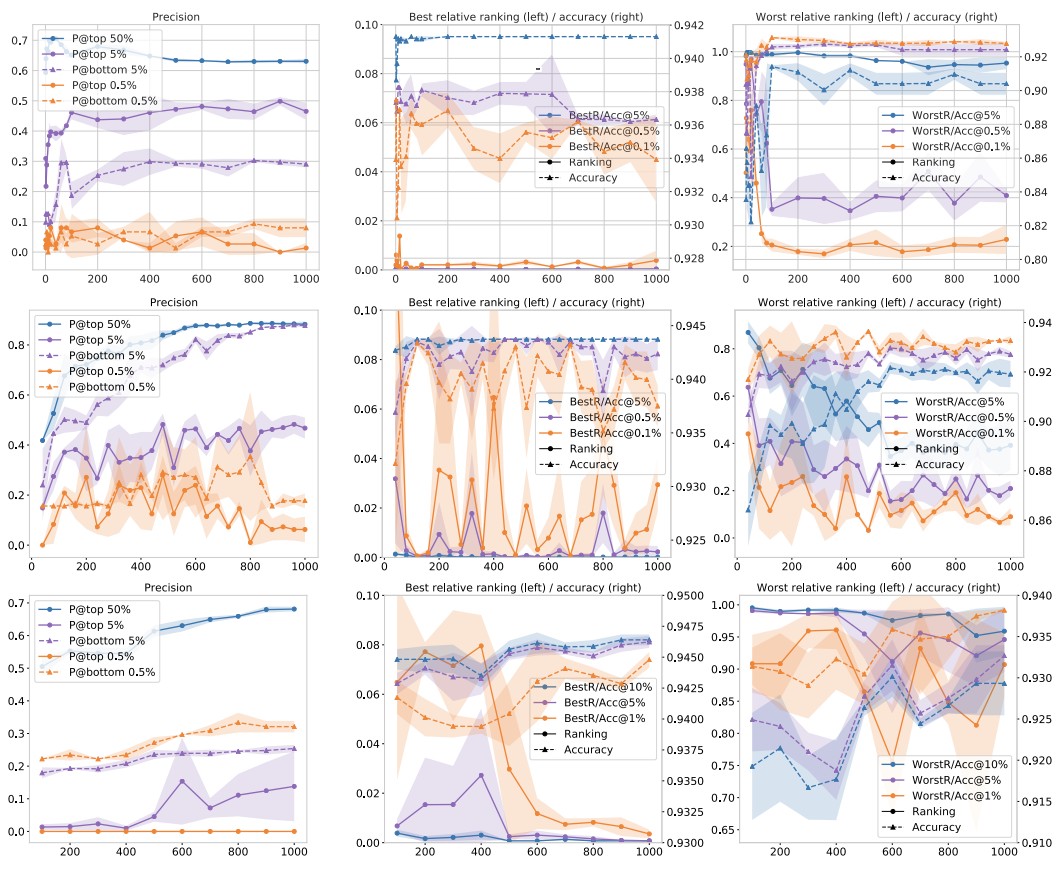

Figure A1: Trend of P@topK, P@bottomK, BR@K, WR@K. Top / Middle / Bottom: NB101-1shot / NB201 / NB301.

architectures and conducts final training, the stability of BR/Acc@K is of concern. Although the BR/Acc@Ks criteria converge very fast, the WR/Acc@Ks become better and better as the training progresses. This indicates that *on NB201, OSE is mainly learning to reduce its chance of regarding bad architectures as good ones in the middle and late training stages*.

On NB301, although WR@5% gets better (the average WR@5% across the 3-supernet decreases from 0.9909 at 100 epoch to 0.9460 at 1000 epoch), it is still very high (lower is better). Nevertheless, the NB201 and NB301 search spaces have distinct properties. Since the GT accuracy distribution of NB301 is more concentrated, and the architectures are more similar, the worst accuracy of top-5% OSE-ranked architectures (WorstAcc@5%) at epoch 1000 (0.9327) is actually better than that on NB201 (0.9191). Even the WorstAcc@5% at epoch 100 (0.9249) on NB301 is better than that on NB201 at epoch 1000. *Therefore, although the ranking quality criteria on NB301 are worse than those on NB201, OSEs can still help find architectures with satisfying accuracy on the harder and better NB301 search space. When analyzing the ranking quality, we still need to consider the absolute accuracy distribution.*

In our experiments, we observe some differences in the behaviors of OSEs across search spaces:

- The phenomena on NB101 are different from those on NB201 and NB301: 1) P@top 5% is higher than P@bottom 5%, and 2) longer training after the first 20 epochs does not improve the ranking quality. These two distinct phenomena might arise from two aspects, respectively: 1) As shown in Fig. A2, the GT distribution of the top GT accuracies on NB101 is less concentrated than that on NB201 and NB301, and thus it is easier to distinguish the top architectures on NB101. 2) Different from NB201 and NB301, NB101 is an operation-on-node search space. The sharing extent of supernet on operation-on-node

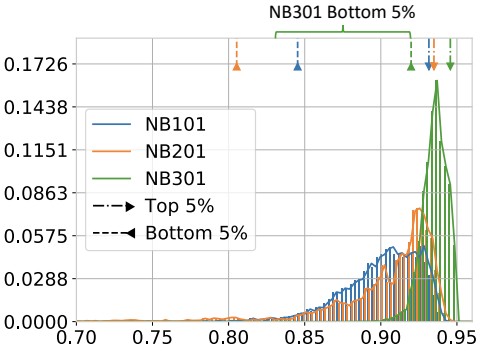

Figure A2: GT accuracy distribution on NB101, NB201 and NB301.

search spaces is larger than that on operation-on-edge search spaces, since for each node, no matter which input connections are chosen, the same parameters are used. A larger sharing extent would limit the potential ranking quality of OSEs, and thus longer training might not bring additional improvements in the ranking quality.

- Using OS loss as the estimated score achieves much better ranking quality than OS accuracy on NB301, but is not better on NB101 and NB201. We analyze that this is because the GT accuracy distribution on NB301 is concentrated, as shown in Fig. A2. In other words, many architectures have similar GT accuracies. Consequently, OS accuracies of these architectures are also close (as shown in Fig. 2 in the main paper), while the OS loss values provide more information on prediction confidences for better architecture ranking.

### A.1.2 Effect of Validation Data Size

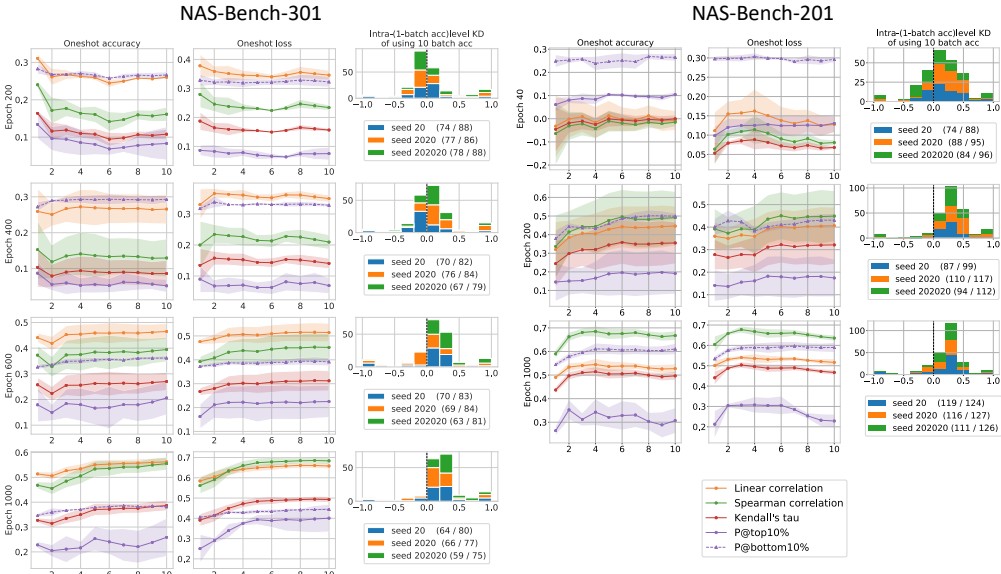

Figure A3: Criteria vary as the batch number (X axis) changes (batch size=128).

We evaluate the influences of the validation data batch number (batch size=128) on the OSE ranking quality, and the full results on NB301 and NB201 are shown in Fig. A3. We can see that *with sufficient training, more validation data helps increase the ranking quality*. More specifically, on NB301, the estimation quality increases as the number of validation data batches increases. And on NB201, at Epoch 1000, although the trend is not monotonically increasing when the batch number increases from 1 to 10, the estimation quality using few validation batches is a lot worst than using the full validation data (KD≈0.5 V.S. KD≈0.76).

On NB301, we observe another interesting phenomenon: In the early training stage of the supernet, the ranking quality of using OS accuracy shows an obvious decreasing trend as the batch number increases from 1 to 10. As we have analyzed in the main paper (Sec. 4.1), this is because using only one validation data batch results in fewer levels (smaller resolution) of OS accuracy and gives many ties. And *when the supernet is under-trained, its ability in distinguishing the intra-level architectures is weak, thus using more data might bring negative effects*. The intra-level KD histogram shown next to the ranking quality plots verifies our speculation. At Epoch 200, negative intra-level KDs occur more often than positive ones. As the training progresses, the distribution of intra-level KDs moves towards positive, indicating that the supernet is becoming better at distinguishing similar architectures.

### A.1.3 Other Datasets

Besides CIFAR-10, NB201 provides GT accuracies on other two datasets: CIFAR-100, ImageNet-16-120 [4]. We run OS training on these two datasets, and calculate the KD and P@top 5% between the OS and GT accuracies across multiple datasets. The results are shown in Fig. A4.

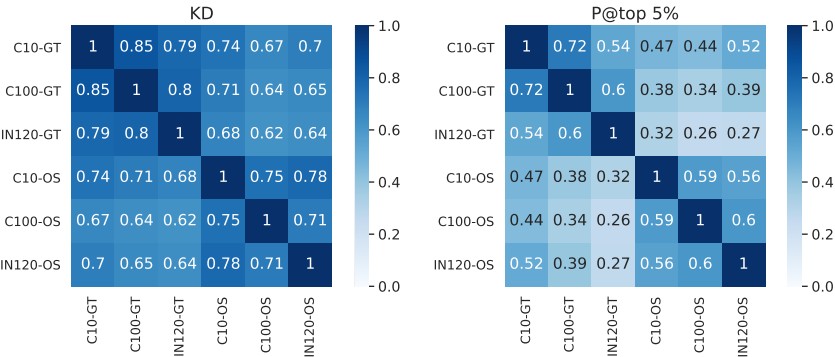

Figure A4: KD and P@top 5% across different datasets on NB201. "C10" refers to CIFAR-10, "C100" refers to CIFAR-100, "IN120" refers to ImageNet-16-120.

A counter-intuitive observation from Fig. A4 (Left) is that no matter of which dataset the GT accuracy we use, the KD of using OS accuracy on CIFAR-10 is the highest. We speculate that since the number of classes on CIFAR-100 and ImageNet-16-120 is larger than that on CIFAR-10 (100 V.S. 10), the classification head might become a parameter-sharing bottleneck. For example, when the init channel number is 16, the classification head on these two datasets is constructed by a global average pooling layer and a single linear layer that convert $16 \times 2^2 = 64$ units to 100 units. This compact FC layer might lack the representational capability to be shared by lots of architectures, which can cause all architectures to be too under-trained to reflect their standalone rankings correctly. Motivated by this speculation, we conduct a simple experiment to train supernets with enlarged channel numbers (32 or 64), and show the average validation accuracy, KD, and P@top 5% of the supernets in Fig. A5. We can see that, intuitively, as the init channel number increases, the average OS validation accuracy increases. On CIFAR-10, although the absolute OS accuracy increases, the ranking quality degrades, which is reasonable as using different init channel numbers induces additional search-final gaps. In contrast, on CIFAR-100 and ImageNet-16-120, the KD increases as the init channel number increases, and this might be due to the alleviated representational bottleneck in the classification head. However, the P@top 5% cannot benefit from an increasing supernet channel number on these two datasets.

### A.1.4 Influences of Proxy Model

Due to memory and time constraints, it is common to use a shallower or thinner proxy model in the search process. The common practice is to search using small proxy models with fewer channels and layers, and then augment the discovered architecture to a larger one for final training. We conduct a small experiment to inspect the correlation gaps brought by using proxy models on NB201 for the CIFAR-10 dataset. From the results shown in Fig. A6(a)(b), we can see that *channel proxy has little influence while layer proxy reduces the reliability of search results*. Thus, for cell-based search spaces, proxy-less search w.r.t the layer number is worth studying [2, 3].

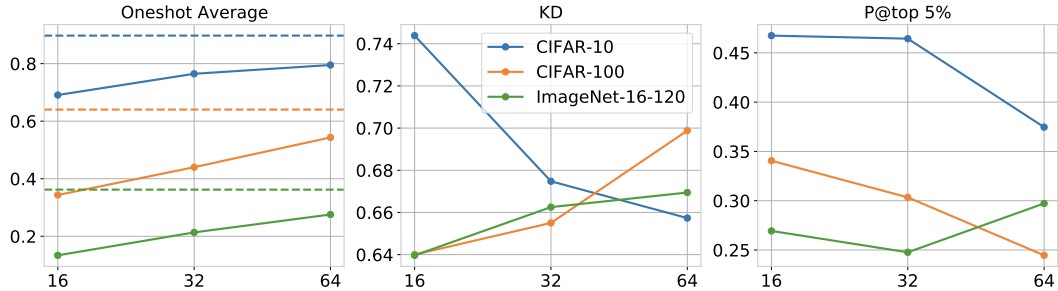

Figure A5: Oneshot Average, KD and P@top 5% when using larger init channels on NB201. X axis: Init channel number. The horizontal dashed lines in the leftmost figure mark the average GT validation accuracy.

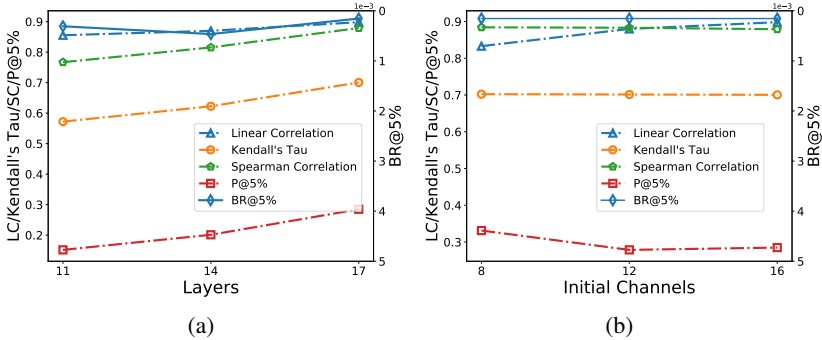

(a)                                          (b)

Figure A6: (a) Influence of layer proxy. (b) Influence of channel proxy.

## A.2   Bias of One-shot Estimators

### A.2.1   Complexity-level Bias

Fig. A7 shows the complexity-level bias on NB101 and NB301, where the five complexity groups are grouped by the parameter size. Note that as the parameter rankings and FLOPs rankings on NB101 and NB201 are identical, thus we only plot one of the RD & KD plots on these two search spaces (the plot of NB201 is in Fig. 5 in the main paper). On NB301, as the training progresses, the underestimation phenomenon of larger architectures gradually vanishes, and the intra-group KD is consistently increasing. While on NB101-1shot, the training after 200 epochs can neither alleviate the bias on NB101, nor improve the intra-group KD.

Apart from the summary statistics (average RD & KD $\tau$) of each complexity group in the above analysis, we also show the scatter plot of GT/OS accuracies and the parameter sizes in Fig. A8 (NB201) and in Fig. A9 (NB301). From the left subplot in the two figures, again, we can witness that *in early training stages, OSEs underestimate large architectures since their training is not sufficient. As the training goes on, the parameter sizes of top-ranked architectures by the OSEs become larger*.

We visualize the Pareto frontiers discovered by OSEs in the right subplots of the two figures. The blue lines with square markers show the one-shot scores of the GT Pareto frontier, while the orange/green/red lines show the GT scores of the OS Pareto frontier. On NB301, the orange line on the blue (GT) scatter show that the OS Pareto frontier architectures with smaller parameter sizes $(0.6\text{-}1.2\times10^6)$ have a larger absolute accuracy difference with the architectures on the GT Pareto frontier. However, in the range with the smaller parameter size, there are fewer architecture points between the blue line (GT Pareto architectures) and the black line (OS Pareto architectures) on the orange scatter (OS 1000 epoch). This means that *by taking multiple levels of the OS Pareto frontier, the large GT accuracy differences of the architectures with small parameter sizes on the OS Pareto frontier can be mitigated*. To visualize this observation more clearly, we plot the absolute GT accuracy difference between the GT best and the OS Pareto best w.r.t. the number of Pareto levels in Fig. A10.

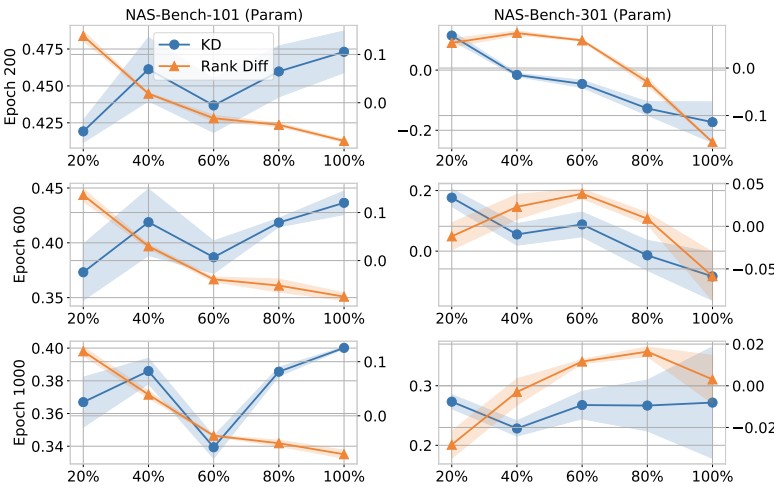

Figure A7: Complexity-level bias (grouped by Param) of one-shot estimators on NB101 and NB301. Y axis left/right: KD $\tau$ / Average RD within the complexity group.

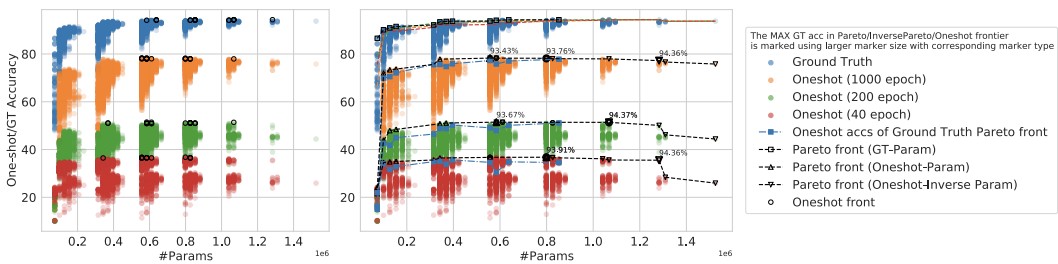

Figure A8: The oneshot front, oneshot-param pareto front, oneshot-inverse param front on NB201. Left: Black circles in each scatter group mark the architectures with the best accuracy. Right: Pareto curves on the scatter plot.

Indeed, in both the OSEs at 200 epoch and 1000 epoch, taking multiple levels of Pareto frontier is effective in shrinking the absolute differences between GT Pareto best accuracy and OS Pareto best accuracy. When the supernet is sufficiently trained (1000 epoch), the GT accuracy differences in Param group 1 can be reduced from $\sim 0.6$ to $\sim 0.2$.

### A.2.2 Operation-level Bias

To study the operation-level bias of OSEs, we inspect the changes of GT accuracy, OS accuracy, and OS loss when one operation is mutated to another operation. Fig. A11(a) shows the histogram of GT accuracy and OS score changes of mutation pairs (edit distance=1) on NB301, and each legend gives out the ratio of mutation pairs (#Mutation pairs with accuracy increase/#All mutation pairs) that get GT/OS accuracy increases or OS loss decreases. We can see that *on NB301, the OSE overestimates the effects brought by dilation (Dil) convolutions (Convs)*, i.e., dil_conv_3x3/5x5: All mutation types from other operations *to* DilConvs witness a higher OS increase ratio than the GT one. And the skip_connect operation is underestimated: All mutation pairs *from* skip_connect cause the OS increase ratio to be higher than the GT one. For example, when mutating one skip_connect operation to dil_conv_5x5, only 39.0% out of 2336 pairs get GT increases, while 94.9% get OS accuracy increases and 90.0% get OS loss decreases.

To take the complexity-level bias and the op-level bias into consideration in the meantime, we show the histogram of GT and OS accuracy changes in the largest complexity group (out of five groups in total) in Fig. A11(b). *We can see that the over- and under-estimation phenomenon of DilConvs and skip_connect are even more remarkable within the largest complexity group* (grouped by Param): For

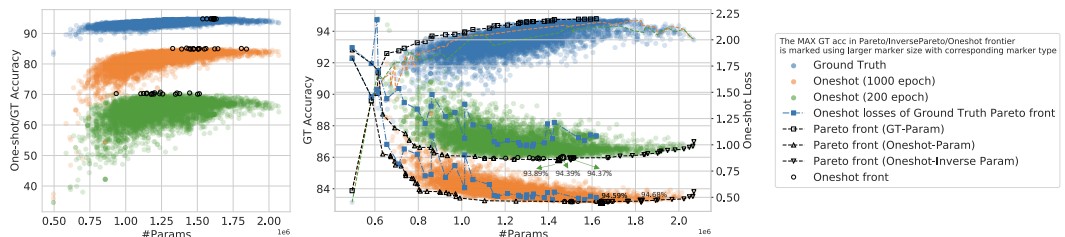

Figure A9: The oneshot front, oneshot-param pareto front, oneshot-inverse param front on NB301. Left: Black circles in each scatter group mark the architectures with the best accuracy. Right: Pareto curves on the scatter plot.

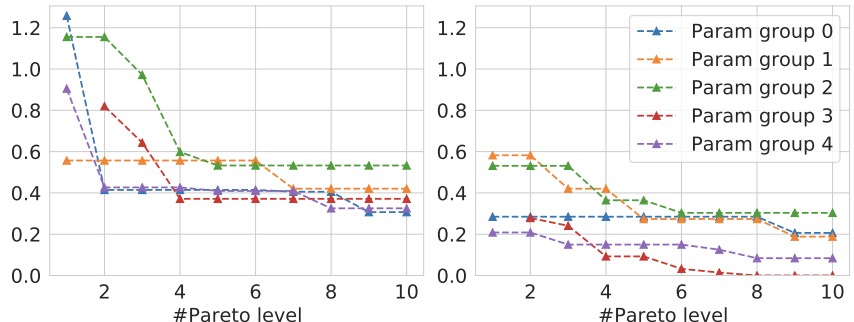

Figure A10: Absolute GT accuracy difference (%) between the GT best and the OS pareto best w.r.t the number of pareto levels (NB301). Left: 200 epoch; Right: 1000 epoch. Different lines: Five architecture groups ordered by Param (Architectures in Param group 0 has the smallest #Param).

example, when mutating one skip_connect operation to dil_conv_5x5, only 15.3% of 569 pairs get GT increases, while 92.3% get OS accuracy increases and 82.4% get OS loss decreases. Fig. A12 shows the GT/OS increase ratios in the five complexity groups. We can see that changing a skip_connect to a DilConv on a large architecture brings negative impacts in most cases (GT increase ratio < 0.5). In the largest complexity group (group 4), the mutation pairs that change a skip_connect to any of the four parametrized operations witness a GT increase ratio < 0.5. However, the OS accuracy still increases in most cases. Actually, when changing one non-parametrized operation to another parametrized operation, the increasing ratio of OS accuracy is always larger than 0.75.

On NB201, based on a similar inspection of the mutation pairs in Fig. A13, we find that *OSE estimations slightly overestimate avgpool3x3 and underestimate conv3x3*. As the mutation pairs to avgpool3x3 have a larger chance of getting OS increases than GT increases, and the mutation pairs to conv3x3 have a smaller chance of OS increase than GT increase. Generally speaking, the histograms of GT and OS changes are similar in all mutation types, which means that the op-level bias on NB201 is not that obvious as on NB301.

## A.3 Mitigations

### A.3.1 Gradient Visualization

Since different architectures require different values for supernet parameters, as the side effect of acceleration, parameter sharing serves as the intrinsic reason for the OSE correlation gap. Fig. A15 shows the distribution of the gradient similarity between architecture pairs on NB201 and NB301. We can see that the gradient similarity between different architecture pairs varies from -0.75 to 1.0, and one common phenomenon is that *the mean similarity between architecture pairs is lower in the middle-stage layers and the architectures' gradients in the very first and last layers are more similar*. For example, the 15 normal cells on NB201 ordered from the smallest mean gradient similarity to the largest are S2C8, S2C7, S2C9, S2C6, S2C5, S1C3, S1C4, S1C2, S310, S1C1, S1C0, S3C11, S3C12, S3C13, S3C14, where "S" denotes "stage" numbered from 1 to 3 (3 stages in total), and "C"

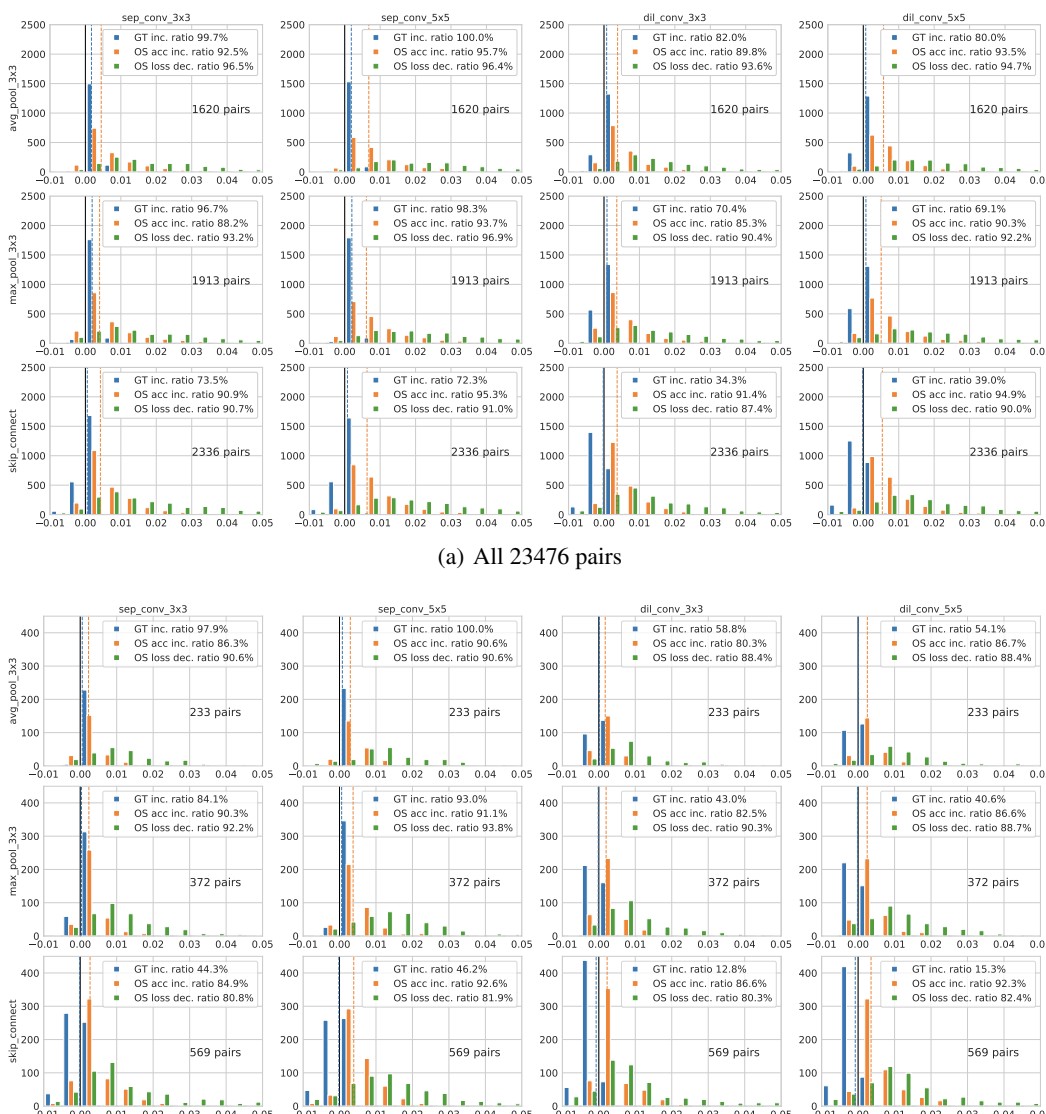

(a) All 23476 pairs

(b) 4696 mutation pairs within the architecture group with the largest complexity (grouped by Param). There are five groups in total, and each group has 4695/4696 ($\approx$23476/5) architectures

Figure A11: (NAS-Bench-301) The histogram of GT accuracy increases, OS accuracy increases and OS loss decreases when one non-parametrized operation is mutated to another parametrized operation. Rows: Mutation from avg_pool_3x3, max_pool_3x3, and skip_connect. Columns: Mutation to sep_conv_3x3, sep_conv_5x5, dil_conv_3x3 and dil_conv_5x5. 23476 mutation pairs in total are examined, and each pair has an edit distance=1.

denotes "cell" numbered from 0 to 14 (15 normal cells in total). And the cells before the second downsampling, S2C7-S2C9, have the lowest mean similarity. Another slightly counter-intuitive fact is that *the gradient directions become more similar as the training goes on, especially on NB201.*

### A.3.2 Variance Reduction

Tab. A2 compares the results of using different MC sample numbers $S$ in supernet training. We adapt the Fair-NAS [5] sampling strategy to the NB201/301 spaces (a special case of MC sample 5 and 7 for NB201 and NB301, respectively), and show the pseudocode in Alg. 1. Tab. A2 shows that *using multiple MC samples have different influences on different spaces. Using multiple MC*

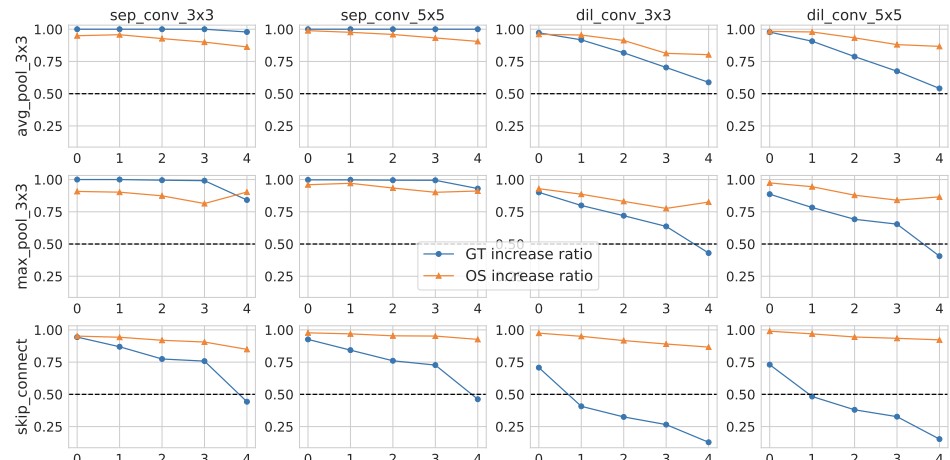

Figure A12: (NAS-Bench-301) The ratio of mutation pairs (#Mutation pairs with accuracy increase/#All mutation pairs) that get GT/OS accuracy increases. X axis: Complexity groups grouped by Param (the architecture group with the fewest parameters is 0, and the architecture group with the most parameters is 4).

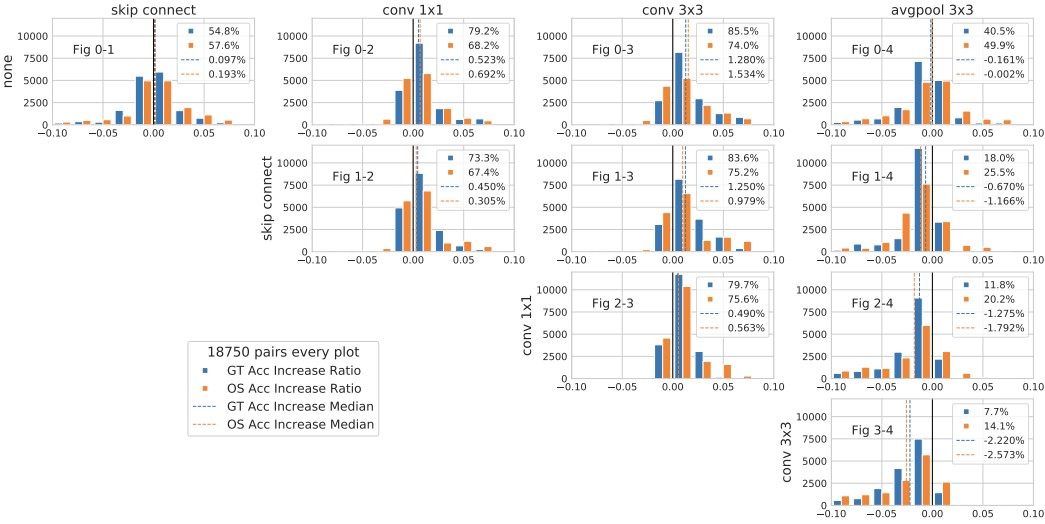

Figure A13: (NAS-Bench-201) The histogram of GT and OS accuracy changes when one operation is mutated to another operation. For each of the $\text{NUM\_OP} \times (\text{NUM\_OP} - 1)/2 = 5 \times 4/2 = 10$ mutation types, all the $\text{NUM\_EDGE} \times \text{NUM\_OP}^{\text{NUM\_EDGE}-1} = 6 \times 5^{6-1} = 18750$ pairs are examined, and each pair has an edit distance=1.

*samples on NB101/NB301 brings slight KD improvements, while the estimation quality on NB201 decreases slightly as the MC sample number increases.* Note that in Tab. A2, the training epochs of all experiments are set so that all training experiments have similar run time to 1000-epoch training with $S$=1.

The performances when the model converges are shown in Tab. A3 (the learning rate is decayed to less than 1e-5). On NB101, the training epochs for $S$=1/3/5/7 are 1000/350/340/380. On NB201, the training epochs for $S$=1/3/5 and FairNAS are all 1000 epochs. On NB301, the training epochs for $S$=1/3/5 and FairNAS are 3000/1100/600/600, respectively.

On NB201, we witness that when the training is sufficient, the KD of OS estimations slightly degrades when $S$>1. We try using the OS losses as the scores, and the phenomenon of degrading KDs still

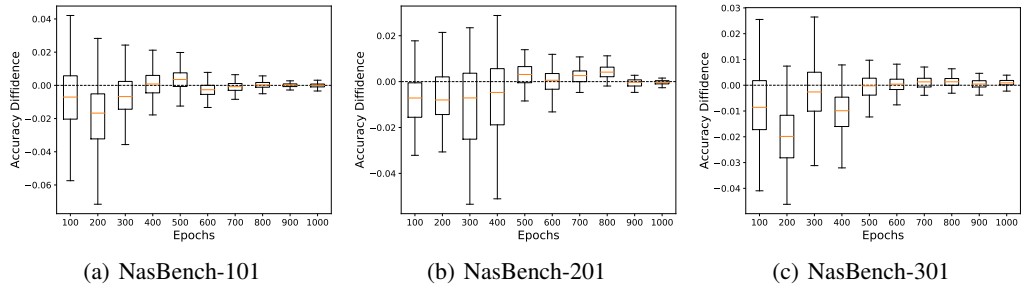

(a) NasBench-101    (b) NasBench-201    (c) NasBench-301

Figure A14: Multi-model forgetting phenomenon.

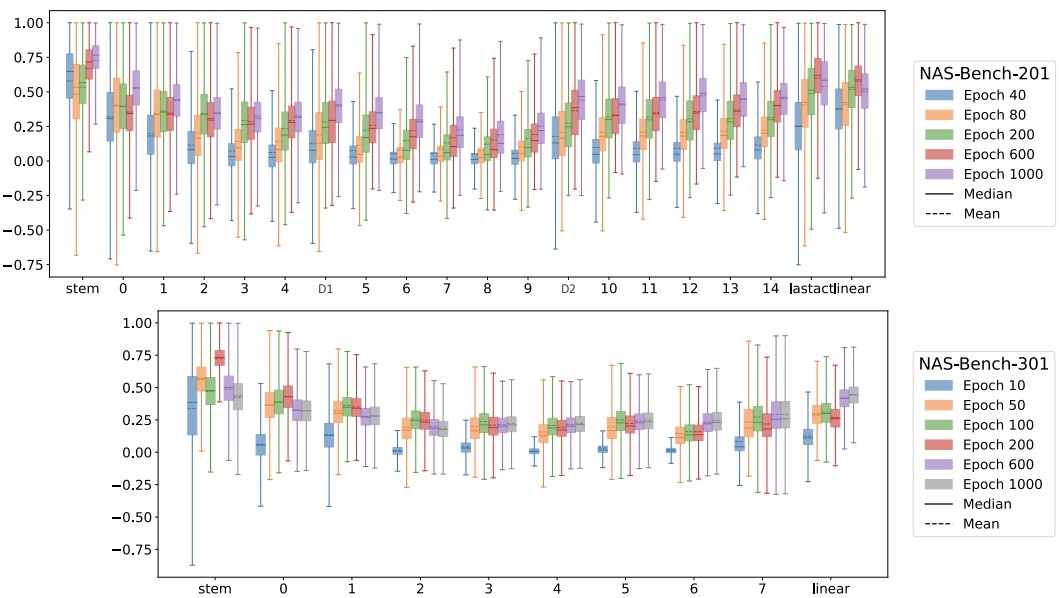

Figure A15: Gradient (cosine) similarity between architecture pairs of different layers at different epochs. On average, architectures tend to have more dissimilar gradients in the middle-stage layers.

exists. To analyze this phenomenon, we plot the histogram of OS accuracy and the scatter of GT-OS scores on NB201 in Fig. A16. We can see from the figure that the average OS accuracy ("avg" in the legend) keeps increasing as the training progresses, and using larger MC samples can bring improvements to the OS accuracy when the training converges. However, too sufficient training can

Table A2: Results of MC sample num ($S$) and FairNAS. Our adapted version of FairNAS samples 5/7 archtectures in each step onNB201/NB301. The training epochs for $S$=1/3/5/7 on NB101 are 1000/330/200/140 (All have similar run time to 1000-epoch training with $S$=1). The training epochs for $S$=1/3/5 and FairNAS on NB201 are 1000/330/200/200 (All have similar run time to 1000-epoch training with $S$=1). And the training epochs for $S$=1/3/5 and FairNAS on NB301 are 1500/500/300/214 (All have similar run time to 1500-epoch training with $S$=1).

| Criteria | NAS-Bench-101 (OS accuracy) | | | | NAS-Bench-201 (OS accuracy) | | | | NAS-Bench-301 (OS loss) | | | |
|---|---|---|---|---|---|---|---|---|---|---|---|---|
| | $S$=1 | $S$=3 | $S$=5 | $S$=7 | $S$=1 | $S$=3 | $S$=5 | FairNAS | $S$=1 | $S$=3 | $S$=5 | FairNAS |
| OS avg | 0.586 | 0.578 | **0.604** | 0.585 | **0.691** | 0.658 | 0.628 | 0.603 | 0.818 | 0.824 | **0.827** | 0.820 |
| KD $\tau$ | 0.384 | 0.405 | 0.442 | **0.446** | **0.744** | 0.709 | 0.714 | 0.706 | 0.530 | 0.515 | **0.548** | 0.527 |
| SpearmanR | 0.541 | 0.568 | 0.616 | **0.624** | **0.910** | 0.890 | 0.895 | 0.887 | 0.720 | 0.705 | **0.739** | 0.715 |
| BR@0.5% | 0.003 | **0.002** | 0.005 | 0.025 | **0.002** | **0.002** | 0.016 | 0.009 | 0.001 | **0.000** | **0.000** | **0.000** |
| P@top 5% | 0.509 | **0.515** | 0.396 | 0.244 | **0.467** | 0.349 | 0.206 | 0.292 | **0.495** | 0.253 | 0.403 | 0.380 |

Table A3: Results of MC sample num ($S$) and FairNAS when the training converges (the learning rate is decayed to less than 1e-5). Our adapted version of FairNAS samples 5/7 archtectures in each step on NB201/NB301. The training epochs for $S$=1/3/5/7 on NB101 are 1000/350/340/380. The training epochs for $S$=1/3/5 and FairNAS on NB201 are all 1000 epochs. And the training epochs for $S$=1/3/5 and FairNAS on NB301 are 3000/1100/600/600.

| Criteria | NAS-Bench-101 (OS accuracy) | | | | NAS-Bench-201 (OS accuracy) | | | | NAS-Bench-301 (OS loss) | | | |
| | $S$=1 | $S$=3 | $S$=5 | $S$=7 | $S$=1 | $S$=3 | $S$=5 | FairNAS | $S$=1 | $S$=3 | $S$=5 | FairNAS |
| --- | --- | --- | --- | --- | --- | --- | --- | --- | --- | --- | --- | --- |
| OS avg | 0.586 | 0.578 | 0.614 | **0.661** | 0.691 | 0.763 | **0.766** | 0.740 | 0.817 | 0.823 | 0.827 | **0.830** |
| KD $\tau$ | 0.384 | 0.405 | 0.446 | **0.489** | **0.744** | 0.673 | 0.659 | 0.711 | 0.530 | 0.540 | **0.543** | 0.531 |
| SpearmanR | 0.541 | 0.568 | 0.622 | **0.672** | **0.910** | 0.854 | 0.877 | 0.887 | 0.723 | 0.729 | **0.735** | 0.718 |
| BR@0.5% | **0.003** | **0.003** | 0.007 | 0.040 | 0.002 | 0.004 | 0.009 | **0.000** | 0.001 | **0.000** | **0.000** | **0.000** |
| P@top 5% | 0.509 | **0.516** | 0.316 | 0.193 | **0.467** | 0.377 | 0.379 | 0.458 | **0.443** | 0.440 | 0.436 | 0.377 |

---

**Algorithm 1** Our adapted FairNAS [5] sampling strategy for NB201 / NB301.

$S$: the number of rollout samples each iteration
$N$: the number of operation choices in the search space
$M$: the number of operations in a cell
$o_i$: the $i$-th operation in the operation choices list
$S = N$
**Initialization**: $P \leftarrow Array(M, N), i \leftarrow 0, j \leftarrow 0, Samples \leftarrow \emptyset$
**while** $i < M$ **do**
    $P[i] \leftarrow$ random permute $(o_0, o_1, ..., o_{N-1})$
    $i + +$
**end while**
**while** $j < S$ **do**
    $Samples.add(P[*, j])$
    $j + +$
**end while**
**Output:** $Samples$ (each row is an $M$-dim array representing an architecture)

---

be detrimental to the ranking quality when $S > 1$. For example, the KDs of $S = 3/5$ and FairNAS at epoch 1000 (the third row in Fig. A16) decrease from 0.749/0.742/0.747 to 0.673/0.659/0.711, respectively. This is because, in the later training stages, the OSEs are mainly increasing the scores of relatively poor architectures. And *when the distribution of the OS scores are concentrated (See "std" in the legend), the overall KD degrades.* Luckily, on NB201, we do not witness significant degradation of OSEs' ability in distinguishing the top architectures (See "P@top 5%" in the legends). On NB201, the best ranking quality achieved by $S = 3/5$ and FairNAS is comparable to that of $S = 1$. In addition, using $S = 1$ does not suffer from the degradation phenomenon[1]. With everything considered, *using $S = 1$ achieves the best results on NB201*.

*As for NB101/NB301, although using multiple MC samples brings small KD improvements, it cannot improve the OSE's ability in distinguishing good architectures* (See "P@top 5%" row). To summarize, *there is no need to use multiple MC samples on these search spaces*.

### A.3.3 De-Isomorphic Sampling

We show the average standard deviations (stds) of the OS scores and rankings within isomorphic groups during the training process in Fig. A17. *As the training progresses, the intra-(isomorphic-)group std gradually shrinks, which indicates that more sufficient training enables OSE to handle isomorphic architectures better*.

Nevertheless, as we have demonstrated in the main paper Fig. 9, even with rather sufficient training, the supernet still overestimates some simple architectures significantly. We have shown in the paper that *some simple architectures with many isomorphic counterparts are overestimated, and we analyze that this is because the equivalent sampling probability of architectures in larger isomorphism groups is higher, thus the shared parameters are trained towards their desired directions*. We propose to conduct deiso sampling during supernet training to mitigate this type of bias, and Tab. A4 shows the

---

[1]We also experiment with $S = 1$ / LR decay patience=60, and do not witness the degradation phenomenon.

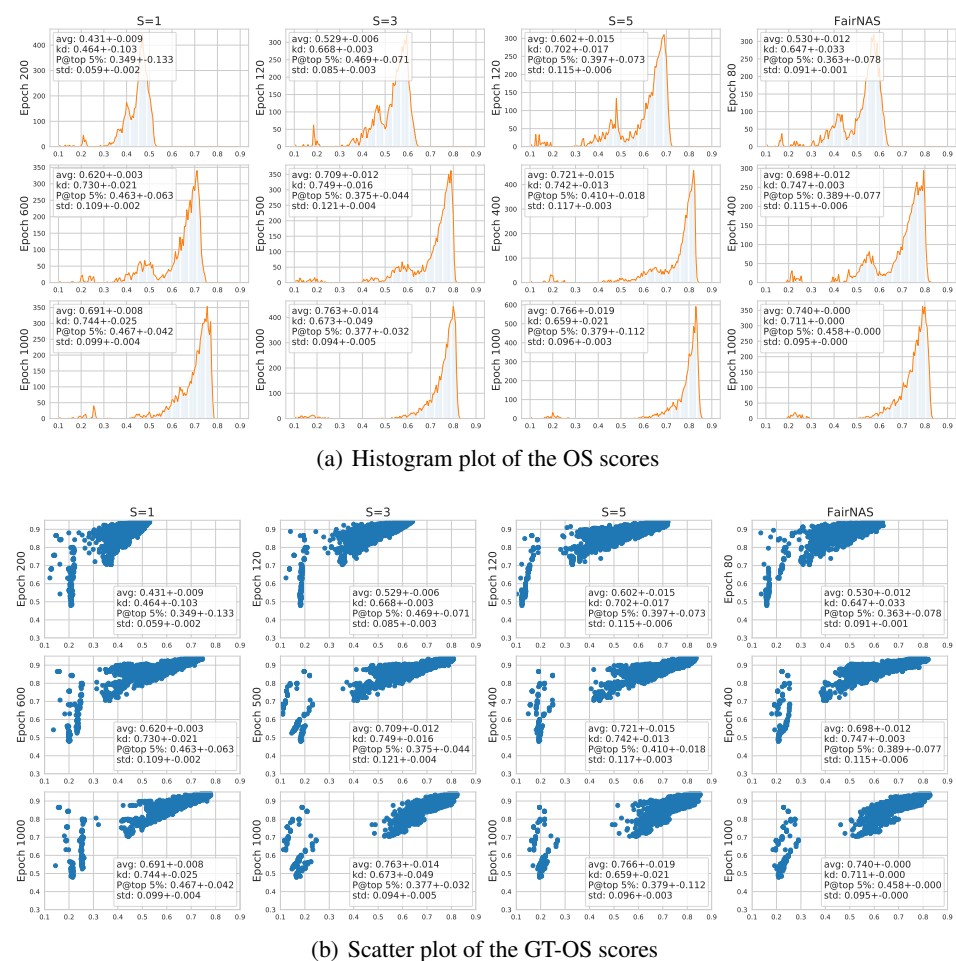

(a) Histogram plot of the OS scores

(b) Scatter plot of the GT-OS scores

Figure A16: The histogram plot of the OS scores and the scatter plot of the GT-OS scores on NB201. If the training is too sufficient when $S > 1$, the ranking quality of OSEs degrades. Legend: "avg" stands for the average one-shot scores; "std" stands for the standard deviation of all one-shot scores.

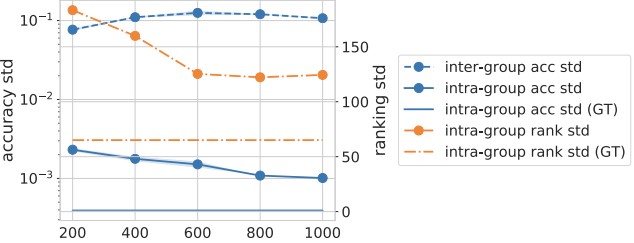

Figure A17: Intra&Inter-Iso group std of accuracy&ranking.

detailed comparison of deiso sampling, iso sampling, and the post-deiso technique. We can see that the supernet trained with the deiso sampling strategy provides better estimations when the training converges. And post-deiso testing achieves slight improvements over "no post-deiso", which might owe to the decreased estimation variances.

It is worth noting that *if the deiso sampling strategy is not used, the quality of the estimations on top architectures decreases as the training progresses*. For example, with iso sampling, P@5% is 21.259% at Epoch 1000, while it is 36.945% at Epoch 200. This indicates that sufficient training exacerbates the bias caused by the imbalance isomorphic group sizes, which is different from the other types of bias analyzed in our study (e.g. complexity-level bias can be alleviated with sufficient

training). *This type of bias belongs to the Type-2 bias that we have analyzed in the main paper* (Sec. 6), *and cannot be mitigated with longer training*: *Architecture are sampled from an unfair distribution, i.e., some architectures have undesirable higher probabilities*. In order to diagnose and mitigate this type of bias, augmenting the sampling strategy is necessary.

Table A4: Comparison of (no) de-isomorphism sampling in supernet training. $\tau$/P@K: Higher is better. BR/WR@K: Lower is better.

| Epochs | criterion | 200 | 400 | 600 | 800 | 1000 |
|---|---|---|---|---|---|---|
| No De-isomorphism (iso) | $\tau$ | 0.5341 | 0.6425 | 0.7049 | 0.7045 | 0.7150 |
| | P@10% | 37.874% | 38.854% | 43.756% | 37.926% | 39.422% |
| | P@5% | 36.945% | 28.586% | 25.284% | 18.369% | 21.259% |
| | BR@1% | 0.541% | 1.547% | 1.706% | 2.454% | 1.887% |
| | BR@0.1% | 1.299% | 1.655% | 11.780% | 3.758% | 5.903% |
| | WR@0.1% | 21.471% | 22.843% | 32.096% | 39.092% | 39.886% |
| Post-De-isomorphism (post-deiso) | $\tau$ | 0.5330 | 0.6431 | 0.7087 | 0.7077 | 0.7178 |
| | P@10% | 38.545% | 40.712% | 45.769% | 40.454% | 41.383% |
| | P@5% | 38.080% | 31.682% | 27.864% | 21.053% | 24.045% |
| | BR@1% | 0.541% | 0.670% | 1.696% | 1.495% | 1.371% |
| | BR@0.1% | 0.588% | 1.660% | 4.217% | 3.737% | 4.346% |
| | WR@0.1% | 21.456% | 23.662% | 30.864% | 21.739% | 30.297% |
| De-isomorphism (deiso) | $\tau$ | 0.4639 | 0.6141 | 0.7296 | 0.7410 | 0.7439 |
| | P@10% | 37.616% | 44.943% | 55.831% | 54.180% | 55.986% |
| | P@5% | 34.778% | 34.985% | 46.440% | 37.668% | 46.749% |
| | BR@1% | 0.938% | 0.144% | 0.052% | 1.794% | 0.227% |
| | BR@0.1% | 3.531% | 6.470% | 0.784% | 4.944% | 2.938% |
| | WR@0.1% | 23.461% | 25.853% | 11.604% | 19.141% | 8.985% |

**Conduct de-isomorphic sampling in practice** To conduct isomorphic sampling, we first design a simple encoding method, which can canonicalize computationally isomorphic architectures to the same string and non-isomorphic architectures to different strings. In our paper, we use the encoding method to find out all isomorphic groups in the search space and make them as a table. Then, during the supernet training process, we sample architecture groups from this table uniformly. This method is feasible since the benchmark search space is not large. In practice, one can use lazy table-making and rejection sampling to conduct de-isomorphic sampling, by only accepting new or representative architecture samples. Specifically, one first encodes each sampled architecture into a canonical string. If this canonical string has not appeared before, this string stands for a new isomorphic group, and the architecture is recorded as the representative architecture for this isomorphic group. This architecture sample is also accepted. If this canonical string has been recorded before, we only accept the architecture sample if it is the representative architecture for its canonical string.

The encoding method goes as follows. Denoting the expression of the $i$-th node as $S_i$, and the operation in the directed edge (j, i) as $A_{ji}$, the expression $S_i$ can be written as

$$S_i = \text{Concat}(\text{Sort}(\{\text{``("} + S_j + \text{")"} + \text{``\%"} + A_{ji}\}_{j \in P(i)})),$$

where $P(i)$ denotes the set of predecessor nodes of $i$, and *Sort* sorts the strings in dictionary order. For NB201, we calculate $S_i(i = 1, \cdots, 4)$ in topological order, and the expression $S_4$ at the final output node is used as the encoding string of the architecture.

### A.3.4  Sharing Extent Reduction

To reduce the sharing extent of the supernet, we experiment with two types of pruning methods (i.e. per-architecture pruning and per-decision pruning). Per-architecture pruning means to throw out certain architectures based on the architecture-level scores, and the outcome of the pruning process is a sub search space (SS) containing the remaining architectures. In contrast, per-decision pruning refers to pruning the space of architectural decisions (e.g., the available operation primitives at some position) instead of directly pruning the architecture space. In the following, we experiment with some cases of these two types of methods.

**Operation Pruning** Operation pruning is a case of per-decision pruning methods. We remove one or two operations in the search space and conduct supernet training on the resulting sub-SS. After

training the supernet, we compare the OS estimations on the sub-SS provided by the supernets trained on full SS and the sub-SS. The results on NB301 and NB201 are shown in Fig. A18 and Tab. A5, respectively.

On NB301, as shown in Fig. A18, compared with the full-SS training, removing one operation in the supernet training process brings non-negative improvements on the average OS accuracies of the remaining architectures, especially in the early training stages. This is intuitive as when fewer architectures are sampled for training, the training for these architectures is more sufficient. Nevertheless, *removing parameterized operations (sep_conv_3x3/5x5, dil_conv_3x3/5x5) leads to better ranking quality on the sub-SS, while removing non-parameterized operations (avg_pool, max_pool, skip_connect) only has slight positive effects in the early training stages*.

On NB201, in general, removing one operation can bring positive or negative impacts on the average OS accuracies on the sub-SS, and the effect is in general positive from -0.011 to +0.0916. However, as for ranking quality on the sub-SS, removing any operations decreases the KD $\tau$ on the sub-SS by 0.014-0.098. Tab. A6 shows the GT accuracy of the top-ranked architectures by OSEs trained on the sub-SS and full-SS. We can see that *a very coarse op-level pruning can neither help the OSE find better architectures* (Tab. A6), *nor help improving the OSE ranking quality on the sub-SS* (Tab. A5).

Table A5: Operation pruning on NB201. The column of "#Archs" shows the number of remaining architectures from 6466 non-isomorphic ones. Note that the tests are conducted on the sub-SS.

| Removed Operation | #Archs | Mean One-shot Accuracy | | Kendall's Tau | |
|---|---|---|---|---|---|
| | | Full-SS training | Sub-SS training | Full-SS training | Sub-SS training |
| skip_connect | 2155 | 0.6991±0.0043 | 0.6881±0.0117 (-0.0110) | 0.7430±0.0088 | 0.7244±0.0040 (-0.0186) |
| nor_conv_1x1 | 1219 | 0.6299±0.0133 | 0.6266±0.0128 (-0.0033) | 0.8100±0.0234 | 0.7120±0.0260 (-0.0980) |
| none | 3131 | 0.6960±0.0102 | 0.7132±0.0198 (0.0172) | 0.7656±0.0306 | 0.7253±0.0255 (-0.0403) |
| nor_conv_3x3 | 1215 | 0.5883±0.0099 | 0.6153±0.0064 (0.0270) | 0.6709±0.0301 | 0.6567±0.0305 (-0.0142) |
| avg_pool_3x3 | 1219 | 0.7240±0.0007 | 0.8156±0.0080 (0.0916) | 0.6072±0.0346 | 0.5816±0.0565 (-0.0256) |
| avg_pool & skip_connect | 114 | 0.6925±0.0056 | 0.7646±0.0237 (0.0721) | 0.6952±0.0142 | 0.6510±0.0506 (-0.0442) |

Table A6: Operation pruning on NB201: GT accuracy of top-ranked architectures by OSEs trained in the sub-SS and full-SS. The column of "#Archs" shows the number of remaining architectures from 6466 non-isomorphic ones.

| Removed Operation | #Archs | BestAcc@top-1 / BestAcc@top-10 (%) | | |
|---|---|---|---|---|
| | | Full-SS training (Full-SS test) | Full-SS training (Sub-SS test) | Sub-SS training (Sub-SS test) |
| skip_connect | 2155 | | 0.9309±0.0001 / 0.9360±0.0023 | 0.9318±0.0011 / 0.9360±0.0001 |
| nor_conv_1x1 | 1219 | | 0.9347±0.0018 / 0.9390±0.0020 | 0.9351±0.0014 / 0.9363±0.0008 |
| none | 3131 | 0.9355±0.0020 / 0.9394±0.0004 | 0.9355±0.0020 / 0.9417±0.0017 | 0.9358±0.0054 / 0.9396±0.0020 |
| nor_conv_3x3 | 1215 | | 0.9038±0.0012 / 0.9102±0.0025 | 0.9055±0.0029 / 0.9107±0.0022 |
| avg_pool_3x3 | 1219 | | 0.9350±0.0020 / 0.9419±0.0013 | 0.9350±0.0030 / 0.9382±0.0040 |
| avg_pool & skip_connect | 114 | | 0.9367±0.0015 / 0.9425±0.0016 | 0.9371±0.0008 / 0.9419±0.0025 |

In summary, sharing extent reduction by removing operations can bring improvements to the average OS scores, especially in the early training stages. However, *whether the improved absolute OS scores can bring ranking quality improvements or help find better architectures is questionable*.

**Per-architecture Hard Pruning** We conduct SS pruning on NB201 by selecting the top 10%, 25%, 50% ranked by the OS scores of supernet (epoch 600 seed 20) out of the 6466 non-isomorphic architectures. We continue to finetune the supernet to 1000 epoch with only the selected architecture samples. The estimation qualities on the sub-SS of the OSEs trained on the full-SS and sub-SS are shown in Tab. A7. We can see that *OS pruning brings consistent improvements on both the average OS score and ranking quality in the sub-SS*: 2.2%/1.3%/0.1% average OS score improvements and 0.189/0.046/0.086 KD improvements when the sub-SS contains 10%/25%/50% architectures. Also, the top-ranked (top-1 and top-10) architectures of the OSEs trained on the sub-SS have better GT performances. This shows that *reducing the sharing extent by OS pruning can improve the OSE quality on good architectures, and thus enable the OSE to find better architectures*.

The above results reveal the potential of dynamic SS pruning for improving the OSE quality, especially for good architectures. However, the per-architecture hard pruning scheme based on the OS scores need an exhaustive test of the full search space, which is not practical for actual use. There are two directions for developing practical dynamic SS pruning methods: 1) Per-architecture (soft) pruning

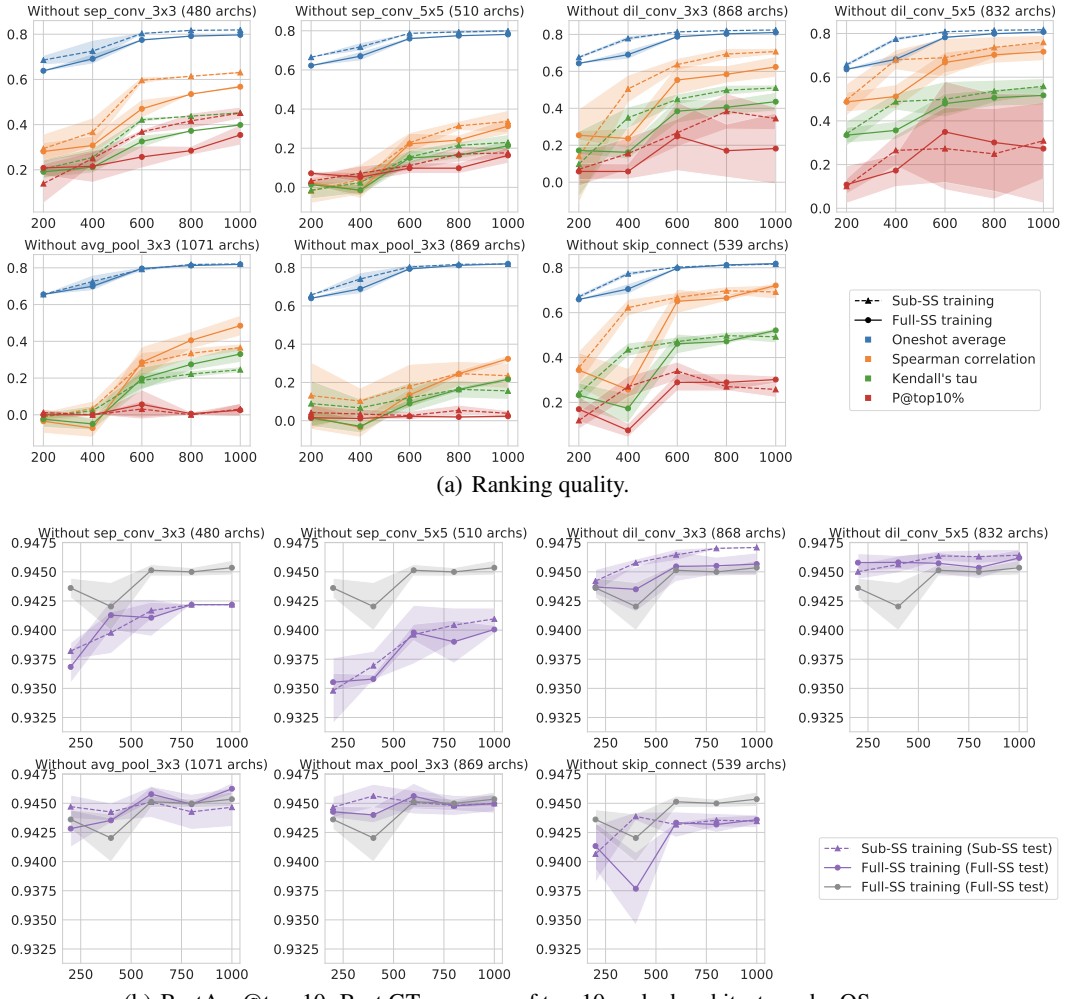

(a) Ranking quality.

(b) BestAcc@top-10: Best GT accuracy of top-10 ranked architectures by OS scores.

Figure A18: Operation pruning on NB301.

with a jointly updated controller, where the controller would give higher sampling probability to the architectures with higher OS scores. 2) Per-decision pruning with a jointly updated controller, where the controller learns to assign different probabilities to architectural decisions instead of architectures. Next, we present a case study on NB301 that jointly updates an evolutionary controller and the supernet, which follows the first direction (i.e. Per-architecture soft pruning).

**Case study: Per-architecture Soft Pruning (with an evolutionary-based controller)**

We use an evolutionary controller, and update the controller along with the supernet training process. We experiment with two types of evolutionary controllers: single-evo and pareto-evo. The controller update is based on the OS rewards/scores estimated by the current supernet, which is very similar to that in non-one-shot NAS methods. There are two alternative phases in a typical non-one-shot parameter-sharing NAS process: Controller update and supernet update. In the supernet update phase, the controller samples $S$ architecture in each step, and the supernet is updated using the accumulated gradients of the $S$ architectures. In the controller update phase of evolutionary controllers, the controller sample $S_c$ architectures, and then the estimated rewards of the $S_c$ architectures (by supernet) are utilized to update the population. And the controller update phase is conducted once every $T_c$ epochs. We summarize how the algorithm works in Tab. A8. And we can see that the only difference between the single-evo controller and the pareto-evo controller lies in the population update step. In the population update, the single-evo controller keeps 100 architectures with the

Table A7: (Per-architecture) one-shot pruning on NB201. The column of "#Archs" shows the number of remaining architectures from 6466 non-isomorphic ones. The number between the parenthesis "()" in the "BestAcc@top-1 / BestAcc@top-10" column is the best (absolute, 1 is the best, 6466 is the worst) GT ranking in the top-1 and top-10 OS ranked architectures (the average result of multi-seed trained supernets is rounded to integer). Note that in the "BestAcc" column, the test is on the full-SS instead of the sub-SS, which is different from the results in "Mean One-shot Accuracy" and "Kendall's Tau".

| Keep Proportion | #Archs | Mean One-shot Accuracy | | Kendall's Tau | | BestAcc@top-1 / BestAcc@top-10 (%) | |
| --- | --- | --- | --- | --- | --- | --- | --- |
| | | Full-SS training | Sub-SS training | Full-SS training | Sub-SS training | Full-SS training | Sub-SS training |
| 10% | 646 | 0.7669 | 0.7890 (+0.0221) | 0.2643 | 0.4532 (+0.1889) | | 94.29 (8) / 94.37 (1) |
| 25% | 1616 | 0.7597 | 0.7722 (+0.0126) | 0.3526 | 0.4386 (+0.0861) | 93.55 (298) / 93.94 (48) | 94.37 (1) / 94.37 (1) |
| 50% | 3233 | 0.7476 | 0.7563 (+0.0088) | 0.5074 | 0.5532 (+0.0046) | | 94.11 (22) / 94.37 (1) |

highest OS rewards from the sampled-and-estimated 200 architectures as the new population. And the pareto-evo controller keeps 100 architectures from the OS-Param Pareto frontier and the OS-Inverse Param Pareto frontier. Our construction of the pareto-evo controller is very similar to that in [16]. The motivation of including the OS-Inverse Param frontier into the population is that since the OSEs might underestimate small architectures in the early training phase, the architectures with the largest parameter sizes should also be put in the population for further training.

Table A8: The detailed settings in our evolutionary-based per-architecture soft pruning study. Random

| Components & Interfaces | | Single-evo | Pareto-evo |
| --- | --- | --- | --- |
| Population size | | 100 | |
| Controller update | Sampling $S_c = 200$ | Whole population for 100 + 50% random, 25% crossover, 25% mutate for another 100 | |
| | Population update | Keep 100 with the highest OS rewards | Keep 100 from the pareto & inverse param pareto frontier |
| | Update every $T_c$ epoch | 10† | |
| Supernet update | Sampling | 1) 100% population 2) 50% random, 50% population | |
| | Warmup epochs | 50† | |

†: We also experiment with $T_c = 5$ and warmup epochs=100/200, and do not observe consistent improvements over the setting $T_c = 10$, warmup epochs = 50. The training process with $T_c = 5$ is about $2\times$ slower than the training process with $T_c = 10$, thus we only demonstrate the results obtained with $T_c = 10$.

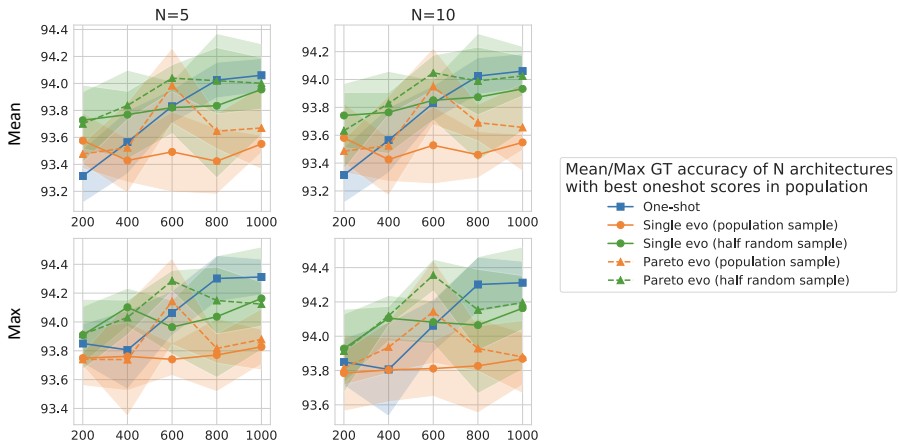

Figure A19: Per-architecture soft pruning with evolutionary-based controllers on NB301: Quality comparison of the OSE estimations after one-shot training and controller-guided training.

For the vanilla one-shot training, we run a tournament-based evolutionary method [12] on the trained supernet at Epoch 200, 400, 600, 800, 1000. For the supernet trained jointly with the evolutionary

controllers, instead of directly taking the controller population at each epoch, we also run the same tournament-based evolutionary method [12] on the supernet at Epoch 200/400/600/800/1000 for a fair comparison. After running the tournament-based evolutionary method, the mean and max GT accuracy of $N$ architectures with the best one-shot scores in the tournament-based controller's population are shown in Fig. A19.

As described in Tab. A8, we experiment with two options of sampling architectures in the supernet update phase: 1) 100% population: Random sample from the population; 2) 50% random, 50% population: Random sample from the whole search space with 50% probability, and random sample from the population otherwise. The first option is the original one adopted by [16], and as shown by the orange lines in Fig. A19, they cannot achieve satisfying results. In this case, even when the supernet is trained longer, the supernet cannot help find better architectures. We analyze that this is because only training the supernet using the population architectures leads to local optimum trapping that the search zooms in onto some architectures too quickly.

As a quick remedy, we experiment with the second option, which is an intermediate between fully random sampling (one-shot) and controller-guided sampling (green lines). We can see that the *supernet-update sampling option 2 with half random sample (green lines) is generally better than the original option 1 (orange lines)* [16], and the mean/max GT accuracy of the discovered top architectures increase as the training goes on. Also, *the pareto-evo controller (green/orange dashed lines) is slightly better than single-evo controller* (green/orange solid lines). However, compared with the one-shot trained supernet (blue line), jointly training the supernet with a per-architecture controller can only bring small improvements in the early training stages. We analyze that this is because the SS is too large such that it is hard for a per-architecture controller to balance the exploration (sufficient training of supernet on lots of architectures) and exploitation (low sharing extent).

Based on the above case study, we regard the per-decision pruning scheme as a more promising choice to dynamically reduce the sharing extent for better OSE training, since the factorized decision spaces are much smaller to assign a proper sampling distribution. Several studies have proposed per-decision search space pruning methods. Hu et al. [8] propose an angle-based metric to shrink the search space progressively. A recent work [6] presents a search space evolving scheme. In each iteration, the supernet is tuned on a sub-search space with only a subset of decision values for each decision (one decision for each layer). Then, after a time-consuming process to get the one-shot Pareto frontier, the union of decision values (type of operations) from all P Pareto-optimal architectures, together with some newly included decision values, are used to assemble the sub search space in the next iteration.

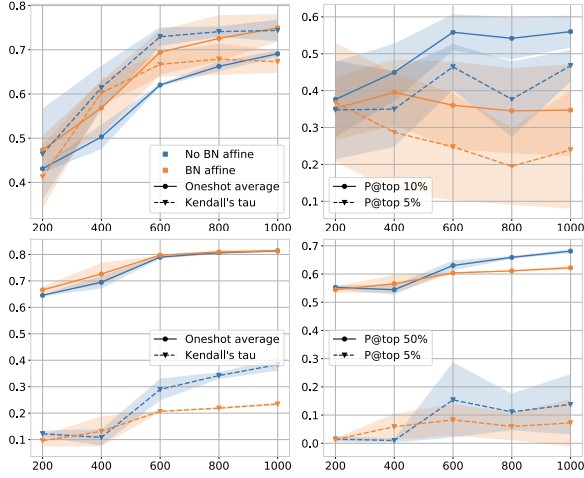

Figure A20: Effect of BN affine operations in OSE. Top / Bottom: NB201 / NB301.

**Influences of BN Affine** Fig. A20 compares the ranking quality of the OSEs that are trained with or without affine operations in BNs.

# B Discussions and Results about Zero-shot Estimators

Our work evaluates six parameter-level ZSEs [1] and two architecture-level ZSEs [9, 10]:

1. Abdelfattah et al.[1] uses the summation of parameter-wise sensitivity as the architecture-level score. The parameter-wise sensitivity calculation methods are from zero-shot pruning literatures: *grad_norm* simply sums up gradients norms of all parameters; *plain*, *snip*, *plain* and *grasp* take the Hadamard product of gradients and parameters into account; *synflow* is proposed to avoid layer-collapse in zero-shot pruning without using any mini-batch data; *fisher* considers the loss changes when removing certain activation channels. Denoting the parameters, activations, and loss as $\theta, z$, and $\mathcal{L}$, the formula of these parameter-wise sensitivity indicators can be written as

$$plain: \mathcal{S}(\theta) = \frac{\partial \mathcal{L}}{\partial \theta} \odot \theta; \quad snip: \mathcal{S}(\theta) = |\frac{\partial \mathcal{L}}{\partial \theta} \odot \theta|; \quad grasp: \mathcal{S}(\theta) = -(H\frac{\partial \mathcal{L}}{\partial \theta}) \odot \theta;$$
$$synflow: \mathcal{R} = \mathbb{1}^T (\prod_{\theta_i \in \theta} |\theta_i|) \mathbb{1}, \mathcal{S}(\theta) = \frac{\partial \mathcal{R}}{\partial \theta} \odot \theta; \quad fisher: \mathcal{S}(z) = \sum_{z_i \in z} (\frac{\partial \mathcal{L}}{\partial z} z)^2. \tag{1}$$

2. Mellor et al. [9] measures the architecture's discriminability of different inputs by

$$S = -\sum_{i=1}^{N} [\log(\sigma_{J,i} + k) + (\sigma_{J,i} + k)^{-1}] \tag{2}$$

where $\sigma_{J,1}, \sigma_{J,2}, \cdots \sigma_{J,N}$ is the eigenvalues of the covariance matrix $\Sigma_J$ of the input jacobian $J$: $\Sigma_J = \text{cov}(J, J)$. $J = (\frac{\partial f_1}{\partial x_1}, \frac{\partial f_2}{\partial x_2}, \cdots, \frac{\partial f_N}{\partial x_N})$ is the jacobian of $N$ images $x_1, x_2, \cdots, x_N$ in a batch.

3. Mellor et al. [10] propose another zero-shot measure of the architecture discriminability. Instead of utilizing the high-order gradients at the input data, they measure the activation differences at all ReLU layers between different input images as the architecture score:

$$s = \log ||K_H||$$
$$\text{where } K_H = \begin{pmatrix} N_A - d_H(c_1, c_1) & \cdots & N_A - d_H(c_1, c_N) \\ \vdots & \ddots & \vdots \\ N_A - d_H(c_N, c_1) & \cdots & N_A - d_H(c_N, c_N) \end{pmatrix}, \tag{3}$$

where $c_i$ is a binary mask indicating whether each feature value is larger than 0 at all ReLU layers for input data $i$. And $d_H(c_i, c_j)$ is the Hamming distance between the binarized activation code of the $i$-th and the $j$-th data.

Besides the results shown in the manuscript, this section demonstrates more detailed results and analysis of these ZSEs.

## B.1 Evaluation of Zero-shot Estimators

As shown in Tab. A9 and Fig. A21, *ZSEs perform very poorly on NB101-1shot*. For most of ZSEs (except relu_logdet), their ranking quality is not only worse than the OSEs, but even worse than directly using parameters or FLOPs as the estimation score. *synflow*, *snip*, and *grad_norm* estimators even have negative KD.

As shown in Tab. A9, *on all three search spaces, the ranking qualities of OSEs surpass all ZSEs (except relu_logdet on NB301), and the best KD achieved by the ZSEs except relu_logdet cannot even beat the KD between the GT accuracies and the parameter sizes*. The situation on the easiest NB201 is the best for the ZSEs: Both *relu_logdet* and *jacob_cov* achieve comparable KDs and SpearmanRs as the GT-Param correlations (KD: 0.611 & 0.608 V.S. 0.606; SpearmanR: 0.798 & 0.788 V.S. 0.784). And the best ZSE based on parameter-wise analysis, *synflow*, can only achieve a KD of 0.573. On harder search spaces, the ranking qualities of ZSEs based on parameter-wise analysis are more questionable. Another thing is that, *although relu_logdet and jacob_cov achieve relatively good KD on NB201, their ability in distinguishing the top architectures is weak*: P@top 5%

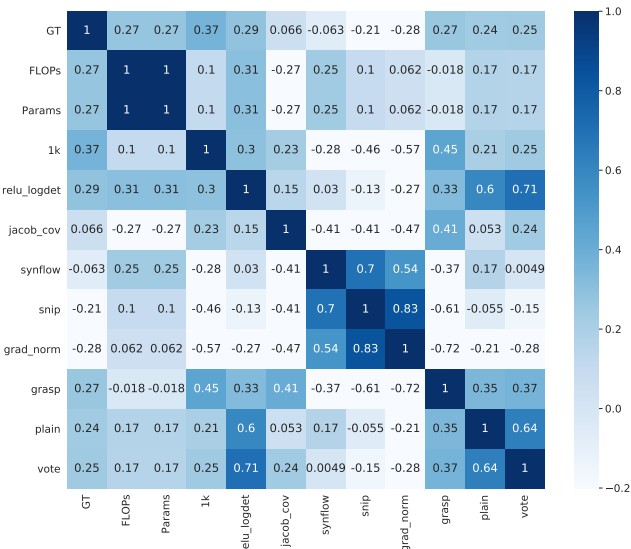

Figure A21: Kendall's Tau between GT, FLOPs/Params, OSEs (1k epoch) and ZSEs on NB101-1shot.

of *jacob_cov* (8.7%) is the lowest among all ZSEs, and P@top 5% of *relu_logdet* (15.8%) is also low. As a comparison, *synflow*'s P@top 5% is 32.5%, and OSE's P@top 5% is 53.3%.

The formula of *relu_logdet* in Eq. 3 indicates that the *relu_logdet* score would be highly correlated to the number of ReLU layers. Based on this speculation, we compute the number of ReLU layers in each architecture and denoted this value as the *relu* score. *relu* is a data-independent score purely calculated according the structure information, thus we list it in Tab. A9 together with Param and FLOPs. We can see that *simply counting the number of ReLU layers as the architecture score is also a competitive data-independent baseline of ZSEs on these topological search spaces*. And we also observe that the KDs between *relu_logdet* and *relu* on these three topological benchmarks are rather high: 0.60 on NB101, 0.88 on NB201, and 0.88 on NB301.

Table A9: Comparison of ZSEs' and OSEs' KD / SpearmanR / P@top 5% / P@bottom 5% / BR@5%. OS accuracy is used as the OS score on NB101 / NB201, and OS loss is used as the OS score on NB301. Best-performing ZSEs are chosen as the voting experts in the *vote* ZSE: *relu_logdet/grasp/plain* on NB101, *relu_logdet/jacob_cov/synflow* on NB201, *plain/grasp/relu_logdet* on NB301.

| ZSE | NAS-Bench-101-1shot | NAS-Bench-201 | NAS-Bench-301 |
|---|---|---|---|
| *synflow* | -0.063 / -0.090 / 0.016 / 0.024 / 0.028 | 0.573 / 0.769 / **0.325** / 0.489 / **0.000** | 0.201 / 0.299 / 0.020 / 0.265 / 0.030 |
| *grad_norm* | -0.276 / -0.397 / 0.000 / 0.016 / 0.123 | 0.401 / 0.546 / 0.108 / 0.539 / 0.011 | 0.070 / 0.105 / 0.017 / 0.095 / 0.030 |
| *snip* | -0.206 / -0.305 / 0.000 / 0.008 / 0.123 | 0.402 / 0.547 / 0.115 / 0.536 / 0.011 | 0.050 / 0.075 / 0.010 / 0.095 / 0.030 |
| *grasp* | 0.266 / 0.378 / 0.052 / **0.148** / 0.002 | 0.348 / 0.496 / 0.102 / 0.031 / 0.008 | 0.365 / 0.525 / 0.082 / 0.214 / 0.004 |
| *fisher* | - | 0.362 / 0.495 / 0.093 / 0.514 / 0.011 | -0.158 / -0.239 / 0.010 / 0.051 / 0.030 |
| *plain* | 0.240 / 0.346 / 0.012 / 0.100 / 0.030 | 0.311 / 0.458 / 0.096 / 0.121 / 0.002 | 0.394 / 0.565 / 0.068 / 0.286 / 0.007 |
| *jacob_cov* | 0.066 / 0.100 / 0.196 / 0.012 / **0.000** | 0.608 / 0.788 / 0.087 / 0.734 / 0.002 | 0.230 / 0.339 / 0.041 / 0.201 / 0.007 |
| *relu_logdet* | **0.290** / **0.421** / **0.252** / 0.144 / 0.000 | 0.611 / **0.798** / 0.158 / **0.799** / 0.003 | **0.539** / **0.736** / **0.296** / **0.357** / 0.006 |
| *vote* | 0.253 / 0.369 / 0.132 / 0.104 / 0.001 | 0.587 / 0.777 / 0.241 / 0.613 / 0.000 | 0.372 / 0.533 / 0.082 / 0.238 / **0.003** |
| *relu* | 0.208 / 0.278 / 0.072 / 0.124 / 0.001 | **0.613** / 0.755 / 0.223 / 0.647 / 0.000 | 0.521 / 0.709 / 0.238 / 0.354 / 0.006 |
| *Param* | 0.274 / 0.394 / 0.112 / 0.152 / 0.000 | 0.606 / 0.784 / 0.282 / 0.551 / 0.000 | 0.515 / 0.709 / 0.286 / 0.347 / 0.006 |
| *FLOPs* | 0.274 / 0.394 / 0.112 / 0.152 / 0.000 | 0.606 / 0.784 / 0.282 / 0.551 / 0.000 | 0.487 / 0.678 / 0.282 / 0.350 / 0.006 |
| *OS (1k epoch)* | **0.369** / **0.521** / **0.480** / **0.260** / **0.000** | **0.766** / **0.925** / **0.533** / **0.882** / **0.000** | 0.515 / 0.708 / 0.330 / **0.395** / **0.002** |
| *OS (1500 epoch)* | - | - | **0.534** / **0.726** / **0.435** / **0.398** / **0.000** |

Since NB201 also provides the architecture GT performances on CIFAR-100 and ImageNet-16-120 datasets, we conduct several experiments to evaluate the ranking quality of ZSEs on these two datasets. As shown in Tab. A10, ZSEs can get a relatively stable performance on NB201, no matter what dataset is used. Then we explore whether the rankings provided by ZSEs change when using different input data distribution (i.e., using data batches from different datasets). We summarize the KD of the GT accuracies and ZSE scores on the three datasets in Fig. A22. And we can see that *most ZSEs except plain are not sensitive to the input data distribution*: their rankings on different datasets are

highly correlated. Actually, most ZSEs get similar architecture rankings even when using uniform and Gaussian random noises as the input. And the ranking quality of ZSEs on a certain dataset might be suboptimal when using its own data batches. This indicates that the architecture performance estimations provided by most ZSEs except *plain* have small and nonideal dependencies on the input distribution.

Table A10: Comparison of ZSEs' and OSEs' KD / SpearmanR / P@top 5% / P@bottom 5% / BR@5% across the CIFAR-10, CIFAR-100, ImageNet-16-120 datasets on NB201. OS accuracy is used as the OS score.

| ZSE | CIFAR-10 | CIFAR-100 | ImageNet-16-120 |
|---|---|---|---|
| *synflow* | 0.573 / 0.769 / 0.325 / 0.489 / 0.000 | 0.549 / 0.743 / **0.362** / 0.433 / 0.000 | 0.511 / 0.695 / **0.347** / 0.492 / 0.001 |
| *snip* | 0.402 / 0.547 / 0.115 / 0.536 / 0.011 | 0.401 / 0.539 / 0.102 / 0.529 / 0.268 | 0.346 / 0.463 / 0.115 / 0.396 / 0.446 |
| *plain* | 0.311 / 0.458 / 0.096 / 0.121 / 0.002 | 0.415 / 0.591 / 0.087 / 0.173 / 0.018 | 0.299 / 0.441 / 0.118 / 0.053 / 0.020 |
| *jacob_cov* | 0.608 / 0.788 / 0.087 / 0.734 / 0.002 | 0.641 / 0.821 / 0.118 / **0.746** / 0.016 | 0.585 / 0.766 / 0.074 / 0.697 / 0.057 |
| *relu_logdet* | 0.611 / 0.798 / 0.158 / 0.799 / 0.003 | 0.636 / 0.819 / 0.201 / 0.799 / 0.011 | 0.597 / 0.780 / 0.316 / 0.656 / 0.004 |
| *Param* | 0.606 / 0.784 / 0.282 / 0.551 / 0.000 | 0.569 / 0.745 / 0.356 / 0.529 / 0.000 | 0.507 / 0.675 / 0.238 / 0.498 / 0.002 |
| *FLOPs* | 0.606 / 0.784 / 0.282 / 0.551 / 0.000 | 0.569 / 0.745 / 0.356 / 0.529 / 0.000 | 0.507 / 0.675 / 0.238 / 0.498 / 0.002 |
| *OS (1k epoch)* | **0.766** / **0.925** / **0.533** / **0.882** / **0.000** | **0.645** / **0.837** / 0.350 / 0.601 / **0.000** | **0.640** / **0.832** / 0.269 / **0.774** / **0.000** |

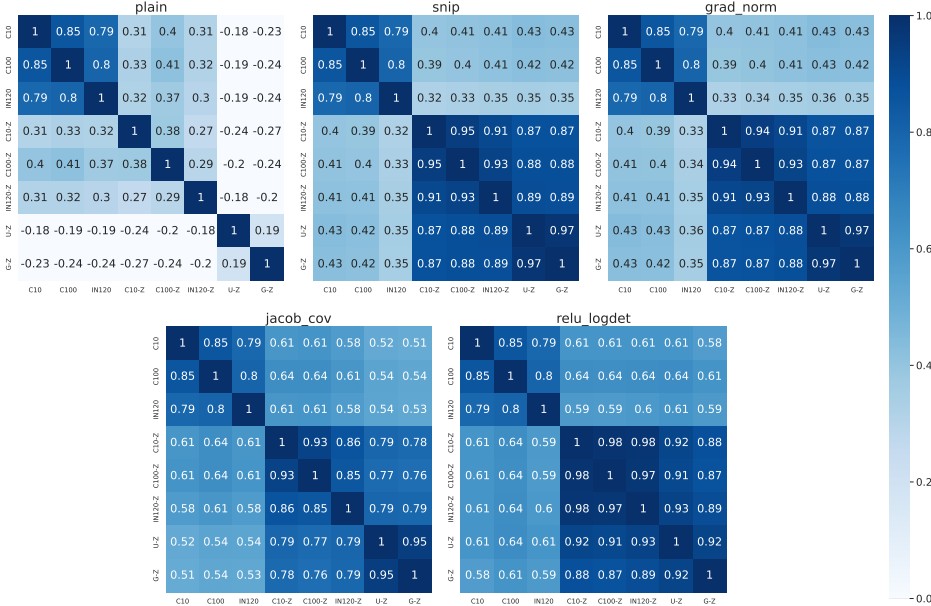

Figure A22: KDs between GT accs and ZSE scores on three datasets. "C10", "C100" and "IN120" represent the GTs on CIFAR-10, CIFAR-100 and ImageNet-16-120, respectively, while "C10-Z", "C100-Z" and "IN120-Z" represent the ZSE scores on these datasets. "U-Z" and "G-Z" represent the ZSE scores using uniform and Gaussian noises as the input, respectively.

We explore whether ZSEs can benefit from OS training and show the results in Tab. A11. We can see that the quality of ZSEs except *relu_logdet* significantly decreases as the training process goes on. That is to say, *ZSEs that utilize high-order information (i.e., gradients) provide the best estimations with randomly initialized weights.* One possible explanation is that the gradients are noisier and of smaller magnitudes in trained models, so that the ranking quality of these gradient-based ZSEs degrades.

Table A11: Quality of zero-shot estimations as training processes.

| ZSE | NAS-Bench-101 (Epoch 0/200/800) | | NAS-Bench-201 (Epoch 0/40/1000) | | NAS-Bench-301 (Epoch 0/200/800) | |
|---|---|---|---|---|---|---|
| | KD $\tau$ | P@top5% | KD $\tau$ | P@top5% | KD $\tau$ | P@top5% |
| *synflow* | -0.063 / -0.015 / 0.128 | 0.016 / 0.008 / 0.008 | 0.573 / 0.565 / 0.423 | 0.321 / 0.337 / 0.297 | 0.200 / -0.379 / -0.241 | 0.020 / 0.000 / 0.000 |
| *grad_norm* | -0.276 / -0.265 / -0.302 | 0.000 / 0.000 / 0.000 | 0.403 / 0.261 / -0.149 | 0.106 / 0.009 / 0.006 | 0.069 / -0.027 / -0.060 | 0.017 / 0.013 / 0.006 |
| *snip* | -0.206 / -0.105 / -0.215 | 0.000 / 0.000 / 0.000 | 0.405 / 0.231 / 0.083 | 0.106 / 0.006 / 0.018 | 0.050 / -0.412 / -0.314 | 0.010 / 0.000 / 0.000 |
| *jacob_cov* | 0.066 / -0.165 / 0.009 | 0.196 / 0.004 / 0.196 | 0.608 / 0.437 / -0.044 | 0.086 / 0.024 / 0.000 | 0.230 / -0.317 / -0.285 | 0.040 / 0.000 / 0.000 |
| *relu_logdet* | 0.290 / 0.325 / 0.325 | 0.252 / 0.236 / 0.264 | 0.611 / 0.556 / 0.628 | 0.158 / 0.183 / 0.201 | 0.539 / 0.531 / 0.528 | 0.296 / 0.241 / 0.248 |

## B.2 Bias of Zero-shot Estimators

**Architecture-level & Op-level Bias** We show the top ranked architectures of various ZSEs on NB201, NB301 and NB101 in Fig. A25, Fig. A27, and Fig. A28, respectively. The operation-level bias on NB201 can also be witnessed from Fig. A26 that shows the scatter plot of GT-ZS scores.

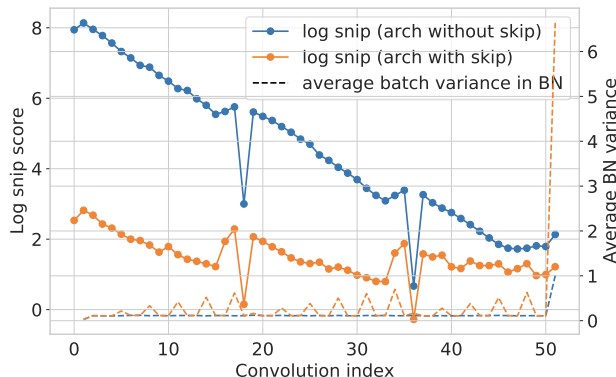

Figure A23: Convolution-wise *snip* values (logarithm) of architectures with / without a 0-3 skip connection. The dashed lines give out the average batch variance in BNs. And the peaks on the orange dashed line correpond to the BNs after the 0-3 skip connections in all the 15 searchable cells.

From Fig. A25 and Fig. A26, we can see that *synflow clearly prefer the largest architectures on NB201*, and we show a more formal explanation of why it is the case in Sec. B.3. *Three other ZSEs, snip, grad_norm and fisher, give similar rankings of architecture. These three ZSEs and grasp all show improper preference on architectures with gradient explosion*. More specifically, these ZSEs show a clear preference for architectures without skip connections, which are far from optimal. For example, the top-1 architectures ranked by *snip*, *grad_norm* and *fisher* is a linear architecture with three 3x3 convolutions, and the ZS scores are 3063, 1922, and 454.8, respectively. This linear architecture has a GT accuracy of 88.52%, and when a skip connection is added from node 0 to node 3, the GT accuracy increases to 93.99%. However, the ZS scores of the architecture with a 0-3 skip connection degrades to 36.66, 18.74, and 0.019, respectively. We plot the logarithm of convolution-wise *snip* values of these two architectures in Fig. A23 when inputting a random data batch, and we can see that in the architecture without 0-3 skip connections, the *snip* score grows by almost three orders of magnitudes $\sim 405\times$ from the last convolution (8.44) to the first convolution in cell ($3.42\times10^3$). While in the architecture with 0-3 skip connections, the *snip* score grows at a much slower pace (only $4.96\times$) from the last convolution (3.38) to the first convolution in cell (16.75). The phenomenon of *grad_norm* and *fisher* is similar. The skip connections prevent gradient explosion, since that the batch variance in the BN after each skip connection is several times larger due to the aggregated feature map, and then the backpropagated gradient would be several times smaller.

To summarize, *the skip connections effectively prevent the gradient explosion issue that indicates the inferiority of the architecture, while these ZSEs have an undesired preference on architectures with exploding gradients*. Actually, due to the exponentially increasing scores shown in Fig. A23, we can see that *these three ZS indicators designed for measuring parameter-wise sensitivity are even not suitable for measuring layer-wise sensitivity in some architectures*.

As for the *relu_logdet* and *jacob_cov* estimator, they achieve the KD of 0.611 and 0.608 on NB201, which are the best among the ZSEs. However, *both relu_logdet and jacob_cov show an improper*

*preference on 1×1 convolution over 3×3 convolution, which account for their weak performance in identifying good architectures*: As shown in Tab. A9, *relu_logdet* has a relatively low P@top 5% of 15.8%, while *jacob_cov* has a lowest P@top 5% of 8.7% among all ZSEs, not to speak of the comparison with OSEs (OSE at 1000 epoch achieves a P@top 5% of 53.3%). And as shown in Fig. A26 and Fig. A25, the *plain* estimator overestimate the avg_pool_3x3 operation.

As shown in Fig. A27, *on NB301, synflow still perfers the largest architectures, and jacob_cov prefers shallower architectures and non-parametrized operations*. Another observation is that apart from a few outlier values for the worst architectures, most *jacob_cov* scores are distributed in a very small range from -132.67 (0.1 quantile) to -132.26 (0.9 quantile) and -131.84 (1.0 quantile). *The highly concentrated distribution means that it is very difficult for jacob_cov to distinguish different architectures on the harder NB301 space* with smaller inter-architecture differences. This explains why the P@top 5% of *jacob_cov* is as low as 4.1% on NB301 (Tab. A9) (the OSE at 1500 epoch achieves a P@top 5% of 43.5%). Compared to all other ZSEs, *relu_logdet* performs best on NB301, with competitive KD/SpearmanR/P@bottom 5% even with OSE (KD: 0.539 V.S. 0.534, SpearmanR: 0.736 V.S. 0.726, P@bottom 5%: 35.7% V.S. 39.8%), according to Tab. A9.

As for NB101, Fig. A28 shows that *synflow still prefers the largest architectures. And jacob_cov and relu_logdet still show an improper preference on 1×1 convolution over 3×3 convolution, which account for their relatively low P@topKs* (See Tab. A9). And the same as our observation on NB201, *plain overestimate non-parametrized operation.*

**Complexity-level Bias** Fig. A29 shows the complexity-level bias of the ZSEs on NB101, NB201 and NB301.

### B.3 Preference Analysis of the *synflow* ZSE

The *synflow* indicator [1, 14] proposes to change all parameters to their absolute values, remove BNs and nonlinear functions, input an all-1 tensor, add up the final feature map as the loss, and then accumulate the multiplication of the loss gradient and magnitude of all parameters. Here, we want to demonstrate three statements when introducing new convolutions into an architecture: 1) The expected loss gradients w.r.t. existing parameters become larger. 2) And since the *synflow* of each parameter is the multiplication of the absolute parameter value and the loss gradient (also positive), the expectation of each *synflow* value increases. 3) And as the number of parameters also increases, the overall *synflow* of the architecture increases. Since the latter two reasoning is obvious, we only need to prove the first statement: "When introducing new convolutions, the expected loss gradients w.r.t. existing parameters become larger".

For simplicity, we study the case of adding a new MLP layer into an MLP architecture to demonstrate our intuition. After *synflow*'s modifications, the architecture is turned into linear transformations, with all weights being positive. For example, an architecture takes $\mathbf{1}_i \in R^{K_i \times 1}$ (the $i$ subscript denotes "input") as the inputs and adds up the output vector of the last MLP layer with weight $W_N$ to get the loss $L$. When a new MLP layer with weight $W_c$ is added into the architecture, $W_c$ split the original architecture into two parts: the previous layers $W_1, \cdots, W_n$, and the latter layers $W_{n+1}, ... W_N$. Since all operations are linear, we use a matrix $W_l = f_l(W_1, ..., W_n) \in R^{K_l \times K_i}$ to substitute all the previous layers, $W_r = f_r(W_{n+1}, \cdots, W_N) \in R^{K_o \times K_l}$ to substitute all the latter layers. Note that whether these two parts contain multiple branches does not influence the fact that their overall computation is linear. And when there exist other branches from the previous part to the latter part, we can only consider the branch with $W_c$ on it. This is because all computations are fully linear, no matter where the branches are merged, $L$ can be decomposed into several accumulation terms with some shared matrices. Thus for each parameter $w$, either $\frac{\partial L}{\partial w}$ is unrelated to the newly added $W_c$ since they are on parallel branches, or $\frac{\partial L}{\partial w}$ can be written as the sum of an unchanged gradient term and another gradient term that is changed due to $W_c$. We'll show that the introduction of $W_c$ causes all the related parameters' expected gradient magnitudes to become larger.

The loss term calculated by the original architecture is $L = \mathbf{1}_l^T W_r W_l \mathbf{1}_i$, where $\mathbf{1}_l \in R^{K_o \times 1}$, and $K_o$ is the output vector dimension. After adding a new MLP layer, the loss term related to $W_c$ becomes $\tilde{L} = \mathbf{1}_l^T W_r W_c W_l \mathbf{1}_i$, where $W_c \in R^{K_l \times K_l}$. Let us compare $g_r = \frac{\partial L}{\partial W_r^T} = W_l(1_i 1_l^T)$ and $\tilde{g}_r = \frac{\partial \tilde{L}}{\partial W_r^T} = W_c W_l(1_i 1_l^T) = W_c g_r$. It is obvious that all elements in each column of $g_r$ are identically distributed. Denoting the expection of each element in $g_r$ as $m$, we have $E[\tilde{g}_r[i, j]] =$

$E[\sum_{k=1,\cdots,K_l} W_c[i,k]g_r[k,j]] = mK_lE[w_c]$, where $E[w_c]$ is the expectation of each parameter in $W_c$. The commonly-used kaiming weight initialization distribution [7] is $U[-\frac{a}{\sqrt{K_l}}, \frac{a}{\sqrt{K_l}}]$, and by taking the absolute value, we know that $w_c \sim U[0, \frac{a}{\sqrt{K_l}}]$, where $a$ is a gain hyperparameter and usually >1. Therefore, the expectation of the absolute weight value is $E[w_c] = \frac{a}{2\sqrt{K_l}}$, thus $E[\tilde{g}_r[i,j]] = mK_lE[w_c] = \frac{a\sqrt{K_l}}{2}E[g_r[i,j]]$. That is to say, as long as $K_l > \frac{4}{a^2} = \frac{2}{3}$ (typical value of $a$ is $\sqrt{6}$), which is always true, the expectation of each gradient element in $\frac{\partial \tilde{L}}{\partial W_r^T}$ increase by a ratio $\frac{a\sqrt{K_l}}{2} > 1$ after adding a $W_c$. And for all parameters $W \in \{W_{n+1}, \cdots, W_N\}$, their gradients are amplified by this ratio according to the chain rule: $\frac{\partial \tilde{L}}{\partial W} = \frac{\partial \tilde{L}}{\partial W_r^T}\frac{\partial f_r}{\partial W}$, where $\frac{\partial f_r}{\partial W}$ is fixed, given $\{W_{n+1}, \cdots, W_N\}$ fixed. The derivation for $\frac{\partial L}{\partial W_l}$ is similar and thus omitted. In summary, the *synflow* indicator prefers architectures with more layers by design.

## C  Results on Non-topological Search Spaces

The search spaces of NB101, NB201 and NB301 are topological search spaces that contain architectural decisions about connection patterns. In practice, non-topological search spaces are also commonly used, especially in hardware-aware NAS [15, 13], since complex architectural connection patterns might deteriorate the efficiency. The architectural decisions in non-topological search spaces usually include hyperparameters of predefined blocks like ResNet or MobileNet blocks, common searchable architectural decisions include depth, width (channel number), kernel size, group number and so on.

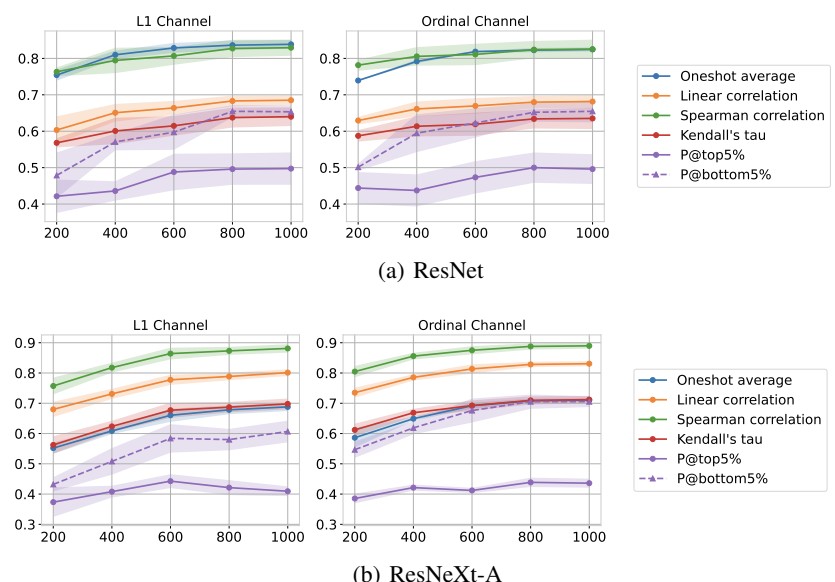

Figure A24: Criteria trend on NDS ResNet and NDS ResNeXt-A.

In order to evaluate OSEs and ZSEs on non-topological search spaces, we use the NDS ResNet and ResNeXt-A benchmarks provided by [11]. The ResNet search space contains depth and width decisions, and the ResNeXt-A search space contains decisions w.r.t depth, width, bottleneck width ratio, and number of groups. We refer the readers to the NDS paper [11], Tab. A1 and our code for the detailed description of these two search spaces.

To conduct OS estimation (i.e., parameter-sharing estimation) of the architecture performances, we should specify how the parameters are shared between the architectures, and which subset of parameters are used when evaluating each architecture. For the depth decision $d \in \{d_1, d_2, \cdots, d_K\}$ in each stage, we initialize a stage with the maximal possible depth $\tilde{d} = \max(\{d_1, d_2, \cdots, d_K\}) = d_K$, and when evaluating an architecture with depth $d$, only the first $d$ blocks are forwarded. As for the decisions about the width and bottleneck width ratio, we init each block with the maximal possible

channel number, and experiment with two strategies of picking channels for each architecture: 1) Picking channels with the largest L1 norms (L1), since using the L1 norm as the channel importance criterion is common in the pruning literature; 2) Picking the first several channels (Ordinal). As for the group number decision $g \in \{g_1, g_2, \cdots, g_K\}$, the convolution layer is initialized with $\tilde{g} = GCD(g_1, g_2, \cdots, g_K)$ groups, which is the greatest common divisor (GCD) of all possible group number choices. And when the actual group number of a convolution is $g$, we need to pick out parameters that correspond to $g/\tilde{g}$ subgroups in each original group. More specifically, for the $i$-th ($i \in \{1, 2, \cdots, C^{out}\}$) convolutional kernel with maximal input channel number $C^{in}$, we choose the convolution parameters corresponding to the $(\lfloor \frac{ig}{C^{out}} \rfloor \mod \tilde{g})$-th subpart of the input channels (each subpart has $\tilde{g}/gC^{in}$ input channels). Note that when group search is needed for a convolution, the channel search for this convolution should use the ordinal channel picking rule, since using the L1 rule is no longer reasonable. Thus in the ResNeXt-A search space, all convolutions that need group number and width search use ordinal channel picking rule, but for convolutions that only need width search, we still experiment with both the L1 and Ordinal picking rules.

Table A12: Comparison of ZSEs' and OSEs' KD / SpearmanR / P@top 5% / P@bottom 5% / BR@0.5% on NDS ResNet and ResNeXt-A. OS accuracy is used as the OS score.

| ZSE | ResNet | ResNeXt-A |
|---|---|---|
| *synflow* | 0.231 / 0.346 / 0.004 / 0.448 / 0.142 | 0.690 / 0.871 / 0.300 / **0.848** / **0.003** |
| *grad_norm* | 0.237 / 0.358 / 0.000 / 0.324 / 0.202 | 0.319 / 0.464 / 0.104 / 0.336 / 0.020 |
| *grasp* | -0.114 / -0.171 / 0.076 / 0.012 / 0.003 | -0.262 / -0.379 / 0.068 / 0.000 / 0.007 |
| *plain* | 0.307 / 0.451 / 0.048 / 0.240 / 0.059 | 0.289 / 0.418 / 0.052 / 0.176 / 0.053 |
| *jacob_cov* | -0.072 / -0.114 / 0.000 / 0.020 / 0.267 | 0.051 / 0.077 / 0.032 / 0.076 / 0.125 |
| *relu_logdet* | 0.182 / 0.273 / 0.008 / 0.352 / 0.133 | 0.459 / 0.645 / 0.088 / 0.560 / 0.019 |
| *Param* | 0.334 / 0.472 / 0.032 / 0.424 / 0.172 | 0.479 / 0.663 / 0.080 / 0.568 / 0.075 |
| *FLOPs* | 0.600 / 0.781 / 0.220 / **0.680** / 0.051 | 0.668 / 0.848 / 0.184 / 0.744 / 0.018 |
| *OS (1k epoch)* | **0.635** / **0.825** / **0.496** / 0.655 / **0.003** | **0.712** / **0.890** / **0.436** / 0.705 / 0.006 |

As the OSE training goes on, the evolving trends of different criteria are shown in Fig. A24. We can see that *the L1 channel picking rule does no help, and using the ordinal channel picking rule for width search can achieve slightly better ranking quality*. The comparison of OSEs and ZSEs is summarized in Tab. A12, where the channel picking rule of OSEs is ordinal. We can see that *similar with the results on topological search spaces, OSEs can provide consistently better estimations than current ZSEs, and the relative effectiveness of different ZSEs vary*. For example, *plain* performs best among all ZSEs on ResNet, while *synflow* performs best among all ZSEs on ResNeXt-A, and *relu_logdet* performs best among all ZSEs on topological search spaces such as NB101, NB201 and NB301.

## D   Training and Evaluation Settings

**Training Settings** We summarize the hyper-parameters used to train all the supernets in Tab. A13. Specifically, we train the supernets via momentum SGD with momentum 0.9 and weight decay 0.0005. The batch size is set to 256 on NB101 / NB301, 512 on NB201, and 64 on ResNet / ResNeXt-A. The initial learning rate is set to 0.05 for NB201 / NB301 / ResNet / ResNeXt-A, and 0.025 for NB101. And the learning rate is decayed by 0.5 each time the supernet's average training loss stops to decrease for 30 epochs. During training, the dropout rate before the fully connected classifier is set to 0.1, and the gradient norm is clipped to be less than 5.0. All the training and evaluation are conducted on CIFAR-10, which is the dataset used by the three benchmarks. 80% of the training set are used as the training data, while the other 20% are used as the validation data. We run every supernet training process with three random seeds (20, 2020, 202020).

**Evaluation Settings** On NB201, we use all the 15625 architectures (6466 non-isomorphic ones) to evaluate the ranking quality of OSEs and ZSEs. On NB101, the supernet is trained by random sampling from the 14580 architectures (without loose end) in search space 3 of NB101-1shot, and the OSE quality is evaluated using 5000 architectures randomly sampled from the 14580 architectures. NB301 provides the tabular performances of 59328 anchor architectures, and we randomly sample 5896 architectures from these architectures with tabular performances for ranking quality estimation.

Table A13: Supernet training hyper-parameters

| optimizer | SGD | initial LR | 0.025 (NB101)
0.05 (NB201/NB301/ResNet/ResNeXt) |
|---|---|---|---|
| momentum | 0.9 | LR schedule | ReduceLROnPlateau[†] |
| weight decay | 0.0005 | LR decay | 0.5 |
| batch size | 256 (NB101/NB301)
512 (NB201)
64 (ResNet/ResNeXt) | LR patience | 30 |
| dropout rate | 0.1 | grad norm clip | 5.0 |

†: See `https://pytorch.org/docs/stable/optim.html#torch.optim.lr_scheduler.ReduceLROnPlateau`.

As for NDS ResNet, we use the 5000 architectures that have 3 GT performances with different training seeds. And for NDS ResNeXt-A, we choose 5000 architectures from the 25k architectures with GT performances for ranking quality estimation.

For evaluating ZSEs on all three topological search spaces and the NDS ResNet search space, we use a batch size of 128. For evaluating ZSEs on the NDS ResNeXt-A search space, we use a batch size of 64, which enables us to run each experiment on only one GPU. We evaluate the ZSEs with 5 batches in total. We find that, as reported by previous zero-shot studies, the variance of ZS estimations across different data batches is very small. And utilizing more data batches does not increase the ranking quality of ZSEs. And the ZSE results reported in our paper are all calculated using the average score of 5 validation batches.

**Resources** We run the experiments on 16 NVIDIA Geforce RTX 2080Ti GPUs, and every experiment is run on only one GPU. By rough estimation, all the training and evaluation experiments take about 300 GPU days.

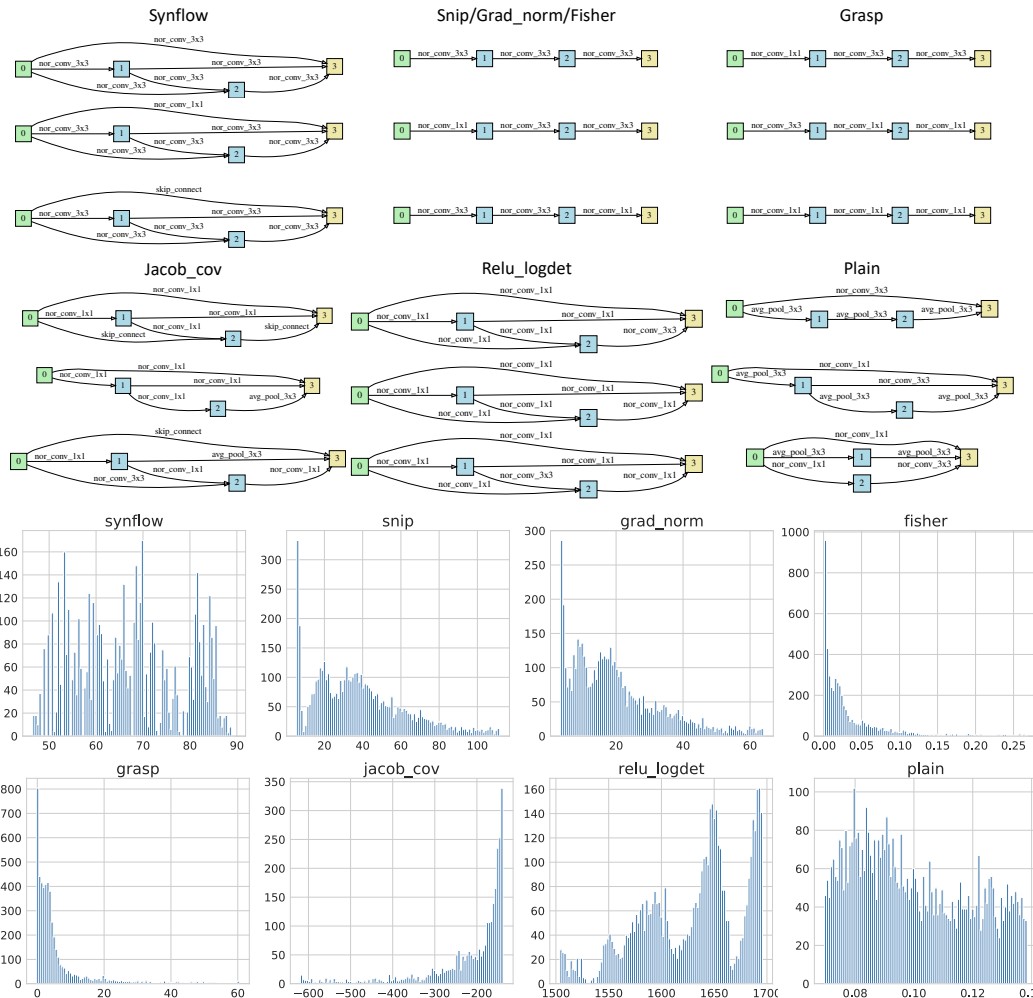

Figure A25: Top: Top-3 zero-shot ranked architectures on NB201. Bottom: Histogram of zero-shot scores. For *synflow*, we plot the histogram of the log values of *synflow* scores (varying from 0 to $7.55 \times 10^{49}$). Since there are many outliers in all types of zero-shot scores, only the histogram of the zero-shot scores between the 0.1 and 0.9 quantile are plotted.

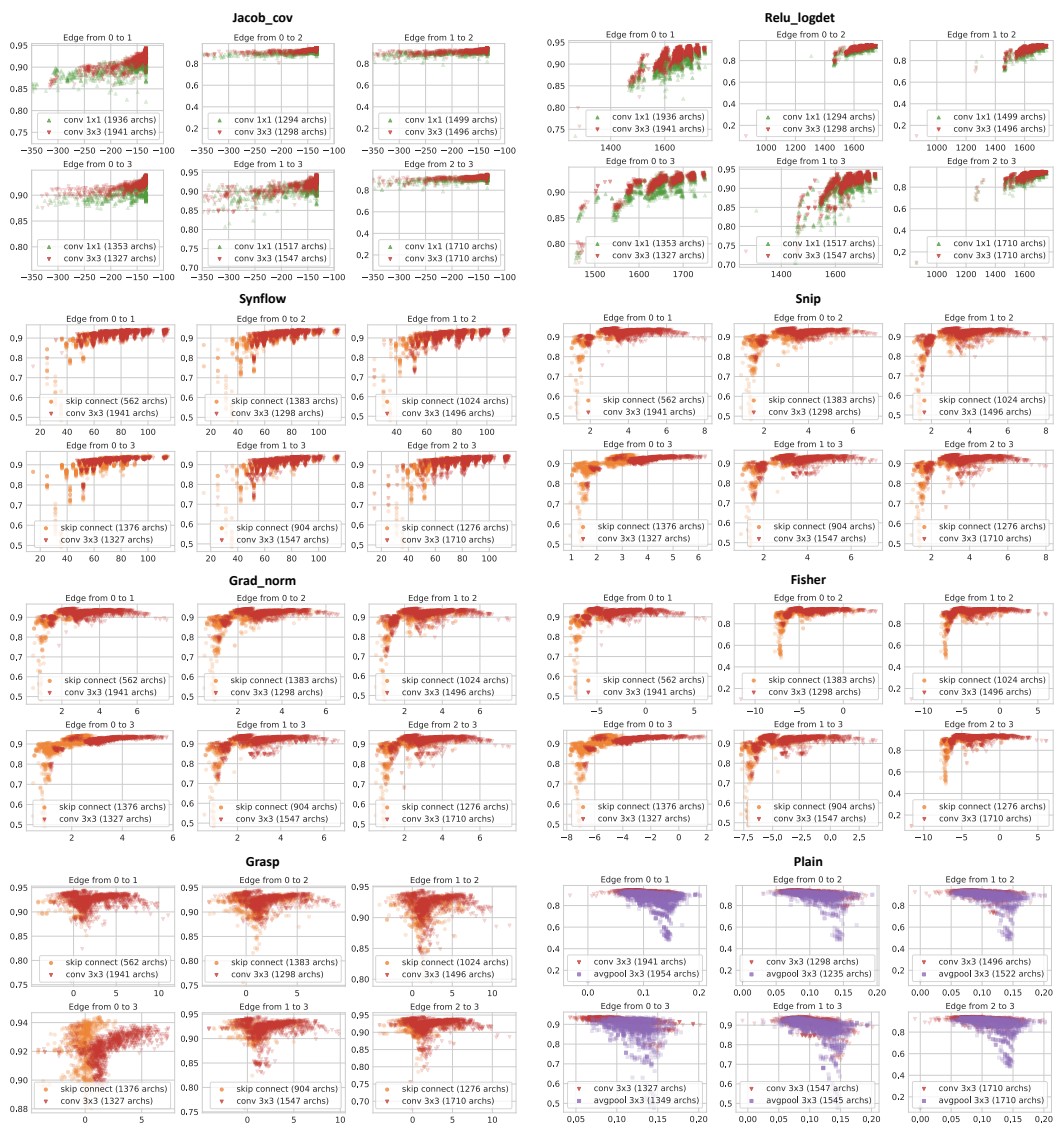

Figure A26: The scatter plot of GT (Y axis) - ZS (X-axis) scores, different subplots stand for different edges (6 edges in total), and different colors & markers stand for different operation type on that edge. Logarithm are taken for *synflow/grad_norm/fisher* values.

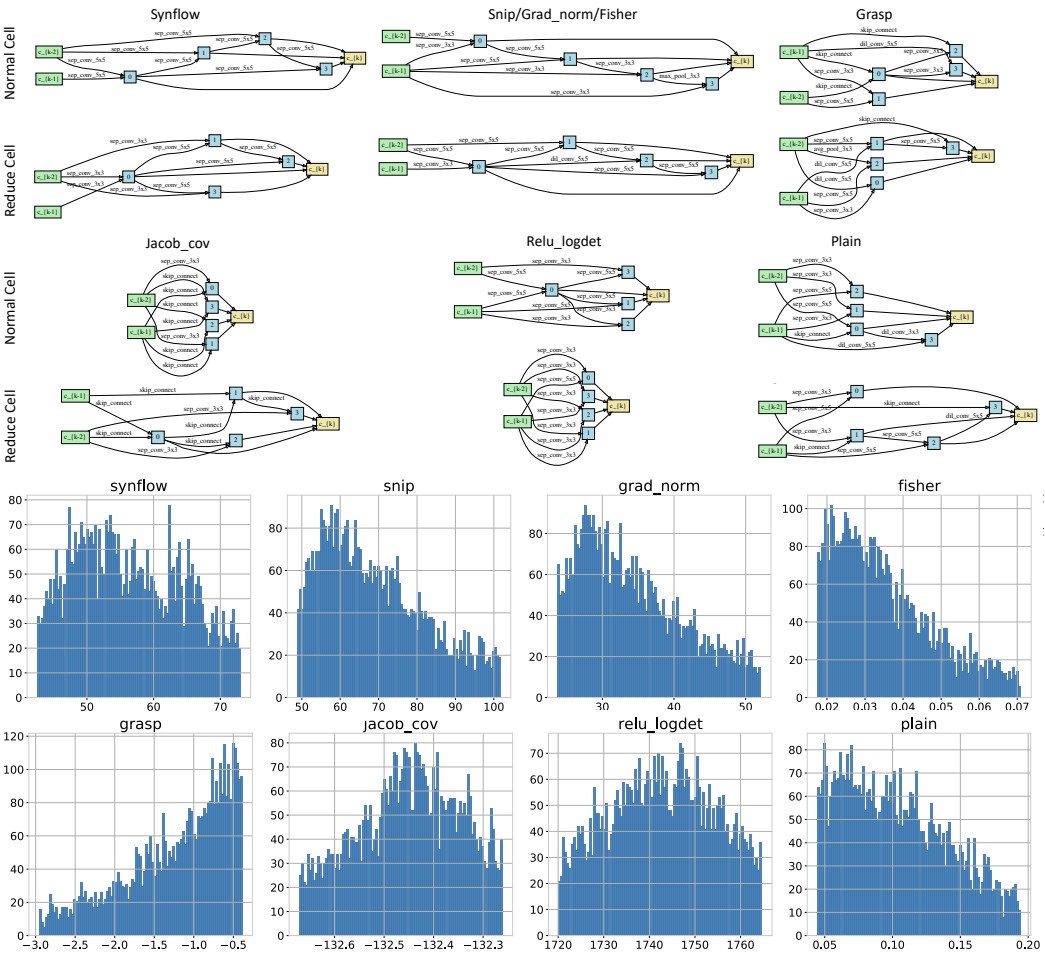

Figure A27: Top: Top-1 zero-shot ranked architectures (normal & reduction cell) on NB301. Bottom: Histogram of zero-shot scores. For *synflow*, we plot the histogram of the log values of *synflow* scores (varying from $1.18 \times 10^{10}$ to $1.18 \times 10^{54}$). Since there are many outliers in all types of zero-shot scores, only the histogram of the zero-shot scores between the 0.1 and 0.9 quantile are plotted.

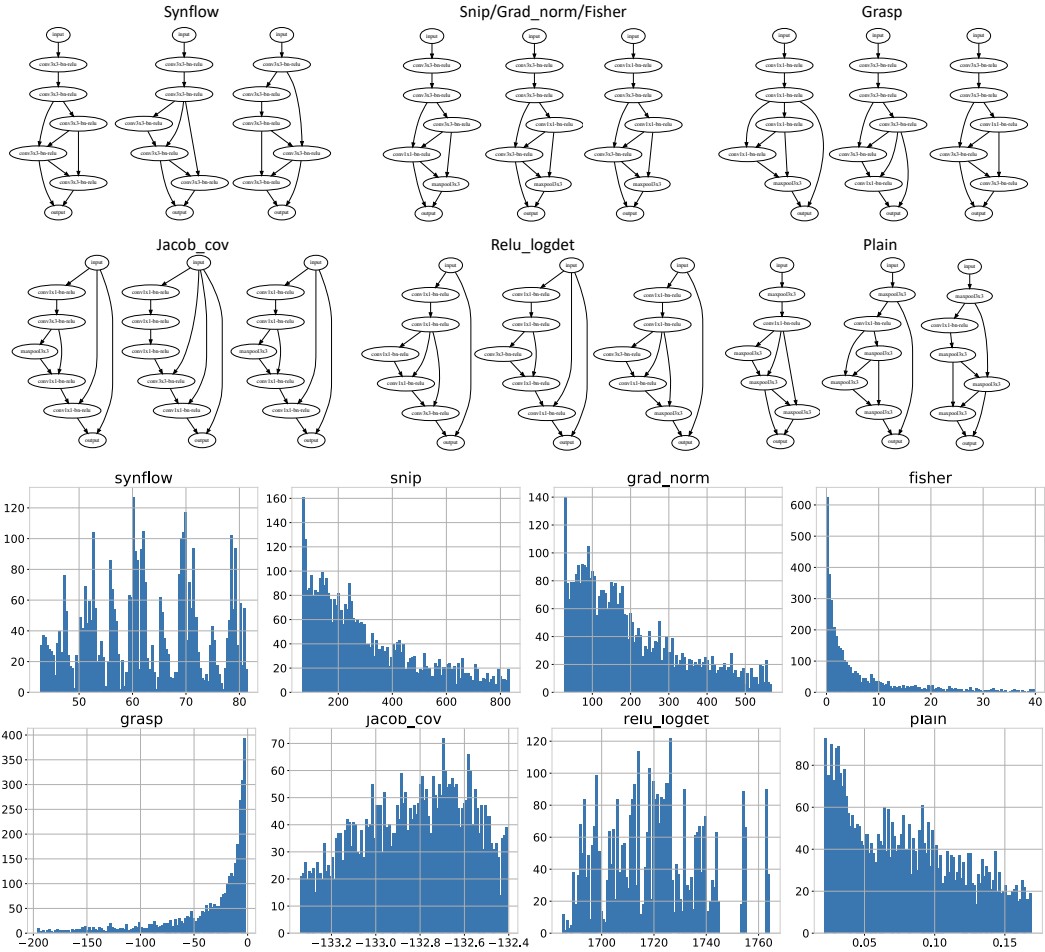

Figure A28: Top: Top-3 zero-shot ranked architectures on NB101. Bottom: Histogram of zero-shot scores. For *synflow*, we plot the histogram of the log values of *synflow* scores (varying from $5.31 \times 10^9$ to $2.06 \times 10^{47}$). Since there are many outliers in all types of zero-shot scores, only the histogram of the zero-shot scores between the 0.1 and 0.9 quantile are plotted.

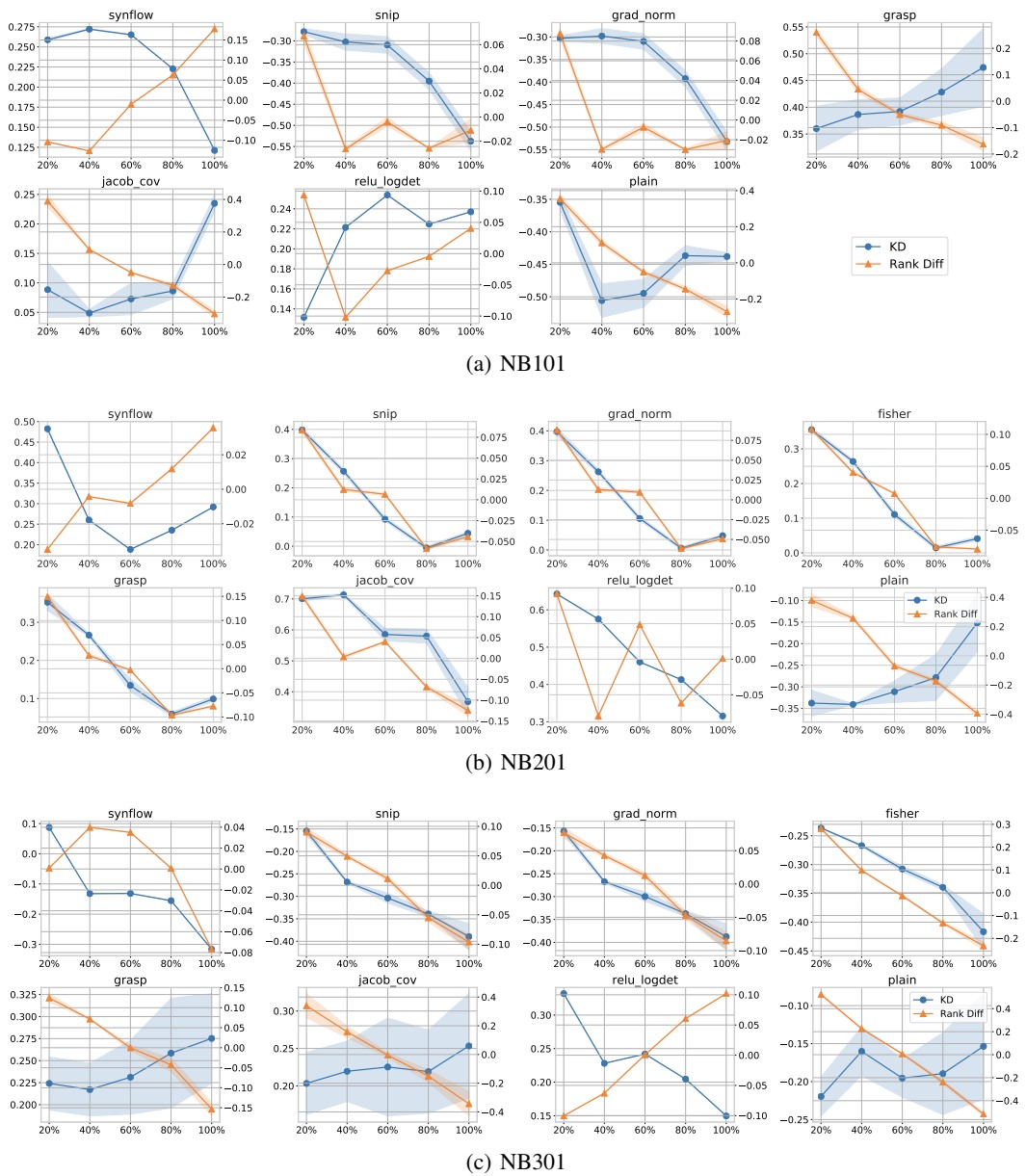

Figure A29: Complexity-level bias of zero-shot estimators on NB101 (a), NB201 (b) and NB301 (c). The architectures are grouped by their parameter sizes (Param).