# OpenReview forum: "Evaluating Efficient Performance Estimators of Neural Architectures"
_NeurIPS.cc/2021/Conference — NeurIPS 2021 Poster_

### Official Review · Reviewer_SU6e · 2021-06-26

**Rating:** 7
**Confidence:** 3

**Summary:**

Neural Architecture Search aims to identify neural network architectures from a human-designed search space which have good accuracies, or good accuracy/size tradeoffs. The submission explores how a number of different design choices affect how well existing heuristics can identify/rank promising architectures from a search space. The submission focuses on two classes of heuristics:
* One-shot models/estimators, which use a single shared set of weights to evaluate/rank many different candidate architectures,
* Zero-shot estimators, which try to evaluate/rank different candidate architectures without any training at all.

Experiments are conducted on the NASBench-101, NASBench-201, and NASBench-301 benchmark tasks.

**Limitations And Societal Impact:**

In order to avoid training/evaluating many network architectures from scratch, experiments are conducted only on NASBench-101/201/301. On its own, each of these benchmark tasks has significant limitations (listed below). However, since the different benchmark tasks have different strengths and weaknesses, I think reporting results on all three (as the submission does) should be sufficient for a NeurIPS paper.
* NASBench-101 and NASBench-201 are -- by design -- very small search spaces (e.g., there are around 423,000 unique architectures for the full NASBench-101 dataset, and much less for NASBench-201), which could skew the conclusions.
* NASBench-301 uses a much larger search space containing 10^18 architectures. However, so-called "ground-truth accuracies" from this dataset actually come from a surrogate regression model. This introduces a different type of skew, since we can penalize a model whose performance is underestimated by the surrogate (i.e., has low predicted performance according to the surrogate but would perform better if we were to actually train/evaluate it).

**Main Review:**

# Overview
The submission conducts a variety of experiments on NASBench-101/201/301, with the goal of understanding how different design choices affect the ability of one-shot models and zero-shot estimators to rank different candidate architectures in a NAS search space.

The submission is an analysis paper, and does not claim to propose a new method. It feels like a large collection of results related to one-shot / zero-shot architecture search, which may be useful for practitioners.

On the positive side, the submission presents some interesting and surprising findings, and provides plausible explanations. For example, in Section 4.1, the authors observe a counterintuitive effect where evaluating on more examples leads to lower one-shot model accuracies, and provide a plausible explanation (basically: this may be a quirk in the way their metrics are defined; evaluating on fewer examples increases the likelihood of ties, or cases where two architectures will have exactly the same estimated accuracy).

On the negative side, I frequently had the feeling that the submission was trying to pack too much information into too little space, with some unfortunate consequences. For example:
* I didn't see details about how one-shot models were trained in the main paper. Some additional information was provided in a zip file in the supplementary material, but unless I missed something, this seems like a surprising and unfortunate omission.
* In a few plots, the legends occlude/cover up relevant data points (Figures 3, 8, and 10).
* A few plots have a large number of overlapping lines, which made them difficult for me to interpret (e.g., Figure 1, Figure 8, Figure 10, Figure 11).
* Textual descriptions of plots and results are sometimes very short and a bit hard to follow, perhaps because of space constraints. I'm particularly confused about how to interpret Figure 3 (right).



Highlights of experimental results for one-shot models:
* Effect of training time on one-shot model performance for different NABench-101/201 (Section 4.1 / Figure 1) and NABench-301 (Section 4.1 / Figure 3).
* Effect of increasing the number of evaluation examples used to estimate accuracy when using a one-shot model to evaluate a candidate architecture.
* Analysis of one-shot models on different slices of the search space (e.g., ability of a one-shot model to rank large vs small architectures; ability of a one-shot model to identify the top vs. bottom 5% of network architectures, bias towards low-FLOPs vs. high-FLOPs architectures).
* Some experiments related to ensembling multiple one-shot models (Section 5.3 / 6.1).
* Experiments with sampling heuristics proposed by the FairNAS paper (Section 6.2), although I'm not sure exactly which sampling heuristic the submission was referring to.
* Experiments on how restricting the search space affect one-shot model quality (Section 6.3).

Highlights of experimental results for zero-shot estimators:
* Section 4.2 evaluates five different (previously proposed) zero-shot estimators on NASBench-201/301: "plain, snip, grasp, synflow, and fisher".
* The paper's main results for zero-shot estimators seem to come from Table 2, which measures how strongly correlated the zero-shot estimators' outputs are with each other, as well as with other metrics such as "ground-truth" accuracies, FLOPs, and parameter counts, and one-shot estimators.
* The submission also touches on experiments evaluated zero-shot estimators for trained vs. untrained networks, although data is only provided in an appendix.
* Some discussion/analysis of biases of zero-shot estimators (e.g., favoring architectures with more parameters, or which do not have skip-connections).


# Detailed Comments
L94:  "However, the correlation [reported by Zhang et al. [32]] is evaluated using 100 architectures randomly sampled from a large search space, which is not a convincing and consistent benchmark metric."

It would be helpful if the authors could provide more details to support their assertion that this approach is not "convincing and consistent," since this isn't obvious to me. Or else rephrase their claim. I agree that the submission's analysis is complementary to Zhang et al.'s. But it seems like using benchmark datasets like NASBench-101/201/301 (which the submission focuses on) comes with a different set of pros and cons (e.g., artificially small search space size in NASBench-101/201; model quality is estimated using a surrogate predictor instead of a ground-truth measurement in NASBench-301).

L107: "Mellor et al propose ta indicator based on [...]"
What is "ta indicator"? Is this a typo?

Does the submission provide details about the training setup used for Section 4?

Figure 1 ("Criteria of using OS accuracy/loss as estimations"):
* It's not clear to me what "Oneshot Average" refers to. I didn't notice any reference to it in the "Evaluation Criteria" section.
* It's not clear to me what learning rate schedule is used. I think this needs to be documented, since it can potentially affect the paper's conclusions. For example: if a model is trained with cosine or linear decay, we may need to adjust the learning rate schedule if the number of training epochs is reduced.

L165: "Another fact is that one-shot (OS) estimations are better at distinguish bad architecture (higher P@bottom5%) than distinguishing good ones (lower P@top5%)."

Looking at Figure 1, it seems like this is true for NASBench-201 and NASBench-301 but just the opposite is true for NASBench-101 (1shot sub 3)?

L182: I initially had trouble parsing the following text: "For example, as shown by NB301 results in Fig. 3, criteria get better when the batch number increases from 1 to 10 at epoch 1000." My current interpretation is that the authors are computing average model accuracies over N batches, where each batch contains 128 examples. But if so, it would be helpful to update the phrasing to clarify what the "batch number" is.

Layout of Figure 3 should be adjusted, since the plot legend currently obscures some of the results.

I don't understand the "Intra-Level KD" plot in Figure 3.

L322: "As expected, averaging the OS accuracy of multiple supernets stabilizes OSE [one-shot] estimations."
I had trouble understanding figuring out which data supports this claim. Figure 8 (b) - middle shows relP@top/bottom 5% for single models vs. ensembles, and these results seem consistent with the authors' claim. But for some reason, relative Kendall's tau only seems to be reported for single models in Figure 8 (b) - left.

L367: "Sec. 5.3 shows that averaging OS scores of several supernets stabilizes the estimations, but this is not practical due to the linearly enlarged consumption."

It's not clear to me what the authors mean by "linearly enlarged consumption", or why the authors believe that this approach is not practical. (If the authors' argument is that temporal ensembling has lower training cost than training separate models, the claim seems reasonable. But if this is the case, I'd suggest updating the phrasing to make the meaning clearer.)

Figure 10:
* The plot in the lower right-hand corner is partially obscured by the legend.
* Temporal ensembling clearly outperforms single-model estimation at the start of training. But it's not clear to me from the figure whether it's actually any better at the end of training (after 1k epochs). The blue ("single") and green ("temporal ensemble 5 ckpts") lines seem quite close together at the end of training.

L374: "We also adapt Fair-NAS sampling strategy to the NB201/301 spaces (a special case of MC sampling 5/7 for 201/301)."

I'm a bit unsure of the terminology ("sampling 5/7") and it's not clear to me whether "Fair-NAS sampling strategy" refers to the paper's notion of "Strict Fairness" or something else. It would be helpful to clarify this.

I found Figure 11 (top left) difficult to read because the "post-deiso" [post-deisomorphic sampling] and "no post-deiso" lines are basically on top of each other, it looks like there's just a single line in the plot (with a symbol that doesn't appear in the legend). I'm a bit unsure of whether this supports the submission's claim that "Post-deiso achieves slight improvements over "no post-deiso".

## Post-Rebuttal Update
I've updated my review score from 6 to 7. Please see my response below for a detailed rationale.

**Time Spent Reviewing:**

7

---

> ### Author Response · Authors · 2021-08-09
> **Response to the reviewer**
>
> Thanks for your careful and valuable suggestions. We sincerely hope that our responses can address your concerns:
>
> Q1: Details about how one-shot models were trained in the main paper.
>
> A1: Thanks for the suggestion. Currently, we provide the training configurations in the appendix. We'll add more background on oneshot training in Sec.2, and add some clarification of our experiment settings in Sec.4.
>
> Q2: In a few plots, the legends occlude/cover up relevant data points (Figures 3, 8, and 10).
>
> A2: Thanks for the suggestion, we'll modify all these plots to make their legend in better positions.
>
> Q3: A few plots have a large number of overlapping lines, which made them difficult for me to interpret (e.g., Figure 1, Figure 8, Figure 10, Figure 11).
>
> A3: Thanks for the suggestion, we'll scale up Fig.1, Fig.11. The lines in Fig.10 are indeed very close, since the improvements of the ensemble techniques are not very large, we can delete the red line of temporal ensembling 3 checkpoints to make it more clear.
>
> Q4: I'm particularly confused about how to interpret Figure 3 (right).
>
> A4: Thanks for pointing out this obscure figure. The experiment in Fig.3 demonstrates a point that if the training is not sufficient, using more data might result in worse ranking quality, especially when using the OS acc as the score. We analyze that this is because in the early training stages, the ability of the supernet to distinguish between "close architectures" is not good, thus giving many tie scores can be better than giving high-resolution scores. "Close architectures" refer to architectures that have the same OS accuracy when only using 1 validation batch, i.e., in the same "level". Since 1 dataM batch only contains 128 images, thus at most 128 different OS acc levels can exist. The legend in Fig.3(Right) shows the actual number of levels, and the figure shows the histogram of intra-level KDs. For example, when seed=20 at epoch 200, there are 74 levels that have more than one architecture, and we plot the histogram of the 74 intra-level KDs as the blue bars in the figure.
>
> We appreciate all these suggestions on the formatting and missing descriptions. Indeed, we omitted some important descriptions due to the space constraints, and we'll do some reorganization: put 1-2 figures and Eq.(1) to appendix and refs them in the main paper. Then we'll add all necessary discussions in the main paper.
>
> ------
>
> Q5: L94: It would be helpful if the authors could provide more details to support their assertion that this approach is not "convincing and consistent," Or else rephrase their claim.
>
> A5: Thanks. We'll rephrase this haste claim to: "[32] compare the correlation of OSEs and their proposed hyper-network based estimator. However, their work is not aiming for a large-scale evaluation of OSEs and ZSEs, thus they only evaluate the OSE correlations on one search space, and do not conduct further analysis."
>
> Q6: L107: " What is "ta indicator"? Is this a typo?
>
> A6: Yes, this is a typo. We'll fix it to "an indicator".
>
> Q7: Does the submission provide details about the training setup used for Sec. 4?
>
> A7: We provide the training and evaluation setup in Appendix C. We'll refer to it in the main paper.
>
> Q8: Figure 1: It's not clear to me what "Oneshot Average" refers to and what learning rate schedule is used.
>
> A8: "Oneshot Average" means the average oneshot score (accuracy or loss). We'll add this to the evaluation criteria section. Currently, we provide all training/evaluation setup in Appendix C. And we use the plateau learning rate schedule, it only decays the learning rate by 0.5 when the training loss stops to decrease for a while. If a pre-defined fixed schedule (e.g., cosine with a predefined max epoch number; step LR decay) is used, one would need to tune the epoch number or the steps for a more sufficient training process. In that way, the evaluation costs of our work would be extremely large, since for each max epoch setting, one needs to conduct a separate training process.
>
> Q9: L165: "Another fact is that one-shot (OS) estimations are better at distinguishing bad architecture (higher P@bottom5%) than distinguishing good ones (lower P@top5%)." Looking at Figure 1, it seems like this is true for NASBench-201 and NASBench-301 but just the opposite is true for NASBench-101 (1shot sub 3)?
>
> A9: Yes, you're right. We've missed this phenomenon in our submission. Thanks for pointing out this. To see whether NB101 is a special SS, we further conduct experiments on NDS ResNet/ResNeXt-A, which are two non-topological search spaces. And the phenomenon of P@bottom5%>P@top5% still exists. Therefore, based on our experiment, we think NB101 is a special case, and we'll explicitly discuss this in our paper: "The phenomena on NB101 are different from those on NB201 and NB301: P@top 5% is higher than P@bottom 5%, and longer training after the first 20 epochs does not improve the ranking quality. The distinct phenomenon might arise from two aspects: 1) As shown in the figure (we'll add a figure on the GT accuracy distribution in the revision), the GT distribution of the top GT accuracies on NB101 is less concentrated than that on NB201 and NB301, thus it is easier to distinguish the top architectures on NB101. 2) Different from NB201 and NB301, NB101 is an operation-on-node search space. The sharing extent of supernet on operation-on-node search spaces is larger than that on operation-on-edge search spaces, since for each node, no matter which input connections are chosen, the same parameters are used. A larger sharing extent would limit the potential ranking quality of OSEs." We'd appreciate more suggestions on our analysis here.
>
> Q10: L182: Update the phrasing to clarify what the "batch number" is.
>
> A10: Yes, your interpretation is right. We'll update the phrase to "We compute the average OS accuracies over N validation batches, where each batch contains 128 examples. And the effect of the batch number N on the ranking quality is shown in ...".
>
> Q11: Layout of Figure 3 should be adjusted, since the plot legend currently obscures some of the results.
>
> A11: Thanks. We'll adjust the layout.
>
> Q12: I don't understand the "Intra-Level KD" plot in Figure 3.
>
> A12: We've discussed this in the above reply to the "lacking of description for Figure.3" issue, please check that discussion. Sure, we will clarify this point in our revision.
>
> Q13: L322: "As expected, averaging the OS accuracy of multiple supernets stabilizes OSE estimations." I had trouble understanding figuring out which data supports this claim. Figure8(b) - middle shows relP@top/bottom 5% for single models vs. ensembles, and these results seem consistent with the authors' claim. But for some reason, relative Kendall's tau only seems to be reported for single models in Figure 8(b)left.
>
> A13: Fig8(b)Left shows the ranking stability of using a single supernet by presenting multiple criteria. And the middle/left figures show the comparison on relP@topKs and relP@bottomKs for two different K (5%, 0.5%). Actually, the comparison on relative KD is similar, we omit it due to the space constraint, and we think it is better for us to change the P@top/bot0.5% figure to a relative KD comparison figure. Thanks for the comment.
>
> Q14: L367: It's not clear to me what the authors mean by "linearly enlarged consumption", or why the authors believe that this approach is not practical.
>
> A14: Yes, you're right. We've omitted this logic. We said "linearly enlarged consumption", since training k supernets takes k-times more computation. And since training one supernet and temporal ensembling different checkpoints in the weight space have lower training costs than training multiple supernets and ensemble their scores, we think temporal ensembling is a more practical method. We'll make this point clearer.
>
> Q15: Figure 10: The plot is partially obscured by the legend. Temporal ensembling clearly outperforms single-model estimation at the start of training. But it's not clear to me from the figure whether it's actually any better at the end of training. The blue ("single") and green ("temporal ensemble 5 ckpts") lines seem quite close together at the end of training.
>
> A15: We'll modify the legend in Fig.10. Actually, temporal ensembling does not bring obvious ranking quality improvements on NB301. Our current statement is "Temporally ensembling 3 or 5 checkpoints brings improvements on NB201, but brings no bias improvements on NB301." But this technique can bring improvements in ranking stability (Fig.8) at a very low cost, thus we encourage this technique to be used.
>
> Q16: L374: I'm a bit unsure of the terminology ("sampling 5/7") and it's not clear to me whether "Fair-NAS sampling strategy" refers to the paper's notion of "Strict Fairness" or something else. It would be helpful to clarify this.
>
> A16: Thanks for the comment. We'll add more clarification (a sampling pseudo-code) on how we adapt the Fair-NAS sampling strategy to topological SSes in the appendix. In short, following the FairNAS's methodology, we sample n_primitive arch samples in each supernet training step (n_primitive=5 on NB201 and n_primitive=7 on NB301), which ensures that each operation primitive at each position is used exactly once in each update.
>
> Q17: I found Figure 11 (top left) difficult to read because the "post-deiso" and "no post-deiso" lines are basically on top of each other, it looks like there's just a single line in the plot. I'm a bit unsure of whether this supports the submission's claim that "Post-deiso achieves slight improvements over "no post-deiso"."
>
> A17: Your interpretation is right: Using post-deiso evaluation does not improve the KD. However, post-deiso does bring improvements on BR@Ks and P@topKs (left lower/right upper plots). We'll make this point clearer in the texts.

---

> > ### Comment · Reviewer_SU6e · 2021-08-18
> > **Thank you for your response**
> >
> > Thanks for your feedback. I plan to increase my score from 6 to 7 for the following reasons:
> >
> > 1) Based on the authors' response, I'm optimistic that the presentation/clarity issues I mentioned in my original review will be addressed in the final version, and that relevant details about the experimental setup will be released publicly.
> > 2) Based on the discussion of ZSE and OSE implementations in this review and others, I have more confidence that the evaluated methods were implemented carefully and correctly, which is critical for an evaluation paper.
> > 3) I agree with the other reviewers that an extensive evaluation of ZSE and OSE methods is a valuable contribution to the research community. (This is something that I might not have weighed heavily enough in my initial review.)

---

> > > ### Author Response · Authors · 2021-08-18
> > > **Thanks for your assessment**
> > >
> > > Thanks for your suggestions and assessment, they really help us to improve our paper. We'll follow the suggestions to prepare the revision.

---

### Official Review · Reviewer_ekzD · 2021-07-15

**Rating:** 8
**Confidence:** 5

**Summary:**

This work analyzes several one-shot and zero-shot NAS methods on multiple NAS benchmark datasets. The authors inspect NAS algorithms from various aspects, including training bias, stability, variance, loss ranking v.s. accuracy ranking, etc. . The authors conclude that one-shot methods and zero-shot methods might fail to work due to multiple reasons and propose several solutions to alleviate these issues.

**Limitations And Societal Impact:**

No foreseeable risk of negative societal impact.

**Main Review:**

[Strength]

The paper is well-written and very easy to follow. The authors did an excellent job in analyzing one-shot and zero-shot NAS methods, with deep insight and comprehensive ablation studies. These results provide a clear picture of when and why one method works or fails under a specific setting. I believe this study will inspire more future works in one-shot and zero-shot NAS works.

There are many empirical studies showing that one-shot NAS is unstable. However, I am still impressed by the analysis presented in this work. The authors give a much clear picture of how these NAS methods fail to work systematically. Especially, the authors analyze the behavior of NAS methods by grouping them in different size, FLOPs and accuracy, and report their performance by various indices. While some characteristics of one-shot/zero-shot NAS methods have been analyzed in previous works, a majority of them are revealed for the first time.

[Weakness]

It is hard to point out obvious weakness of this work. The following are suggestions for potential improvements:

1) While this work focuses on one-shot NAS more than zero-shot NAS, it is more interesting to investigate why zero-shot NAS fail to match the performance of one-shot NAS and how to fix these issues in zero-shot NAS. Is it possible that zero-shot NAS can outperform one-shot NAS under a particular setting?

2) Most experiments in this work are conducted via passive sampling (uniform sampling or random sampling on existing architectures). In recent zero-shot NAS methods, the algorithm can actively sample structures. This will make the two distributions considerably different. For example, a zero-shot method might work well in high-accuracy regime but poorly in low-accuracy regime. Since networks of high accuracy are difficult to sample via uniform sampling, the evaluation of zero-shot NAS in this work might be incomplete.

3) No validation on ImageNet. It is not clear whether the conclusions of this work still hold on ImageNet or other large-scale datasets.


**Time Spent Reviewing:**

3

---

> ### Author Response · Authors · 2021-08-09
> **Responses to the reviewer**
>
> Thanks for your positive recognition and valuable suggestions for our work. Our replies are as follows:
>
> Q1: While this work focuses on one-shot NAS more than zero-shot NAS, it is more interesting to investigate why zero-shot NAS fails to match the performance of one-shot NAS and how to fix these issues in zero-shot NAS. Is it possible that zero-shot NAS can outperform one-shot NAS under a particular setting?
>
> A1: Many thanks for the suggestions. To make our work more valuable for the zero-shot development, we conduct some further analysis. First, we add an experiment for a new ZSE relu_logdet[1], which is a reasonable and good method in our opinion. It can achieve a similar KD as OSE on NB301, but on other SSes, it is still a lot worse than OSE. And its architecture-level bias is also prominent: prefer Conv1x1 over Conv3x3, which leads to its low P@topKs.
>
> Second, we find that the rankings provided by the ZSEs do not change much when we change the data distribution, even when the input distribution changes to uniform/gaussian: On NB201, the KDs between relu_logdet scores on CIFAR10 and CIFAR100/Image-Net-16-120/Uniform-distribution/Gaussian-distribution are 0.98/0.98/0.92/0.88, respectively. On NB301, the KDs between relu_logdet on CIFAR10 and Uniform-distribution/Gaussian-distribution are 0.93/0.92, respectively.
>
> As for the improvement directions, we have several thoughts to share: 1) Architecture-level analysis is necessary, instead of parameter-level analysis; 2) How to make ZSE utilize the input data information; 3) Based on some prominent bias of existing ZSEs, we can add some structural knowledge into ZSE voting ensembles (e.g., receptive field analysis seems promising for improving jacob_cov or relu_logdet).
>
> [1] Joseph Mellor, Jack Turner, Amos Storkey, and Elliot J. Crowley. Neural architecture search without training. In International Conference on Machine Learning (ICML), 2021.
>
> Q2: Most experiments in this work are conducted via passive sampling (uniform sampling or random sampling on existing architectures). In recent zero-shot NAS methods, the algorithm can actively sample structures. This will make the two distributions considerably different. For example, a zero-shot method might work well in high-accuracy regime but poorly in low-accuracy regime. Since networks of high accuracy are difficult to sample via uniform sampling, the evaluation of zero-shot NAS in this work might be incomplete.
>
> A2: We agree with your points. Currently, our paper focuses on passive sampling (or one-shot supernet training, in our terminology), since we think it is a cleaner case for ranking quality evaluation without complex hyperparameter-tuning of the active sampling strategy. Our only usage of active sampling is in Appendix A.3.4, where we train OSEs with evolutionary-based controllers. Also, we think the criteria BR@topK can reflect the results found by the active sampling strategy to some extent. Nevertheless, we think your suggestion on providing the results of ZSEs via active sampling is meaningful since it directly reflects the NAS results.
>
> Q3: No validation on ImageNet. It is not clear whether the conclusions of this work still hold on ImageNet or other large-scale datasets.
>
> A3: We think it is a very important future direction to conduct controllable and extensive verification on larger datasets. But since most benchmarks do not provide GT on ImageNet (201 only have a down-scaled smaller ImageNet) now, we leave this aspect for future work. Nevertheless, we follow your suggestion to add the OS/ZS evaluation results on NB201 ImageNet-16-120 and CIFAR-100, we think this helps for our completeness. Thanks for your suggestion.

---

> > ### Comment · Reviewer_ekzD · 2021-08-23
> > **Thanks for the informative feedback**
> >
> > The authors effectively address all my concerns in the initial review. I do appreciate other reviewers pointed out several detailed issues in the  manuscript which I was not aware in the initial review. Especially, the current paper is full of all sorts of details and conclusions, as Reviewer SU6e mentioned. It is necessary to re-organized the paper for better readability and fix the ambiguities listed by the other reviewers.
> >
> > To verify the authors' results, **I reproduce the experiments using my own pipeline**. I am fully convinced by the major conclusion of this work. The missing ImageNet validation is a pity but not the fault of this work. I believe this work will be a valuable reference in one-shot and zero-shot NAS.
> >
> > I keep my vote 8 for this work.

---

### Official Review · Reviewer_Q7zt · 2021-07-17

**Rating:** 6
**Confidence:** 4

**Summary:**

This paper gives evaluations of methods to evaluate the performance of a neural network before it is fully trained. This has applications to neural architecture search (NAS). Specifically, the authors study two types: one-shot estimators (OSEs) and zero-shot estimators (ZSE) which have both been popular recently. The authors give a large study of various aspects of these methods. They give several new insights about biases in their performance and ideas to mitigate them. All experiments are conducted on three popular search spaces: nas-bench-1shot1, nas-bench-201, and nas-bench-301.

**Limitations And Societal Impact:**

The authors answered yes to 1.b, but I do not see a discussion of limitations anywhere. Can the authors point it out to me? Also, the authors only discuss the societal impacts in the checklist itself. The NeurIPS call for papers said that “the page limit was increased to ensure that the authors have space to address the checklist questions” so these should have been addressed in the main body.

**Main Review:**

### Strengths
- The authors give some good insights into OSE’s and ZSE’s. The ones I find particularly interesting are
   - Improving OSE performance by de-isomorphic sampling. The sampling routine is not studied much in NAS, and the authors find that sampling by weighting all isomorphisms equally, improves performance.
   - The fact that some of the ZSE’s consistently overestimate larger architectures and underestimate smaller architectures is interesting and can lead to better NAS methods. They also show this is true in OSEs.
   - The analysis for OSEs on overestimating vs underestimating based on nparams flops and ops is also interesting.
- The type of analysis in this work is very beneficial to the NAS community, so this paper can have high impact.

### Weaknesses
- For this type of project, novelty is low. On the other hand, I think that the field of neural architecture search is greatly in need of this type of analysis, so this type of project is more important than yet another novel algorithm.
- There are some papers that also evaluate performance estimators which overlap with some experiments conducted by the authors. These should be added to the related work, with a short discussion of the differences. Since there is some overlap, this reduces the paper’s impact a little.
   - [1], [2], [3], and [4] attempt to run large evaluations of performance estimators. In particular, [2] ran experiments with several zero-cost methods on nas-bench-101, nas-bench-201, and nas-bench-301. [3] has experiments with ZSE’s on nas-bench-201, and other search spaces as well. So the impact for ZSE experiments is reduced. [4] runs evaluations of super-net training, so it would be good to explain the differences.
   - As a side note, [5] also has overlap, since it shows ZSE’s perform worse as the model is trained, but this is concurrent work so it does not count against the authors.
- Some of the insights mentioned by the authors are a bit shallow or need more justification.
   - Line 56, it is expected that ZSEs would perform worse than OSEs, since ZSEs take less time to run.
   - Line 61, it is expected that training for longer would make one-shot estimations better.
   - The authors claim that OSEs are better at ranking worse architectures, but the distribution of architecture accuracies is skewed left, so this is a big confounding factor in that claim.
   - Line 153, "on larger search spaces, the training time should be longer", cannot really be concluded after looking at just 2 or 3 search spaces.
- For this type of project that evaluates popular methods in the field of NAS, citing and describing all of the related work is important. But there are some methods missing.
   - ZSE’s: [6] [7] [8] [9]
   - OSE’s: [10] [11]
- Unfortunately, the authors only provided some of their code, and left a note saying that it was too challenging to anonymize their code. Therefore, this project does not yet have high reproducibility unless they include the full code. During the rebuttal period, if they have time, they can upload the anonymized code. The following type of linux command is helpful to remove a phrase from all files in a repo:
```
grep -rl John\ Smith . | xargs sed -i '' ’s/John\ Smith/xxxx\ xxxxx/g’
```
I believe this paper can be impactful in the NAS community, but there are currently several weaknesses. I am giving this paper a weak reject, but I am willing to increase my score if the weaknesses are partially or fully addressed.

### Additional minor feedback
- Fig 5 is hard to understand. The y axis is labeled, and the caption also labels the y axis? What is the x axis?
- Section 4.1 jumps into an experiment without explaining the methodology; They should refer back to it
- What is linear correlation in fig 1?
- Line 167 “as shown in the appendix figure” Which appendix figure?
- The text (titles, axes, legends, etc) in fig. 4, 6, 7, 8, 9, 11 are too small and need to be increased
- Line 268, “actually, this is evident from the formula” should be explained in a bit more detail

References
- [1] https://arxiv.org/abs/2008.03064 (A Surgery of the Neural Architecture Evaluators)
- [2] https://arxiv.org/abs/2104.01177 (How Powerful are Performance Predictors)
- [3] https://arxiv.org/abs/2006.04492 (Speedy Performance Estimation)
- [4] https://arxiv.org/abs/2003.04276 (How to Train Your Super-Net)
- [5] https://arxiv.org/abs/2106.04010 (FEAR)
- [6] https://openreview.net/forum?id=Cnon5ezMHtu (NTK)
- [7] https://arxiv.org/abs/2011.06006 (NNGP)
- [8] https://arxiv.org/abs/2102.08099 (EPE-NAS)
- [9] https://arxiv.org/abs/2102.01063 (Zen-NAS)
- [10] https://arxiv.org/abs/2002.04289 (To Share or not to Share)
- [11] https://arxiv.org/abs/2001.01431 (Deeper Insights into Weight Sharing)

=====================

Update: After discussion, the authors have addressed my concerns and I am raising my score from 5 to 6. The most notable changes the authors claim are (1) releasing the full code, (2) additional related work discussion, (3) theoretical motivation that synflow prefers larger architectures, (4) making sure important information such as experimental setups are in the main paper, and (5) revising the claim that all ZSEs are outperformed by the param baseline.

**Time Spent Reviewing:**

5

---

> ### Author Response · Authors · 2021-08-09
> **Response to the reviewer**
>
> Thanks for your careful and valuable suggestions. We'll add discussions on the suggested references. We sincerely hope that the following responses can address your concerns.
>
> Q1: These should be added to the related work, with a short discussion of the differences [1][2][3][4][5].
>
> A1:
> * [2][5] are concurrent studies with our submission, we'll add discussion on them.
> * We thank the reviewer for pointing out the valuable reference [3], and we'll add the discussion on [3]: Our work studies more ZSEs than [3] (8 V.S. 3), including synflow, grad_norm, snip, grasp, fisher, plain, jacob_cov, relu_logdet. We also conduct more analysis on biases of ZSEs and present more criteria.
> * We'll add discussion on [4]. It is a good work on how the hyper-parameters of supernet training influence the KDs and search results of NAS (e.g., batch size, weight decay, dropout, and so on). Their work has some overlap with the one-shot part of our work. As for the differences, we report more criteria including P@top/botKs to present a better view of the estimation quality, and some findings are new. We also conduct analysis on the bias and variance of OS estimations, and we further experiment with several improving techniques from three organized perspectives (including ensemble, deiso sampling, several types of SS pruning, and so on).
>
>
> Q2: Line 56, it is expected that ZSEs would perform worse than OSEs, since ZSEs take less time to run.
>
> A2: Yes, the fact that ZSEs are worse than OSEs is intuitive, but we find that the ranking quality of current ZSEs cannot surpass the GT-param/FLOPs, which is counter-intuitive (at least to us). Maybe we should emphasize a little more on this counter-intuitive message in the introduction.
>
> Q3: Line 61, it is expected that training for longer would make one-shot estimations better.
>
> A3: Yes, we can make this point weaker if it is too intuitive.
>
> Q4: The authors claim that OSEs are better at ranking worse architectures, but the distribution of architecture accuracies is skewed left, so this is a big confounding factor in that claim.
>
> A4: Yes, you're right. We plot the GT accuracy distribution, and we think that the right-skewed GT accuracy distribution also accounts for the phenomenon that P@topK < P@botK on NB201/NB301. And on NB101, the phenomenon is actually reversed, and we attribute it to that the top GT acc distribution on NB101 is less concentrated than that on NB201/NB301. We'll change our claim to "As in most search spaces, including NB201, NB301, NDS ResNet, and NDS ResNeXt, poorly-performing architectures' accs are distributed less concentratedly while the well-performing architectures' accs are distributed more concentratedly, thus it is more likely that OSEs are better at ranking worse architectures." We'll add these new discussions into the revision.
>
> Q5: Line 153, "on larger search spaces, the training time should be longer", cannot really be concluded after looking at just 2 or 3 search spaces.
>
> A5: Thanks for the suggestion, we'll remove this unsupported claim.
>
>
> Q6: For this type of project that evaluates popular methods in the field of NAS, citing and describing all of the related work is important.
>
> * ZSE’s: [6] [7] [8] [9]
>
> A6.1: Thanks for the suggestion. We'll cite and describe all these related work, as to provide a better overview of the field.
>
> * OSE’s: [10] [11]
>
> A6.2: Thanks for the suggestion, they are indeed related to our study. We intend to discuss the following differences: [10] only uses NB101, and use SpearmanR as the ranking quality measure. Our work presents a more extensive evaluation and more analysis on more benchmarks. [11] does not use benchmarks, thus their experiments are of small scale (~200 archs), and they only use one search space (a reduced DARTS SS).
>
> Q7: During the rebuttal period, if they have time, they can upload the anonymized code.
>
> A7: Thanks for the suggestion. We're confident that our code, analysis and plotting have full reproducibility. We spend some time anonymizing our code and provide it at https://github.com/Anonymous-1112/anonymous. We provide the full codes, and some of the checkpoints/logs/plotting scripts for verification. Other checkpoints are omitted since they are too large. We still need time to clear up all the plotting scripts for easy use. We promise to release all checkpoints and plotting scripts.
>
> -----
> **Additional minor feedback**
>
> Q8: Fig 5 is hard to understand. What is the x axis?
>
> A8: Thanks for the comment. Our analysis method of complexity-level bias is to group architectures by the complexity (FLOPs or Param), and then plot the average Ranking Diff in each complexity group. The X-axis denotes the five complexity groups, with the smallest group at the leftmost. We'll add these clarifications.
>
> Q9: Section 4.1 jumps into an experiment without explaining the methodology.
>
> A9: We'll add more background knowledge of OS training (a loop of 1. sample by random/controller 2. forward 3. update) in Sec.2, and add some introducing description in Section 4.1.
>
> Q10: What is linear correlation in fig 1?
>
> A10: The linear correlation refers to the Pearson coefficient of linear correlation (Line 129).
>
> Q11: Line 167 “as shown in the appendix figure” Which appendix figure?
>
> A11: It is Fig. A1, we'll explicitly refer to it in our revision.
>
> Q12: The text (titles, axes, legends, etc) in fig. 4, 6, 7, 8, 9, 11 are too small and need to be increased.
>
> A12: Thanks. We'll fix these font issues.
>
> Q13: Line 268, “actually, this is evident from the formula” should be explained in a bit more detail.
>
> A13: Thanks for the suggestion, we'll add some explanation to demonstrate the point that "From its formula, synflow prefer larger model, which coincides with our exps", and will append it to the appendix.
>
>
> The synflow indicator proposes to change all parameters to their absolute values, remove BNs and nonlinear functions, input an all-1 tensor, add up the final feature map as the loss, and then accumulate the multiplication of the loss gradient and magnitude of all parameters. Here, we want to demonstrate three statements when introducing new convolutions into an architecture: 1) The expected loss gradients w.r.t. existing parameters become larger. 2) And since the synflow of each parameter is the multiplication of the absolute parameter value and the loss gradient (also positive), the expectation of each synflow value increases. 3) And as the number of parameters also increases, the overall synflow of the architecture increases.
>
> Since the latter two reasoning is obvious, we only need to prove the first statement: "When introducing new convolutions, the expected loss gradients w.r.t. existing parameters become larger". Our proof relies on the assumption of using the commonly-used kaiming_uniform weight initialization and proves that $E[\tilde{g}_r] = \frac{a \sqrt{K_l}}{2} E[g_r]$, where $\tilde{g}_r$ and $g_r$ are the gradients of a single weight after and before adding a layer into an architecture, respectively, and $K_l$ is the fan out of the newly added layer, and $\frac{a \sqrt{K_l}}{2}>1$ since the typical value for $a$ is $\sqrt{6}$. Due to the space limit, we cannot put the formal proof of this statement in this response, but we promise to add the formal proof to our appendix.

---

> > ### Comment · Reviewer_Q7zt · 2021-08-15
> > **Thank you for your replies**
> >
> > I thank the authors for their thorough replies to my review and the other reviews. I know it is a lot of work to do in one week.
> >
> > I agree with the authors' responses to my review. I thank the authors especially for their work in releasing all of the code, which is appreciated. It is also great that the authors will add the relevant related work, especially the papers that have a bit of overlap with their work. I also think the proof for synflow is very interesting and including the full proof would be a great addition to the paper.
> >
> > I feel more positive about the paper now, although I do have two new questions now (apologies for not asking in my original review).
> >
> > 1. I agree with reviewer SU6e that it felt like the paper tries to pack in too much information at the expense of clarity on important details such as the experimental setup for many experiments. This is also evidenced by many reviewers' clarifying questions on details in the paper. I encourage the authors to go through the paper and make sure the important information for each experiment is in the main body, possibly moving a couple lesser experiments to the appendix if needed to make room.
> >
> > 2. Thinking about it more, the claim that all ZSE's are outperformed by the "param" baseline is a very strong claim, because jacob_cov was accepted to ICML 2021 as a long talk, and synflow was accepted to ICLR 2021.
> >
> > I have also done a little bit of experiments with ZSE’s, and recently by coincidence I did the same experiment that compares the ZSE’s to params. I got the same conclusions for NB301, but on NB201 I found that jacob_cov outperforms params for Kendall’s Tau, and for Spearman I found that jacob_cov and synflow both outperform params.
> >
> > This seems like an important detail to figure out. I have a hunch to explain this. Based on the code you provided, it looks like you used the code from the Abdelfattah et al. paper for all of your ZSE experiments. Is that correct? If yes, I think that the authors have two options
> > - Remove the claim that params always outperform all ZSE's, and mention in the paper that you use the Abdelfattah et al implementation, which is not the official implementation of jacob_cov.
> > - Redo the experiments with the Mellor et al. implementation of jacob_cov. In my experience, the two implementations are different and the Mellor et al. implementation of jacob_cov performs better, and params does not outperform jacob_cov with that implementation.

---

> > > ### Author Response · Authors · 2021-08-15
> > > **Thanks for your suggestions, here are our responses.**
> > >
> > > We appreciate the reviewer's efforts in proposing the two valuable suggestions. We address these two concerns as follows, and will modify our paper accordingly.
> > >
> > > Q1. The paper tries to pack in too much information at the expense of clarity on important details such as the experimental setup for many experiments. I encourage the authors to go through the paper and make sure the important information for each experiment is in the main body, possibly moving a couple lesser experiments to the appendix if needed to make room.
> > >
> > > A1. Yes, we apologize for the missing descriptions. And we'll do some reorganization: put 1-2 figures and Eq.(1) to appendix and refer to them in the main paper. Then we'll add all necessary discussions in the main paper.
> > >
> > > Q2. Thinking about it more, the claim that all ZSE's are outperformed by the "param" baseline is a very strong claim, because jacob_cov was accepted to ICML 2021 as a long talk, and synflow was accepted to ICLR 2021. I have also done a little bit of experiments with ZSE’s, and recently by coincidence I did the same experiment that compares the ZSE’s to params. I got the same conclusions for NB301, but on NB201 I found that jacob_cov outperforms params for Kendall’s Tau, and for Spearman I found that jacob_cov and synflow both outperform params. This seems like an important detail to figure out. I have a hunch to explain this. Based on the code you provided, it looks like you used the code from the Abdelfattah et al. paper for all of your ZSE experiments. Is that correct? If yes, I think that the authors have two options: 1) Remove the claim that params always outperform all ZSE's, and mention in the paper that you use the Abdelfattah et al implementation, which is not the official implementation of jacob_cov. 2) Redo the experiments with the Mellor et al. implementation of jacob_cov. In my experience, the two implementations are different and the Mellor et al. implementation of jacob_cov performs better, and params does not outperform jacob_cov with that implementation.
> > >
> > > A2. Many thanks for your careful suggestions. We'd like to first clarify a detail: The `jacob_cov` indicator in our experiment is proposed in the first version of Mellor et al.'s paper (version 1: https://arxiv.org/abs/2006.04647v1).  And the implementation of Abdelfattah et al. for `jacob_cov` is the same as Mellor et al.'s [v1 official implementation](https://github.com/BayesWatch/nas-without-training/blob/c895924c999a053e83ddc3e4e21bc7a7ee79098b/search.py#L58). And we have incorporated your suggestions as follows:
> > >
> > > * Suggestion 2: We have implemented and evaluated the newest version of Mellor et al. in ICML 21 based on their official implementation, and our anonymous code is [here](https://github.com/Anonymous-1112/anonymous/blob/master/zero-cost-nas/foresight/pruners/measures/relu_logdet.py). To distinguish between these two versions of Mellor et al.'s work, we named the newest version `relu_logdet`. Our experimental results are similar to your findings, `relu_logdet` has a higher KD with the GT performances than Param on NB201 and NB301: 0.611 V.S. 0.606 on NB201, and 0.539 V.S. 0.515 on NB301. `relu_logdet` is indeed a well-performing indicator in this sense, and its KD even outperforms OSE on NB301: 0.539 V.S. 0.534. Nevertheless, its performances on other search spaces and top architectures are still to be improved, and we add the corresponding bias analysis on `relu_logdet` into our paper.
> > > * Suggestion 1: After finding that `relu_logdet` achieves a competitive KD with Param-GT KD on topological search spaces, we modified our description to "On NB101/NB201/NB301, the ranking qualities of OSEs surpass all ZSEs (except `relu_logdet` on NB301), and it is still very challenging for ZSEs to provide better ranking quality than Param/FLOPs: The best KD achieved by the current ZSEs except `relu_logdet` cannot beat the KD between the GT accuracies and the parameter sizes".
> > >
> > > We sincerely hope the responses can address your concerns.

---

> > > > ### Comment · Reviewer_Q7zt · 2021-08-17
> > > > **Thanks for the clarification**
> > > >
> > > > A1: That sounds good.
> > > >
> > > > A2: I see, so Abdelfattah et al. used version 1 of Mellor et al., and then Mellor et al. released version 2, `relu_logdet`. And it looks like `relu_logdet` was already in your codebase. I agree that there is only a small improvement of `relu_logdet` compared to params, but I think it is still better not to claim that all ZSE's are beaten by params (and ZSEs are comparatively better than params on other metrics such as Spearman). I agree with the sentence you suggested. It is still a good message to the community that ZSEs should be improved even more, to beat the baseline by larger margins.
> > > >
> > > > Yes, this has answered my concerns. I am increasing my score.

---

> > > > > ### Author Response · Authors · 2021-08-18
> > > > > **Thanks for your assessment**
> > > > >
> > > > > Thanks for your suggestions and assessment, they really help us to improve our paper. We'll follow the suggestions to prepare the revision.

---

### Official Review · Reviewer_WA1h · 2021-07-22

**Rating:** 7
**Confidence:** 5

**Summary:**

Weight sharing is a very important technique in NAS to speed up the search and reduce resources.
However, such technique suffers from performance drop, and attracts a lot of interset from the community to investigate into it.
This paper is one of the works that investigate the behavior and underline principles of such weight sharing technique (aka one-shot). Besides it also investigate current popular zero-shot estimators.
The paper analyzes the one-shot and zero-shot estimators from multiple perspectives on three benchmark datasets, with sufficient experiments and further suggestions.
Though some of the points/findings are not novel and well-known, the sufficient experiments still make a convening verification on these.
It will bring much to the community. The findings and suggestions will make using one-shot and zero-shot NAS easier for practitioners.

**Ethics Review Area:**

["I don’t know"]

**Main Review:**

Weight sharing is a very important technique in NAS to speed up the search and reduce resources.
Previous works analyzed the issues/problems in one-shot estimators with weight sharing from several handful points.
This work conducts a more comprehensive study on one-shot estimators, as well as recent popular zero-shot estimators, from a quantative way, with metrics such as KD, SpearmanR, Precision, BR, etc.
It contains multiple aspects within the estimators that affect the performance, compared to previous works.
Though some points are not new and well-known, the paper still conducts experiments across three representative benchmark datasets to further verify them.
Overall, this paper conducts broad and comprehensive study on one-shot and zero-shot estimators, that would benefit the community a lot.

However, there are still some issues/questions:
1. Line 51, "knowledge revealed by our work include", however, 1) and 3) are not only revealed by you.
2. Line 58, what does "inter-parameters sensitivity" mean?
3. Line 62, what does "inter-architecture difference" refer to?
4. Line 140, the A_K is a set of architectures rather than the indices. The subscript of argmin_{i\in{A_K}} seems not correct. if i\inA_{K}, then i is an architecture here, and cannot be used to index r_i.
5. The same to point 4 in Line 144.
6. In Figure 1 caption, if left Y axis if the OS loss value, then what is the right Y axis? The right Y axis cannot be metrics since the scales are large than 1.0. Is this a typo？
7. In Line 170, the finding is interesting. Do you have explanation or guess for this, because this is really important to guide the future work and practical use of NAS in different tasks.
8. In Line 190-192, what do "levels" and "smaller resolution" mean here? BTW, if 1 batch gives many ties at early epochs, it means the architectures are hard to be distinguished and ranked propersly. Then how can the metrics be higher than that of 1- batches?
9. Figure 3. It would be better if the values on the X axis are integers. Just a suggestion.
10. In Line 315-316, why is the latter one taken as the GT? I think taking the ground-truth ranking as the GT is more reasonable.
11. Line 333, how many architecture pairs do you select?
12. Line 371, what is 3/5 here? Do you mean \frac{3}{5}? If so, how does the number come?
13. Line 373, what does MC mean? It is mentioned without explanation.
14. Some points are not new to the community. Though the authors further verify them through extensive and convincing experiments, the authors should refer to those works (e.g., training longer leads to better ranking, complex network converge slower than simple ones in one-shot training, skip connections are overestimated in supernet training.)
15. Some metrics or points differ across the three benchmark datasets. Then the authors should summarize or analyze the common/different points/underline principles of the three search spaces that lead to the different behaviors. This shall be a big contribution to the community for the practitioners to apply the tricks to their tasks regarding the search space they design.
16. Why your one-shot Kendall tau results on NAS-Bench 201 in Figure 1 and Table A1 in Appendix can reach 0.744? As far as I know, previous work can only reach 0.54 [1][2] (even few-shot [2] can only reach 0.752 with 125 supernets).
17. In Appendix A.3.4, you provide two views for pruning: per-architecture and per-decision. Can you provide more explanations of their differences?
18. Figure 2 presents the accuracy/loss distribution where accuracy distribution is more concentrated than loss. But I cannot figure out why it indicated the loss carries more information about prediction confidence. Can you provide more insights or explanations?

References:
[1] Hu, Yiming, et al. "Angle-based search space shrinking for neural architecture search." European Conference on Computer Vision. Springer, Cham, 2020.
[2] Zhao, Yiyang, et al. "Few-shot neural architecture search." International Conference on Machine Learning. PMLR, 2021.

I would consider my final rating based on the authors` responses and other reviewers' comments.

Typos/Grammar:
1. Line 35, supernets -> supernet
2. Line 107, ta -> a
3. Line 137, is -> are

**Time Spent Reviewing:**

8

---

> ### Author Response · Authors · 2021-08-09
> **Response to the reviewer**
>
> Thanks for your careful and valuable suggestions. We appreciate your positive recognition of our work. Our responses are as follows.
>
> Q1: Line 51, 1) and 3) are not only revealed by you.
>
> A1: Thanks. We'll carefully check and discuss related previous studies.
>
> Q2: Line 58, what does "inter-parameters sensitivity" mean?
>
> A2: "inter-parameters sensitivity" refers to the relative sensitivity of parameters. Since indicators from the pruning literature can reflect the relative sensitivity or importance of different parameters, which helps to prune out unimportant parameters. But in NAS, we want to compare different architectures instead of different parameters, thus we suggest that architecture-level analysis is necessary, instead of adapting parameter-level analysis from the pruning literature. We agree that "Inter-parameter sensitivity" is somehow confusing, and we'll change it to "the relative sensitivity of parameters".
>
> Q3: Line 62, what does "inter-architecture difference" refer to?
>
> A3: "inter-architecture difference" refers to the GT accuracy difference between architectures. We'll add a GT distribution plot and specify this in our revision.
>
> Q4 & Q5: Line 140/144, the subscript of $argmin_{i \in {A_K}}$ seems not correct.
>
> A4 & A5: Thanks a lot, we'll fix it.
>
> Q6: In Figure 1 caption, is “left Y-axis: OS loss value” a typo？
>
> A6: Yes, this is a typo, we'll fix this to "right Y-axis".
>
> Q7 & Q18: In Line 170 and fig.2, Why loss is more informative? Can you provide more insights or explanations?
>
> A7 & A18: Fig.2 shows that on NB201, the distribution's entropies of OS acc/loss are similar (they are distributed almost symmetrically). But on NB301, the OS accuracies are distributed more concentratedly. And our point is that since loss considers the network's output distribution rather than a single label prediction, it is more informative and its distribution would be not that concentrated. We'll change our statement to be more concise. Furthermore, we analyze that the reason for this distinct phenomenon on NB301 is that the architecture GT performances on NB301 are closer, and we'll also add the GT distribution plot in the revision to help demonstrate that.
>
> Q8: In Line 190-192, what do "levels" and "smaller resolution" mean here, and how can the metrics be higher than that of 1-batches?
>
> A8: Thanks for pointing out this obscure description. This experiment demonstrates a point that if the training is not sufficient, using more data might result in worse ranking quality, especially when using the OS acc as the score. This might seem counter-intuitive at first glance. We think it is because, in the early training stages, the ability of the supernet to distinguish between close architectures is not good. Here, "close architectures" refer to architectures that have the same OS accuracy when only using 1 validation batch, i.e., in the same "level". Note that since 1 data batch only contains 128 images, thus at most 128 different OS acc levels can exist, and the legend in Fig.3(Right) shows the actual number of levels. In a word, using fewer data result in many ties, fewer levels, i.e., smaller resolution. And in the early training stages, the intra-level distinguishability of the supernet is low, thus using more data (increasing the score resolution) might bring negative impacts, while giving many tie scores can avoid making wrong comparisons between close architectures.
>
> Q9: Figure 3. It would be better if the values on the X-axis are integers.
>
> A9: Thanks, we'll modify this.
>
> Q10: In Line 315-316, why is the latter one instead of ground-truth ranking taken as the GT?
>
> A10: This experiment inspects the relative stability of OSE during training, instead of the overall ranking quality w.r.t. the actual GT. For example, even if the KD w.r.t. the actual GT is the same, the ranking for certain architectures could fluctuate during the training process (reflected by the relative stability measures).
>
> Q11: Line 333, how many architecture pairs do you select?
>
> A11: We plot the gradient visualization using all the pairs between the 6466 deiso archs.
>
> Q12: Line 371, what is 3/5 here?
>
> A12: "3/5" means "3 or 5". We'll modify this.
>
> Q13: Line 373, what does MC mean?
>
> A13: Thanks for the point, we'll add some background introduction on this in Sec.2. MC means Monte-Carlo. In each supernet training step, S architectures are sampled and their gradients are averaged, where S denotes the number of Monte-Carlo architecture samples.
>
> Q14: Improve the references.
>
> A14: Sure, we'll check related work carefully (some are kindly provided by R2) and discuss them.
>
> Q15: Some metrics or points differ across the three benchmark datasets. Then the authors should summarize or analyze the common/different points/underline principles of the three search spaces that lead to the different behaviors. This shall be a big contribution to the community for the practitioners to apply the tricks to their tasks regarding the search space they design.
>
>
> A15: Thanks for your suggestions. Indeed, there exist phenomena that are not common across all search spaces (SS). The most obvious difference is the metrics on NB101. To analyze this, we add a search space property summary table and a GT distribution figure into the appendix: "The phenomena on NB101 are different from those on NB201 and NB301: P@top 5% is higher than P@bottom 5%, and longer training after the first 20 epochs does not improve the ranking quality. The distinct phenomenon might arise from two aspects: 1) As shown in the figure (we'll add a figure on the GT accuracy distribution in the revision), the GT distribution of the top GT accuracies on NB101 is less concentrated than that on NB201 and NB301, thus it is easier to distinguish the top architectures on NB101. 2) Different from NB201 and NB301, NB101 is an operation-on-node search space. The sharing extent of supernet on operation-on-node search spaces is larger than that on operation-on-edge search spaces, since for each node, no matter which input connections are chosen, the same parameters are used. A larger sharing extent would limit the potential ranking quality of OSEs."
>
> We'll reorganize/revise the knowledge summary in our introduction to summarize the common and different points on different search spaces.
> What's more, although there exist phenomena that are not common across all search spaces, the general analysis framework and analysis methods of our work are valuable. They can provide a comprehensive view of the estimators' properties, and can be used for future studies on architecture performance estimators.
>
>
> Q16: Why your one-shot Kendall tau results on NAS-Bench 201 in Figure 1 and Table A1 in Appendix can reach 0.744? As far as I know, previous work can only reach 0.54 [1][2] (even few-shot [2] can only reach 0.752 with 125 supernets).
>
> A16: Yes, we recently find that too. We think that the large gap comes from the different training configurations: We use a plateau scheduling LR and train the supernet for 1000 epochs till convergence, as we stated in the appendix. We choose the plateau schedule since it only decays the learning rate after the training loss does not decay for a while. This would make the training more sufficient. While [2] only trains the supernet for 300 epochs using a pre-defined LR schedule instead of a dynamic plateau-based schedule.
>
> Actually, the KD of parameter size and GT is 0.647 (15k archs) and 0.606 (6466 deiso archs), and our supernet reaches a 0.731$\pm$0.020 KD when using all 15k archs, 0.744$\pm$0.025 when evaluating using 6466 deiso archs. If the scores of three supernets are averaged, the KD on 6466 deiso archs can reach 0.766. We think it would be better to compare the supernets' performances when they are trained rather sufficiently and when their KD is better than the Param-GT KD. We'll release all the checkpoints and codes, and we're glad that our work can supply strong and usable baselines. Also, as suggested by Reviewer Q7zt, we provide an anonymous code at https://github.com/Anonymous-1112/anonymous during this discussion period for verification.
>
> Q17: Explain more about per-architecture and per-decision in Appendix A.3.4.
>
> A17: Per-architecture pruning means to throw out certain architectures based on some architecture-level scores, and the outcome of the pruning process is a sub search space (SS) containing the remaining archs. For example, "Per-architecture Hard Pruning" in Appendix A.3.4 throws out the architectures with lower OS scores, while the "Per-architecture Soft Pruning" experiment on NB301 uses two types of evolutionary-based controllers to concentrate more on architecture population with higher OS scores. In contrast, per-decision pruning refers to pruning the space of architectural decisions (e.g., the available operation primitives at some position) instead of directly pruning the architecture space. We'll add more clarifications on this in our revision.
>
> Typos/Grammar: Many thanks, we'll fix the grammar.

---

> > ### Comment · Reviewer_WA1h · 2021-08-18
> > **Thanks for your reply**
> >
> > Thanks for your reply for all the questions raised and the promise to revised the paper!
> > I have one new up-coming question:
> > In A15, you say that "A larger sharing extent would limit the potential ranking quality of OSE". It is not clear to me how this affects the P@top 5% and P@bot 5%, and how this explains the inverse phenomena between NB101 and NB201/301? In my opinion, the larger sharing extent would limit the ranking quality of OSE, in both terms of P@top 5% and P@bot 5%.

---

> > > ### Author Response · Authors · 2021-08-19
> > > **Thanks for your question**
> > >
> > > Thanks for your question. The organization of statements in our A15 is misleading. Actually, reason 1 (GT accuracy distribution) results in the inverse phenomenon of P@top5% and P@bot5%, and reason 2 (more sharing extent due to the op-on-node property) results in the phenomenon that OSE does not benefit from longer training on NB101 as it does on other search spaces.
> > >
> > > In other word, our statement in A15 should be:
> > > > The phenomena on NB101 are different from those on NB201 and NB301: 1) P@top 5% is higher than P@bottom 5%, and 2) longer training after the first 20 epochs does not improve the ranking quality. These two distinct phenomena might arise from two aspects, **respectively**: 1) As shown in the figure (we'll add a figure on the GT accuracy distribution in the revision), the GT distribution of the top GT accuracies on NB101 is less concentrated than that on NB201 and NB301, thus it is easier to distinguish the top architectures on NB101. 2) Different from NB201 and NB301, NB101 is an operation-on-node search space. The sharing extent of supernet on operation-on-node search spaces is larger than that on operation-on-edge search spaces, since for each node, no matter which input connections are chosen, the same parameters are used. A larger sharing extent would limit the potential ranking quality of OSEs, thus longer training might not bring additional improvements in the ranking quality.
> > >
> > > Hope our response can address your question, and we'll modify the statement in our revision. Any suggestions on this discussion are appreciated.

---

### Decision · Program_Chairs · 2021-09-27

**Decision:**

Accept (Poster)

**Comment:**

This paper goes deep into the issues of one-shot supernet based NAS methods as well as recent zero-cost NAS methods. The authors have done an excellent job analyzing empirically the various issues and suggesting improvements to one-shot estimators like de-isomorphic sampling and the fact that zero-cost ones consistently overestimate the performance of larger networks (and the short proof of it) and so on.

During the rich discussion phase one concern that consistently came up was that the paper was packing a lot of dense information (see concerns of reviewer SU6e) and at times it was hard to read because of that. But the reviewers and authors have excellent suggestions on improving the readability. Please incorporate those aspects into the manuscript.

Reviewer Q7zt has suggested a number of improvements including citations to other relevant work and even concurrent work that should be included in the next version. Also the new results using the original implementation 'relu_logdet' and the changed conclusion with respect to number of params.

Overall this paper presents valuable insights to the NAS community!